# Characterisation of the RNA Virome of Nine *Ochlerotatus* Species in Finland

**DOI:** 10.3390/v14071489

**Published:** 2022-07-07

**Authors:** Phuoc T. Truong Nguyen, C. Lorna Culverwell, Maija T. Suvanto, Essi M. Korhonen, Ruut Uusitalo, Olli Vapalahti, Teemu Smura, Eili Huhtamo

**Affiliations:** 1Department of Virology, Medicum, University of Helsinki, Haartmaninkatu 3, FI-00290 Helsinki, Finland; lorna.culverwell@helsinki.fi (C.L.C.); maija.t.suvanto@helsinki.fi (M.T.S.); essi.m.korhonen@helsinki.fi (E.M.K.); ruut.uusitalo@helsinki.fi (R.U.); olli.vapalahti@helsinki.fi (O.V.); teemu.smura@helsinki.fi (T.S.); eili.huhtamo@helsinki.fi (E.H.); 2The Natural History Museum, Cromwell Road, South Kensington, London SW5 7BD, UK; 3Department of Veterinary Biosciences, Faculty of Veterinary Medicine, University of Helsinki, Agnes Sjöbergin Katu 2, P.O. Box 66, FI-00014 Helsinki, Finland; 4Department of Geosciences and Geography, Faculty of Science, University of Helsinki, Gustaf Hällströmin Katu 2, P.O. Box 64, FI-00014 Helsinki, Finland; 5Virology and Immunology, Diagnostic Center, HUSLAB, Helsinki University Hospital, FI-00029 Helsinki, Finland

**Keywords:** Aedini, mosquito virome, *Ochlerotatus*, NGS, Finland, RNA virus, *Sindbis virus*

## Abstract

RNA viromes of nine commonly encountered *Ochlerotatus* mosquito species collected around Finland in 2015 and 2017 were studied using next-generation sequencing. Mosquito homogenates were sequenced from 91 pools comprising 16–60 morphologically identified adult females of *Oc. cantans*, *Oc. caspius*, *Oc. communis*, *Oc. diantaeus*, *Oc. excrucians*, *Oc. hexodontus*, *Oc. intrudens*, *Oc. pullatus* and *Oc. punctor*/*punctodes*. In total 514 viral Reverse dependent RNA polymerase (RdRp) sequences of 159 virus species were recovered, belonging to 25 families or equivalent rank, as follows: *Aliusviridae*, *Aspiviridae*, *Botybirnavirus*, *Chrysoviridae*, *Chuviridae*, *Endornaviridae*, *Flaviviridae*, *Iflaviridae*, *Negevirus*, *Partitiviridae*, *Permutotetraviridae*, *Phasmaviridae*, *Phenuiviridae*, *Picornaviridae*, *Qinviridae*, *Quenyavirus*, *Rhabdoviridae*, *Sedoreoviridae*, *Solemoviridae*, *Spinareoviridae*, *Togaviridae*, *Totiviridae*, *Virgaviridae*, *Xinmoviridae* and *Yueviridae*. Of these, 147 are tentatively novel viruses. One sequence of *Sindbis virus*, which causes Pogosta disease in humans, was detected from *Oc. communis* from Pohjois-Karjala. This study greatly increases the number of mosquito-associated viruses known from Finland and presents the northern-most mosquito-associated viruses in Europe to date.

## 1. Introduction

Mosquitoes (Diptera, Culicidae) are vectors of a variety of medically significant pathogens worldwide. The known endemic mosquito-borne viral pathogens in Finland are *Sindbis virus* (*Togaviridae*: *Alphavirus*) [1,2], Inkoo virus [3,4] and Chatanga virus (*Peribunyaviridae*: *Orthobunyavirus*) [5,6]. Three insect-associated flaviviruses are present in the southern half of the country: Lammi virus [7], Hanko virus [8] and Ilomantsi virus [9]. Of these, Lammi and Ilomantsi viruses represent a separate flavivirus group genetically associated with vector-borne flaviviruses. Most recently, a novel *Negevirus* was isolated from mosquitoes collected in eastern Finland [10].

Forty-three species of mosquitoes are recorded from Finland, which belong to *Aedes*, *Aedimorphus*, *Culex*, *Culiseta*, *Dahliana* and *Ochlerotatus* [11]. Some species have rarely been encountered during recent or historical collections, but most have been reported as human-biting either in Finland or in neighbouring countries [11,12]. Species of the genus *Ochlerotatus* are most numerous, with 23 recorded from across Finland, but distributions vary according to species-specific life strategies. Halophilic species, including *Oc. caspius* and *Oc. dorsalis*, are usually restricted to costal locations, while other species, including *Oc. communis* and *Oc. punctor*, are widely distributed across the entire country. In Lapland, the dominant human-biting species include *Oc. communis*, *Oc. excrucians*, *Oc. hexodontus*, *Oc. impiger*, *Oc. nigripes*, *Oc. pullatus* and *Oc. punctor/punctodes.* Further south, the most commonly encountered human-biting species include *Oc. cantans*, *Oc. caspius*, *Oc. communis*, *Oc. diantaeus*, *Oc. intrudens*, *Oc. pullatus* and *Oc. punctor/punctodes.* Mosquitoes, generally, are abundant in summer months between June and August, but only *Ochlerotatus* species have been associated with the known mosquito-borne pathogens in Finland, although vector species associations are not yet confirmed. *Sindbis virus* has been isolated from mosquitoes twice: once from a pool of unidentified specimens at least containing *Ochlerotatus* [1] and again, from a pool of 13 specimens morphologically identified as species of *Ochlerotatus* [2]. Californian serogroup orthobunyaviruses Inkoo virus and Chatanga virus have also been isolated from *Ochlerotatus* species. Inkoo virus was first identified from *Oc. communis* and/or *Oc. punctor*/*punctodes* from Inkoo in southern Finland [4], while Chatanga virus was originally isolated in eastern Finland from pooled unidentified specimens, likely including specimens of *Ochlerotatus* or *Aedes* [5].

More broadly, *Ochlerotatus* is a widely distributed genus, with 199 species located in tropical, subtropical and temperate regions, and is known to include several species which are naturally infected with arboviruses [13]. In Europe, four native species of *Ochlerotatus*, *Oc. caspius*, *Oc. communis*, *Oc. dorsalis* and *Oc. excrucians*, are classed as being of particular interest for targeted surveillance due to their vector potential for a series of listed pathogens by the European Centre for Disease Prevention and Control [14]. Although other known vector species are present in Finland, e.g., *Culex pipiens*, it is of interest to first pursue the study of identified *Ochlerotatus* species in order to ascertain their potential virus associations, particularly when they have been implicated as vectors for all three endemic mosquito-borne viruses in Finland and are regularly attracted to humans. As such, females of nine commonly encountered species, *Oc. cantans*, *Oc. caspius*, *Oc. communis*, *Oc. diantaeus*, *Oc. excrucians*, *Oc. hexodontus*, *Oc. intrudens*, *Oc. pullatus* and *Oc. punctor/punctodes* were chosen from suitable specimens that were collected for a nation-wide distribution study [11] for inclusion in NGS studies to analyse their RNA viromes. From 91 pools of identified adult female *Ochlerotatus* mosquitoes that were collected from a variety of habitats around Finland in summer 2015 and 2017, 514 unique sequences of RNA-dependent RNA polymerase (RdRp) > 1000 nt, belonging to 159 viruses, were recovered. Of these, 147 potentially novel viruses were identified as well as sequences belonging to 12 established viruses, including *Sindbis virus*. Final decisions on the taxonomic placement and species’ status of these viruses will be determined by the ICTV.

## 2. Materials and Methods

### 2.1. Mosquito Collection and Identification

As part of a larger study, 52,466 mosquitoes were collected from around Finland between 2012 and 2018, using a variety of collection methods and from a multitude of different habitats [11]. The primary goal of that study was to collect distribution data for each of the native species, and the secondary aim was to collect specimens that were suitable for other studies. Each of the 1031 collections were numbered with a unique running code prefixed with “FI” (Figure 1, Table 1). Specimens were stored and processed in several ways, such that they could be used in one or more distribution, morphology, genetics or virus studies. Multiple factors, including access to dry ice, RNAlater or specialist freezers, time available for processing, whether the specimens were rare or common, and whether they were alive when reaching field stations, affected their designation for virus or other studies. In total, 18,394 specimens were not suitable for virus studies; 15,096 specimens were stored in RNA stabilisation solutions, including RNAlater; and 18,976 specimens were deep frozen at −70 °C or colder. Deep-frozen specimens were processed along a cold chain of initially −20 °C, −70 °C or on dry ice, and transported in dry ice to storage at −80 °C prior to the study. Mosquitoes were identified over dry ice using morphological keys [15,16] and then either (i) pooled by species, or (ii) stored individually in 1.2 mL collection microtubes (QIAGEN, Venlo, The Netherlands). From the 18,976 deep-frozen specimens, 14,092 were collected as adults, of which 13,927 were females, and 11,835 were adult female *Ochlerotatus*. A subset of 2333 of these deep-frozen adult female specimens was chosen for inclusion in this study (see below). Notes were made if any specimens were visibly engorged with blood, or if they had ectoparasites (Acarid mites).

### 2.2. Pooling and Homogenisation

Pools were constructed using identifiable females of commonly encountered human-biting *Ochlerotatus*, by species, collection location and collection date (Figure 1, Table 1). Rare species with fewer than 16 specimens were not considered; neither were specimens which were found in low numbers over several collection sites over several years such that location or temporal data would not be confused in the results. Since these species are difficult to identify when scales are denuded, 2176 specimens were immediately excluded from the potential specimens as they were either unidentified or the identification was not confirmed. To suit the available resources, 2333 females belonging to nine species, which were collected in May–August 2015 and July–August 2017, met these criteria, and were divided into 91 pools, as follows: *Oc. cantans* (*n* = 1), *Oc. caspius* (*n* = 11), *Oc. communis* (*n* = 35), *Oc. diantaeus* (*n* = 6), *Oc. excrucians* (*n* = 3), *Oc. hexodontus* (*n* = 8), *Oc. intrudens* (*n* = 14), *Oc. pullatus* (*n* = 2) and *Oc. punctor/punctodes* (*n* = 11) (Table 1).

**Table 1 viruses-14-01489-t001:** Details of the 91 mosquito pools included in this study by collection site (see Figure 1). Pools shaded grey were made up of specimens from more than one collection. Where several collections were combined, the “number of specimens from a collection/total number of specimens in the pool” are given.

Collection No.	Latitude (N)	Longitude (E)	Location-Pool No.	No. of Specimens	Collection Date	Mosquito Species
FI 432	61.0766	24.3912	FIN/KH-2018/029	30	27 May 2015	*Oc. pullatus*
			FIN/KH-2018/047	20	27 May 2015	*Oc. punctor/punctodes*
FI 437	61.0285	24.4596	FIN/KH-2018/048	20	02 June 2015	*Oc. communis*
FI 441	61.0201	24.4877	FIN/KH-2018/038	13/20	02 June 2015	*Oc. intrudens*
			FIN/KH-2018/049	24	02 June 2015	*Oc. communis*
FI 442	61.0223	24.4912	FIN/KH-2018/038	7/20	02 June 2015	*Oc. intrudens*
FI 474	59.8372	23.1595	FIN/U-2018/050	20	14 June 2015	*Oc. communis*
FI 483	63.0630	21.5680	FIN/Po-2018/022	24	16 June 2015	*Oc. communis*
FI 487	63.0410	21.3539	FIN/Po-2018/009	27	16 June 2015	*Oc. excrucians*
FI 500	63.6071	22.7055	FIN/Po-2018/031	20	17 June 2015	*Oc. communis*
FI 505	64.1637	23.6876	FIN/PP-2018/010	60	17 June 2015	*Oc. communis*
FI 513	63.6039	24.7534	FIN/KP-2018/032	25	18 June 2015	*Oc. communis*
			FIN/KP-2018/033	16	18 June 2015	*Oc. diantaeus*
			FIN/KP-2018/034	20	18 June 2015	*Oc. intrudens*
FI 520	62.7665	24.6814	FIN/KS-2018/035	24	18 June 2015	*Oc. communis*
FI 525	61.3473	24.7655	FIN/Pi-2018/051	20	19 June 2015	*Oc. communis*
			FIN/Pi-2018/052	20	19 June 2015	*Oc. communis*
			FIN/Pi-2018/053	20	19 June 2015	*Oc. communis*
			FIN/Pi-2018/054	20	19 June 2015	*Oc. communis*
			FIN/Pi-2018/055	21	19 June 2015	*Oc. communis*
FI 531	61.2013	28.9019	FIN/EK-2018/056	22	25 June 2015	*Oc. communis*
FI 532	62.7189	31.0050	FIN/PK-2018/041	9/24	25 June 2015	*Oc. hexodontus*
			FIN/PK-2018/057	20	25 June 2015	*Oc. intrudens*
			FIN/PK-2018/058	20	25 June 2015	*Oc. diantaeus*
			FIN/PK-2018/059	20	25 June 2015	*Oc. communis*
			FIN/PK-2018/060	20	25 June 2015	*Oc. communis*
			FIN/PK-2018/061	20	25 June 2015	*Oc. intrudens*
FI 537	62.7189	31.0050	FIN/PK-2018/011	60	26 June 2015	*Oc. punctor/punctodes*
			FIN/PK-2018/041	15/24	26 June 2015	*Oc. hexodontus*
			FIN/PK-2018/042	20	26 June 2015	*Oc. cantans*
			FIN/PK-2018/062	20	26 June 2015	*Oc. communis*
			FIN/PK-2018/063	20	26 June 2015	*Oc. diantaeus*
			FIN/PK-2018/064	20	26 June 2015	*Oc. diantaeus*
			FIN/PK-2018/065	20	26 June 2015	*Oc. intrudens*
			FIN/PK-2018/066	20	26 June 2015	*Oc. intrudens*
			FIN/PK-2018/067	20	26 June 2015	*Oc. punctor/punctodes*
			FIN/PK-2018/068	20	26 June 2015	*Oc. intrudens*
			FIN/PK-2018/069	20	26 June 2015	*Oc. intrudens*
			FIN/PK-2018/070	20	26 June 2015	*Oc. communis*
			FIN/PK-2018/071	18	26 June 2015	*Oc. punctor/punctodes*
FI 538	62.7700	30.9733	FIN/PK-2018/072	20	26 June 2015	*Oc. intrudens*
			FIN/PK-2018/073	20	26 June 2015	*Oc. intrudens*
FI 540	62.7666	31.1629	FIN/PK-2018/021	24	26 June 2015	*Oc. communis*
FI 550	62.7650	30.3541	FIN/PK-2018/036	20	27 June 2015	*Oc. communis*
			FIN/PK-2018/074	20	27 June 2015	*Oc. communis*
			FIN/PK-2018/075	20	27 June 2015	*Oc. intrudens*
			FIN/PK-2018/076	20	27 June 2015	*Oc. communis*
			FIN/PK-2018/077	20	27 June 2015	*Oc. communis*
			FIN/PK-2018/078	20	27 June 2015	*Oc. communis*
			FIN/PK-2018/079	20	27 June 2015	*Oc. communis*
FI 551	62.7241	30.8721	FIN/PK-2018/080	21	27 June 2015	*Oc. intrudens*
FI 566	65.1798	25.8002	FIN/PP-2018/020	16	03 July 2015	*Oc. diantaeus*
FI 571	67.6588	24.9049	FIN/L-2018/008	48	03 July 2015	*Oc. intrudens*
FI 575	68.4076	23.8850	FIN/L-2018/005	32/48	04 July 2015	*Oc. communis*
			FIN/L-2018/027	8/24	04 July 2015	*Oc. communis*
FI 582	69.0870	20.7600	FIN/L-2018/005	8/48	02 July 2015	*Oc. communis*
FI 607	69.7904	27.0549	FIN/L-2018/001	48	07 July 2015	*Oc. hexodontus*
			FIN/L-2018/006	48	07 July 2015	*Oc. communis*
FI 618	66.3588	29.3260	FIN/PP-2018/015	40/57	09 July 2015	*Oc. punctor/punctodes*
			FIN/PP-2018/28	20	09 July 2015	*Oc. intrudens*
FI 620	66.3639	29.3429	FIN/PP-2018/015	17/57	09 July 2015	*Oc. punctor/punctodes*
			FIN/PP-2018/016	60	09 July 2015	*Oc. communis*
FI 641	66.1148	29.1976	FIN/PP-2018/082	20	18 July 2015	*Oc. communis*
			FIN/PP-2018/083	17	18 July 2015	*Oc. communis*
FI 642	66.4756	29.0116	FIN/L-2018/024	10/24	19 July 2015	*Oc. communis*
FI 648	66.4597	28.8963	FIN/L-2018/024	14/24	19 July 2015	*Oc. communis*
FI 649	69.2558	27.2301	FIN/L-2018/007	40/48	22 July 2015	*Oc. excrucians*
			FIN/L-2018/084	24	22 July 2015	*Oc. excrucians*
			FIN/L-2018/085	20	22 July 2015	*Oc. hexodontus*
			FIN/L-2018/086	20	22 July 2015	*Oc. hexodontus*
FI 652	68.9008	27.0658	FIN/L-2018/023	8/16	22 July 2015	*Oc. pullatus*
FI 654	69.6249	29.0415	FIN/L-2018/019	4/16	23 July 2015	*Oc. diantaeus*
			FIN/L-2018/007	1/48	23 July 2015	*Oc. excrucians*
FI 655	69.5095	28.5965	FIN/L-2018/019	12/16	23 July 2015	*Oc. diantaeus*
			FIN/L-2018/007	7/48	23 July 2015	*Oc. excrucians*
FI 663	69.4178	26.1809	FIN/L-2018/088	21	24 July 2015	*Oc. communis*
FI 671	69.0617	20.7936	FIN/L-2018/002	48	26 July 2015	*Oc. hexodontus*
			FIN/L-2018/003	48	26 July 2015	*Oc. punctor/punctodes*
			FIN/L-2018/026	24	26 July 2015	*Oc. punctor/punctodes*
FI 674	69.0205	20.9304	FIN/L-2018/089	20	28 July 2015	*Oc. hexodontus*
			FIN/L-2018/090	20	28 July 2015	*Oc. hexodontus*
FI 675	69.0227	20.9380	FIN/L-2018/030	22	28 July 2015	*Oc. hexodontus*
FI 701	65.6855	29.1345	FIN/PP-2018/004	48	23 August 2015	*Oc. punctor/punctodes*
FI 728	68.9490	20.9210	FIN/L-2018/005	8/48	02 July 2015	*Oc. communis*
			FIN/L-2018/023	8/16	02 July 2015	*Oc. pullatus*
FI 730	68.7270	21.4220	FIN/L-2018/027	16/24	03 July 2015	*Oc. communis?*
FI 976	61.0569	28.6785	FIN/EK-2018/040	20	04 July 2017	*Oc. communis*
	61.0569	28.6785	FIN/EK-2018/091	20	04 July 2017	*Oc. communis*
FI 988	60.5481	21.3696	FIN/VS-2018/017	60	11 July 2017	*Oc. caspius*
FI 1009	59.8439	23.2466	FIN/U-2018/092	20	22 August 2017	*Oc. caspius*
			FIN/U-2018/093	17	22 August 2017	*Oc. punctor/punctodes*
FI 1010	59.8439	23.2466	FIN/U-2018/018	60	22–23 August 2017	*Oc. caspius*
			FIN/U-2018/039	25	22–23 August 2017	*Oc. punctor/punctodes*
			FIN/U-2018/094	20	22–23 August 2017	*Oc. caspius*
			FIN/U-2018/095	20	22–23 August 2017	*Oc. caspius*
FI 1011	59.8439	23.2466	FIN/U-2018/044	20	23–24 August 2017	*Oc. caspius*
			FIN/U-2018/045	21	23–24 August 2017	*Oc. punctor/punctodes*
			FIN/U-2018/096	20	23–24 August 2017	*Oc. caspius*
			FIN/U-2018/097	19	23–24 August 2017	*Oc. caspius*
FI 1015	60.5481	21.3696	FIN/VS-2018/098	20	24 August 2017	*Oc. caspius*
			FIN/VS-2018/099	20	24 August 2017	*Oc. caspius*
			FIN/VS-2018/100	26	24 August 2017	*Oc. caspius*

Pools varied in size, from 16–60 whole individuals, with most later pools comprising 20 specimens. Females that were noticeably blood fed or gravid, or which had one or more ectoparasites were maintained in individual tubes for homogenisation. Pools were assigned a running number corresponding to the date when they were processed, from FIN/L-2018/001 to FIN/VS-2018/100 (Table 1). Most pools comprised mosquitoes from a single collection site, but several contained specimens from up to three locations. In these few cases, specimens were pooled from the same region and within a few days of being collected.

For the purpose of interpreting the collection locations when reading the phylogenetic trees, an additional code was added after “FIN” to represent the 11 (of 19) regions of Finland from which collections were made, as follows: EK, Etelä-Karjala; KH, Kanta-Häme; Kl, Kymenlaakso; KP, Keski-Pohjanmaa; KS, Keski-Suomi; L, Lappi/Lapland; PK, Pohjois-Karjala; Pi, Pirkanmaa; Po, Pohjanmaa; PP, Pohjois-Pohjanmaa; U, Uusimaa; and VS, Varsinais-Suomi.

Individually stored specimens were homogenised in microtubes with 100 µL of Dulbecco’s phosphate-buffered saline (PBS) + 0.2% bovine serum albumin (BSA), sterile sand and a 3 mm tungsten carbide bead (QIAGEN, Venlo, The Netherlands). After homogenisation, the tubes were centrifuged at full speed for 5 min at 5 °C. Subsequently, 50 µL of supernatant from each specimen was then combined in a “super pool”. For pre-pooled mosquitoes, 1.8 mL of Dulbecco’s PBS + 0.2% BSA was added to each 2 mL tube, with a 5 mm tungsten carbide bead. These were homogenised using the QIAGEN TissueLyser II for 2 min at full speed, then centrifuged at 5 °C for 5 min. From each of the 91 pooled mosquito homogenates, aliquots were taken for next-generation sequencing (NGS).

### 2.3. Illumina MiSeq Sequencing

Prior to sequencing, the mosquito homogenates were treated following an established protocol [17] with minor modifications. Specifically, they were each filtered through a 0.8 µm polyethersulfone (PES) filter and treated with micrococcal nuclease (New England Biolabs, Ipswich, MA, USA) and benzonase (Millipore, Merck KGaA, Darmstadt, Germany). RNA was then extracted using TRIzol (Invitrogen, Thermo Fisher Scientific, Waltham, MA, USA) according to the manufacturer’s instructions. The RNA samples were treated with DNase I and purified with Agencourt RNA Clean XP magnetic beads (Beckman Life Industries). Ribosomal RNA was removed using a NEBNext rRNA depletion kit according to the manufacturer’s protocol, followed by amplification using a whole transcriptome amplification WTA2 kit (Sigma-Aldrich, Merck KGaA, Darmstadt, Germany). The sequencing libraries were prepared using a Nextera XT kit (Illumina, San Diego, CA, USA) and sequenced using the Illumina Miseq platform and v2 reagent kit with 150 bp paired-end reads.

### 2.4. NGS Data Analysis

Sequence reads from the initial homogenates (Appendix A) were analysed in Lazypipe v.1.2, an automated bioinformatics pipeline [18]. Preassembly quality control was first performed on the FASTQ reads using Trimmomatic v.0.39 [19] to remove and trim low quality reads, bases and Illumina adapters. MEGAHIT v.1.2.8 [20] was used to perform *de novo* assembly with the initial quality-controlled reads. Gene-like regions were detected using MetaGeneAnnotator [21] and translated to amino acids with BioPerl [22]. The amino acid sequences were then queried against the UniProtKB database using SANSparallel [23] and assigned NCBI taxonomy IDs. Any sequences that were unclassified according to NCBI Taxonomy were not possible to identify following the steps, above, so were manually identified using BLASTx. Any contigs longer than 1000 nt, with the highest similarity to viral RNA-dependent RNA polymerases (RdRps), were selected for phylogenetic analyses.

Analyses were performed on amino acid sequences, which were derived by analysing each contig with getorf [24] to identify open reading frames (ORFs) and converting them into an amino acid format. These were aligned with MAFFT v. 7.490 [25] and the resulting alignments trimmed with trimal v.1.2 [26]. Finally, maximum likelihood (ML) trees were constructed with IQ-TREE2 v.2 [27], which employs the ModelFinder algorithm [28] to determine the optimal protein substitution model, and the UFBoot2 algorithm [29] to compute 1000 bootstraps. The final trees were visualised in R v.4.1.2 using the GGTREE package v.3.0.4 [30].

The novel viruses discovered in this study (Appendix A) were named according to the nearest town or municipality to the, or one of the site(s) from which the mosquitoes were collected, but with diacritical marks removed as they were not supported in GGTREE. If more than one virus variant or species was found from the same pool an additional, final, running number was appended to the end. Representative virus sequences for each virus family were downloaded from those available in GenBank, compared to newly generated sequences, and included in the ML trees.

## 3. Results

### 3.1. RNA Viromes Obtained Directly from Mosquito Homogenates

#### 3.1.1. Positive-Sense ssRNA Virus Sequences

Positive-sense ssRNA viruses belonging to eight established viral families were detected during this study; *Endornaviridae, Flaviviridae*, *Iflaviridae*, *Permutotetraviridae*, *Picornaviridae*, *Solemoviridae*, *Togaviridae* and *Virgaviridae*. Sequences which belong to two proposed taxa, *Negevirus* and *Quenyavirus* were also recovered. The +ssRNA viruses are listed below, with all variant names and associated mosquito species in Table 2.

**Table 2 viruses-14-01489-t002:** +ssRNA viruses sequenced from Finnish mosquitoes. Previously described viruses are shaded grey. Where more than one virus was sequenced from a pool, an additional code was appended to the pool number.

Virus Family/ Taxon	Virus Name	Pool/Variant No.	Associated Mosquito Species	GenBank Accession
*Endornaviridae*	Hallsjon virus	FIN/U-2018/93	*Oc. punctor/punctodes*	ON955055
*Endornaviridae*	Tvarminne alphaendornavirus	FIN/U-2018/93	*Oc. punctor/punctodes*	ON955056
*Flaviviridae*	Hameenlinna flavivirus	FIN/KH-2018/38	*Oc. intrudens*	ON955057
*Flaviviridae*	Kilpisjarvi flavivirus	FIN/L-2018/90	*Oc. hexodontus*	ON949931
*Flaviviridae*	Lestijarvi flavi-like virus	FIN/KP-2018/33	*Oc. diantaeus*	ON955060
*Flaviviridae*	Hanko virus	FIN/U-2018/94 FIN/U-2018/95 FIN/U-2018/96FIN/U-2018/97	*Oc. caspius* *Oc. caspius* *Oc. caspius* *Oc. caspius*	ON949927 ON949928 ON949929 ON949930
*Flaviviridae*	Inari jingmenvirus	FIN/L-2018/30FIN/L-2018/86	*Oc. hexodontusOc. hexodontus*	ON955058 ON955059
*Iflaviridae*	Enontekio iflavirus	FIN/L-2018/02-1 FIN/L-2018/02-2 FIN/L-2018/89	*Oc. hexodontusOc. hexodontus* *Oc. hexodontus*	ON955061 ON955062 ON949932
*Iflaviridae*	Hanko iflavirus 1	FIN/PK-2018/11 FIN/L-2018/24 FIN/L-2018/27 FIN/PP-2018/28 FIN/U-2018/50 FIN/PK-2018/66 FIN/PK-2018/80	*Oc. punctor/punctodes* *Oc. communis* *Oc. communis* *Oc. intrudens* *Oc. communis* *Oc. intrudens* *Oc. intrudens*	ON949934 ON955063 ON949933 ON949936 ON949937 ON949935 ON955064
*Iflaviridae*	Hanko iflavirus 2	FIN/U-2018/94 FIN/U-2018/97	*Oc. caspius* *Oc. caspius*	ON955065 ON949938
*Iflaviridae*	Mekrijarvi iflavirus	FIN/PK-2018/69	*Oc. intrudens*	ON949939
*Iflaviridae*	Pedersore iflavirus	FIN/Po-2018/31 FIN/KP-2018/33 FIN/U-2018/92 FIN/U-2018/94	*Oc. communis* *Oc. diantaeus* *Oc. caspius* *Oc. caspius*	ON949941 ON949940 ON949942 ON955066
*Negevirus*	Cordoba virus	FIN/L-2018/02 FIN/PP-2018/04-1 FIN/PP-2018/04-2 FIN/PP-2018/04-3 FIN/L-2018/06 FIN/PP-2018/16-1 FIN/PP-2018/16-2 FIN/PP-2018/82-1 FIN/PP-2018/82-2 FIN/PP-2018/82-3 FIN/PP-2018/82-4	*Oc. hexodontus* *Oc. punctor/punctodes* *Oc. punctor/punctodes* *Oc. punctor/punctodes* *Oc. communis* *Oc. communis* *Oc. communis* *Oc. communis* *Oc. communis* *Oc. communis* *Oc. communis*	ON955067 ON955069 ON955070 ON955071 ON955068 ON955072 ON955073 ON955074 ON955075 ON955076 ON955077
*Negevirus*	Dezidougou virus	FIN/PP-2018/82	*Oc. communis*	ON949943
*Negevirus*	Kustavi negevirus	FIN/VS-2018/100	*Oc. caspius*	ON949944
*Negevirus*	Mekrijärvi negevirus	FIN/PK-2018/41-1 FIN/PK-2018/41-2 FIN/PK-2018/68 FIN/PK-2018/69	*Oc. hexodontus* *Oc. hexodontus* *Oc. intrudens* *Oc. intrudens*	ON955078 ON955079 ON955080 ON955081
*Negevirus*	Utsjoki negevirus 1	FIN/L-2018/02-1	*Oc. hexodontus*	ON955082
		FIN/L-2018/02-2	*Oc. hexodontus*	ON955083
		FIN/L-2018/02-3	*Oc. hexodontus*	ON955084
		FIN/L-2018/03-1	*Oc. punctor/punctodes*	ON955085
		FIN/L-2018/03-2	*Oc. punctor/punctodes*	ON955086
		FIN/PP-2018/04-1	*Oc. punctor/punctodes*	ON955088
*Negevirus*	Utsjoki negevirus 1	FIN/PP-2018/04-2 FIN/PP-2018/04-3 FIN/PP-2018/04-4 FIN/U-2018/06 FIN/PP-2018/16 FIN/PP-2018/82 FIN/L-2018/84 FIN/L-2018/85 FIN/L-2018/90	*Oc. punctor/punctodes* *Oc. punctor/punctodes* *Oc. punctor/punctodes* *Oc. communis* *Oc. communis* *Oc. communis* *Oc. excrucians* *Oc. hexodontus* *Oc. hexodontus*	ON955089 ON955090 ON955091 ON949945 ON955092 ON949948 ON955087 ON949946 ON949947
*Negevirus*	Utsjoki negevirus 2	FIN/L-2018/02-1 FIN/L-2018/02-2 FIN/L-2018/02-3 FIN/PP-2018/04-1 FIN/PP-2018/04-2 FIN/L-2018/06 FIN/L-2018/85	*Oc. hexodontus* *Oc. hexodontus* *Oc. hexodontus* *Oc. punctor/punctodes* *Oc. punctor/punctodes* *Oc. communis* *Oc. hexodontus*	ON955093 ON955094 ON955095 ON955098 ON955099 ON955096 ON955097
*Negevirus*	Utsjoki negevirus 3	FIN/L-2018/02 FIN/L-2018/06	*Oc. hexodontus* *Oc. communis*	ON955100 ON955101
*Permutotetraviridae*	Inari permutotetravirus	FIN/PP-2018/04 FIN/L-2018/07-1 FIN/L-2018/07-2 FIN/L-2018/85 FIN/L-2018/86 FIN/L-2018/89	*Oc. punctor/punctodes* *Oc. excrucians * *Oc. excrucians* *Oc. hexodontus* *Oc. hexodontus* *Oc. hexodontus*	ON955107 ON955102 ON955103 ON955104 ON955105 ON955106
*Picornaviridae*	Hanko picorna-like virus	FIN/U-2018/92-1 FIN/U-2018/92-2	*Oc. caspius* *Oc. caspius*	ON955108 ON955109
*Picornaviridae*	Jotan virus	FIN/VS-2018/99-1 FIN/VS-2018/99-2 FIN/VS-2018/99-3	*Oc. caspius* *Oc. caspius* *Oc. caspius*	ON955110 ON955111 ON955112
*Quenyavirus*	Enontekio quenyavirus	FIN/L-2018/90 FIN/U-2018/93	*Oc. hexodontus* *Oc. punctor/punctodes*	ON955113 ON955114
*Solemoviridae*	Enontekio sobemovirus	FIN/L-2018/02 FIN/L-2018/26 FIN/L-2018/89	*Oc. hexodontus* *Oc. punctor/punctodes* *Oc. hexodontus*	ON955115 ON955116 ON955117
*Solemoviridae*	Evros sobemo-like virus	FIN/VS-2018/17 FIN/U-2018/18 FIN/U-2018/92 FIN/U-2018/94 FIN/U-2018/95 FIN/U-2018/98	*Oc. caspius* *Oc. caspius* *Oc. caspius* *Oc. caspius* *Oc. caspius* *Oc. caspius*	ON955122 ON955118 ON955119 ON955120 ON955121 ON955123
*Solemoviridae*	Hanko sobemovirus	FIN/U-2018/96	*Oc. caspius*	ON955124
*Solemoviridae*	Ilomantsi sobemovirus	FIN/L-2018/07 FIN/PK-2018/42	*Oc. excrucians* *Oc. cantans*	ON955125 ON955126
*Solemoviridae*	Joensuu sobemovirus	FIN/L-2018/19 FIN/PK-2018/75 FIN/PP-2018/82	*Oc. diantaeus* *Oc. intrudens* *Oc. communis*	ON955127 ON955128 ON955129
*Togaviridae*	Sindbis virus	FIN/PK-2018/62	*Oc. communis*	ON955130
*Virgaviridae*	Enontekio virga-like virus 1	FIN/L-2018/90	*Oc. hexodontus*	ON955131
*Virgaviridae*	Enontekio virga-like virus 2	FIN/L-2018/90	*Oc. hexodontus*	ON955132
*Virgaviridae*	Pedersore virga-like virus	FIN/Po-2018/31 FIN/EK-2018/40 FIN/L-2018/88 FIN/L-2018/90 FIN/U-2018/93	*Oc. communis* *Oc. communis* *Oc. communis* *Oc. hexodontus* *Oc. punctor/punctodes*	ON955136 ON955133 ON955134 ON955135 ON955137

Two species belonging to *Alphaendornavirus* in *Endornaviridae*, a family of viruses known to infect plants, fungi and oomycetes, were recovered from one pool of *Oc. punctor*/*punctodes* (Figure 2, Table 2). The first was a strain of Hallsjon virus (GenBank accession: QGA70950.1; amino acid identity: 99.77%) and the second was a novel virus, named “Tvarminne alphaendornavirus”, that was distantly similar to *Vicia faba alphaendornavirus* (GenBank accession: YP_438201.1; amino acid identity: 49.12%). Complete genomes were sequenced for both virus species (GenBank accessions ON955055 and ON955056).

Five species belonging to two genera of *Flaviviridae* were sequenced from nine mosquito pools, four of which are tentative novel viruses (Figure 3, Table 2). Three viruses grouped within genus *Flavivirus*, one with flavivirus-like viruses and one within genus *Jingmenvirus*. Two of the four novel species were named “Hameenlinna flavivirus” and “Kilpisjarvi flavivirus” and these fell within the insect-specific group of flaviviruses. Hameenlinna flavivirus was most similar to another insect-specific flavivirus species that was first detected in Finland, Hanko virus (GenBank accession: YP_009259489.1; amino acid identity: 79.87%). Kilpisjarvi flavivirus was most similar to Xishuangbanna aedes flavivirus (GenBank accession: YP_009350102.1; amino acid identity: 61.88%) although it clustered with Ochlerotatus scapularis flavivirus (GenBank accession: BCI56825.1; amino acid identity: 61.37%) in the phylogenetic tree. The full genome of Kilpisjarvi flavivirus was sequenced (GenBank accessions ON949931). A novel flavivirus-like species, “Lestijarvi flavi-like virus”, was most similar to Hymenopteran flavi-related virus (GenBank accession: QTJ63659.1; amino acid identity: 47.75%), although in the phylogenetic tree it clustered with Gudgenby flavi-like virus (GenBank accession: QTJ63659.1; amino acid identity: 47.3%). Hanko virus, a species which was first described in 2012, was also present in four pools of mosquitoes collected near to the virus’ type locality, which had an average amino acid identity of >99% (Figure 3, Table 2). The full genome of Hanko virus was sequenced from these variants (GenBank accession ON949927–ON949930). One novel member of the genus Jingmenvirus was detected, with two variants provisionally named “Inari jingmenvirus”. This species was not closely related to any species, although it weakly resembled Wuhan aphid virus 1 (GenBank accession: BBV14756.1; amino acid identity: 48.82%), which was derived from aphids from Japan.

Seventeen variants of sequences representing five tentative novel viruses which grouped within *Iflaviridae* were sequenced from 15 pools comprised of *Oc. caspius*, *Oc. communis*, *Oc. diantaeus*, *Oc. hexodontus*, *Oc. intrudens* and *Oc. punctor*/*punctodes* (Figure 4, Table 2). These were named “Enontekio iflavirus”, “Hanko iflavirus 1 and 2”, “Mekrijarvi iflavirus” and “Pedersore iflavirus”. Enontekio iflavirus sequences were most similar to both Culex iflavi-like virus 4 and Yongsan picorna-like virus 1 (GenBank accessions: AXQ04788.1 and AXV43880.1; amino acid identities of 50.2% and 54.28–59.58%, respectively). Hanko iflavirus 1 sequences were most closely related to Perrin Park virus (GenBank accession: QIJ25864.1; amino acid identity: 68.36%) and Armigeres iflavirus (GenBank accession: YP_009448183.1; amino acid identity: 69.56–79.19%) were similar to Yongsan picorna-like virus 1 (GenBank accession: AXV43880.1; amino acid identity: 77.46–81.43%). Mekrijarvi iflavirus resembles most Thrace picorna-like virus 2 (GenBank accession: QRD99887.1; amino acid identity: 89.87%). Lastly, Pedersore iflavirus sequences were most similar to Redbank virus (GenBank accession: QIJ25857.1; amino acid identity: 50.09%), Budalangi iflavi-like virus (GenBank accession: UCW41643.1; amino acid identity: 54.95%) and Fitzroy Crossing iflavirus 1 (GenBank accession: QLJ83497.1; amino acid identity: 49.1%), although these clustered close to Budalangi iflavi-like virus (GenBank accession: UCW41643.1; amino acid identity: 54.95–55.33%).

Forty-one strains of seven viruses belonging to the proposed taxon *Negevirus* were sequenced from 13 mosquito pools. While not yet formally recognised by the ICTV, Negeviruses have been recorded from mosquitoes and sandflies, among other arthropod species. Four of the viruses, “Kustavi negevirus” and “Utsjoki negevirus 1 to 3” were novel; while three, Cordoba virus, Dezidougou virus and Mekrijärvi negevirus (Figure 5, Table 2) have previously been described. Kustavi negevirus is most similar to Dezidougou virus (GenBank accession: AFI24669.1; amino acid identity: 72.41%); Utsjoki negevirus 1 to Ying Kou virus (amino acid identity: 74.11–87.5%) and Mekrijärvi negevirus (amino acid identity: 72.33–78.99%); and Utsjoki negeviruses 2 and 3 are most closely related to Dezidougou virus (protein identities of 62.63–78.81% and 81.93–82.15%). The newly sequenced strains of Cordoba virus and Dezidougou virus shared a high similarity to previously described strains of the same virus species (GenBank accessions: AQM55308.1 and AQM55309.1; amino acid identity: 90.99–95.52%; and QIN93579.1; amino acid identity: 90.12%, respectively). Newly generated Mekrijärvi negevirus sequences were nearly identical to the proposed type of virus species (amino acid identity: 99.37–100%). Full genomes were assembled for Kustavi negevirus, Dezidougou virus and Utsjoki negevirus 1 (GenBank accession ON949944, ON949943 and ON949945–ON949948, respectively).

**Figure 3 viruses-14-01489-f003:**
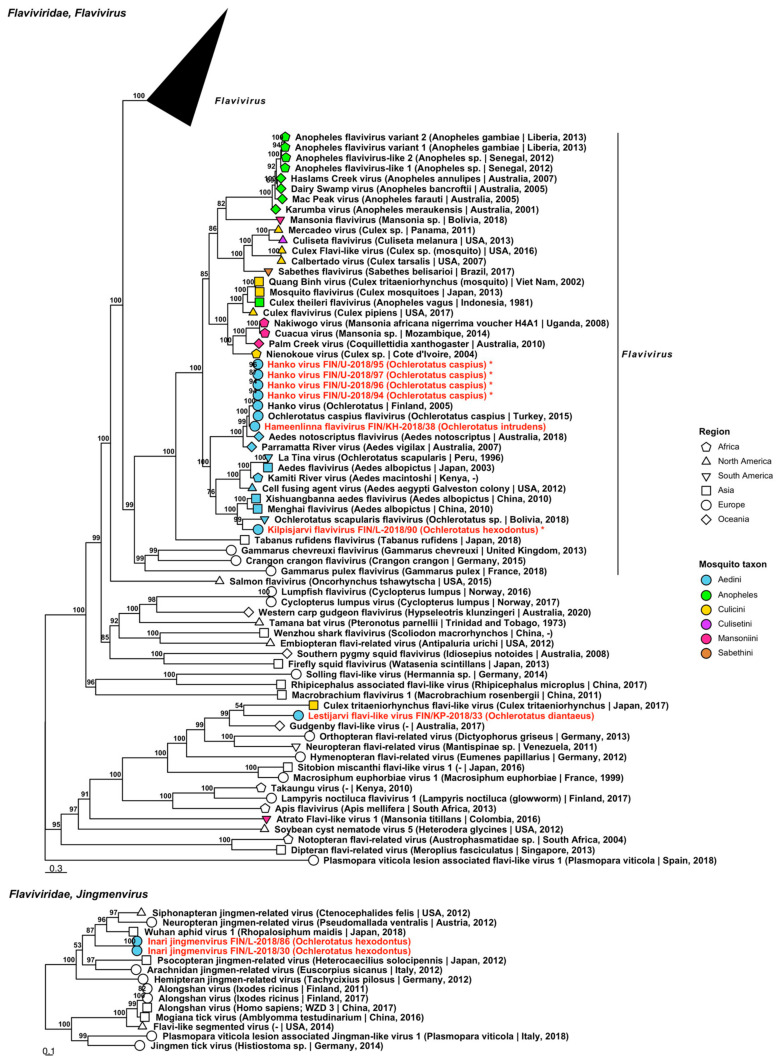
Maximum likelihood trees of *Flaviviridae*. Tentative novel viruses are displayed in red and the mosquito species from which they were derived are in parentheses. Sequences from GenBank are black and display the following information after the virus or species name: “(sampled organism(s)|collection country, collection year)”. Tip colours represent the tribe (Culicinae) or genus (Anophelinae) of mosquito from which viruses were obtained. Tip shape represents the continent or region from which the specimens were collected. Trees were constructed from amino acid sequences of virus polymerases >1000 nt, aligned with MAFFT and computed with IQ-TREE2 using. Asterisks denote that the complete genome was recovered.

**Figure 4 viruses-14-01489-f004:**
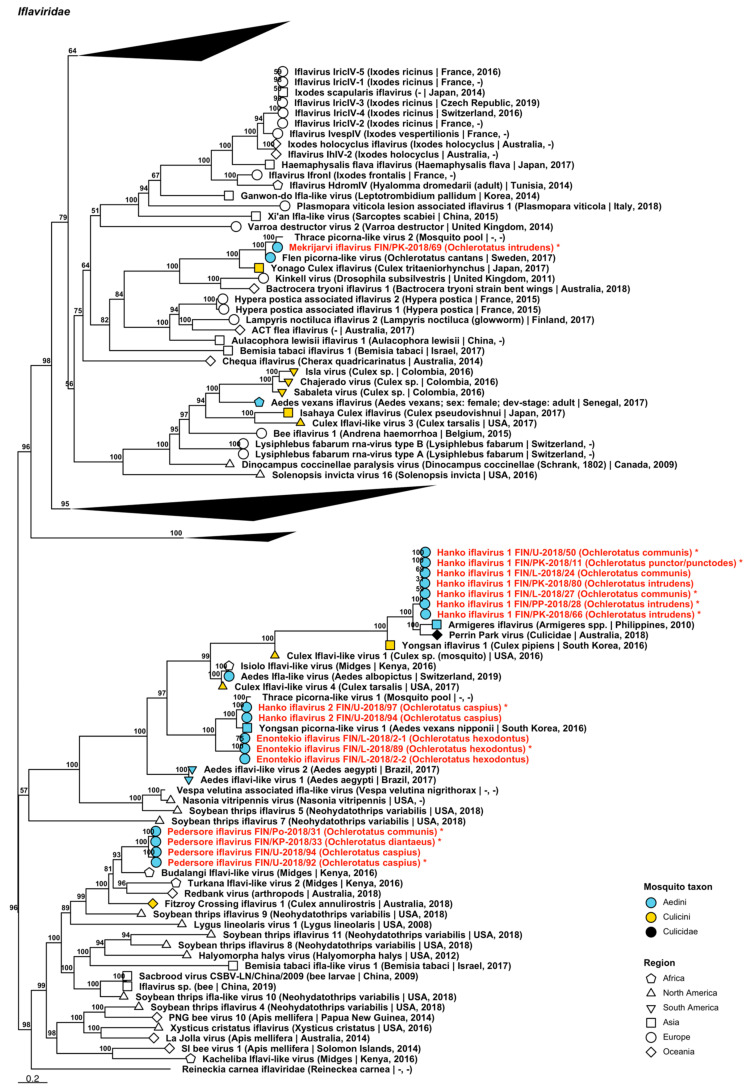
Maximum likelihood tree of *Iflaviridae*. Tentative novel virus species are displayed in red and the mosquito species from which they were derived are in parentheses. Sequences from GenBank are black and display the following information after the virus or species name: “(sampled organism(s)|collection country, collection year)”. Tip colours represent the tribe of mosquito from which viruses were obtained. Tip shape represents the continent or region from which the specimens were collected. Trees were constructed from amino acid sequences of virus polymerases >1000 nt, aligned with MAFFT and computed with IQ-TREE2 using ModelFinder and 1000 bootstraps. Asterisks denote that the complete genome was recovered.

**Figure 5 viruses-14-01489-f005:**
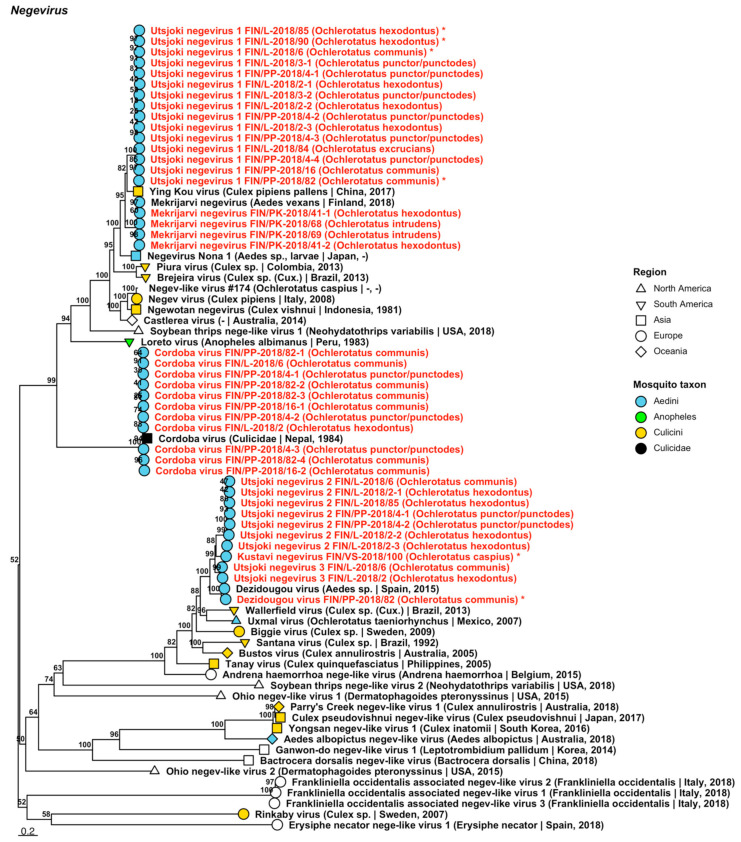
Maximum likelihood tree of *Negevirus*. Tentative novel viruses are displayed in red and the mosquito species from which they were derived are in parentheses. Sequences from GenBank are black and display the following information after the virus or species name: “(sampled organism(s) | collection country, collection year)”. Tip colours represent the tribe (Culicinae) or genus (Anophelinae) of mosquito from which viruses were obtained. Tip shape represents the continent or region from which the specimens were collected. Trees were constructed from amino acid sequences of virus polymerases >1000 nt, aligned with MAFFT and computed with IQ-TREE2 using ModelFinder and 1000 bootstraps. Asterisks denote that the complete genome was recovered.

Six variants of one novel species belonging to *Permutotetraviridae*, a family associated with arthropods, were sequenced from five mosquito pools (Figure 6, Table 2). Named “Inari permutotetravirus”, its amino acid identity was most similar to Smithfield permutotetra-like virus (GenBank accession: QIJ25871.1/QIJ25875.1; amino acid identity: 42.72–66.32%), which were both sequenced from unspecified arthropods collected from Queensland, Australia.

Five variants of two species of *Picornaviridae*, a family of viruses that infect a broad range of vertebrates, were sequenced from two pools of *Oc. caspius* (Figure 6, Table 2). The first species was a previously described but as yet unnamed RNA virus, tentatively named here as “Hanko picorna-like viruses”. The previously described virus was obtained from an anal swab taken from a passerine bird in a Chinese zoo and was nearly identical to the Finnish variant (GenBank accession: QKN89015.1; amino acid identity: 97.15–99.47%). The second species, Jotan virus, shared high amino acid identity with its previously described counterpart from *Culex* mosquitoes in Sweden (GenBank accession: QGA70904.1; amino acid identity: 98.25–98.8%).

One virus sequence grouped with the proposed insect-specific taxon *Quenyavirus*, and was named “Enontekio quenyavirus”, despite being found in specimens collected from northern Lapland and from Uusimaa in the far south of Finland (Figure 6, Table 2). Based on amino acid identity, it is relatively distant from its closest relative, Nete virus (GenBank accession: QIQ61196.1; amino acid identity: 39.71–39.77%) which was sequenced from the moth, *Crocallis elinguaria*, from the UK.

Fifteen variants belonging to the plant-specific *Solemoviridae* were sequenced, which corresponded to one established virus, Evros sobemo-like virus, and four novel species (Figure 7, Table 2). The novel viruses, “Enontekio sobemovirus”, “Hanko sobemovirus”, “Ilomantsi sobemovirus” and “Joensuu sobemovirus”, clustered with other viruses in *Sobemovirus* based on our phylogenetic analysis. Enontekio sobemovirus was closely related to Guadeloupe mosquito virus (GenBank accession: QRW42396.1; amino acid identity: 82.86%) and Kellev virus (GenBank accession: QRW41864.1; amino acid identity: 85.91–86.14%). Based on protein similarity, however, it clustered with Atrato Sobemo-like virus 5 (GenBank accession: QHA33869.1; amino acid identity: 80.8–82.31%). The other novel viruses, Hanko sobemovirus (amino acid identity: 83.46%), Ilomantsi sobemovirus (amino acid identity: 84.69–86.06%) and Joensuu sobemovirus (amino acid identity: 83.7–86.07%), in turn, were most similar with Atrato sobemo-like virus 4 (GenBank accession: QHA33876.1). Six sequences (Table 2) shared a high protein similarity with Evros sobemo-like virus (GenBank accession: QRD99867.1/QRD99868.1; amino acid identity: 97.6–98.86%).

One variant of *Sindbis virus* (*Togaviridae*) was sequenced from a pool of *Oc. communis* collected on 26 June 2015 in Mekrijärvi, Pohjois-Karjala (Figure 8, Table 2). It was closely related to another Finnish mosquito-derived strain (GenBank accession: AFL65801.1; amino acid identity: 99.76%). This new variant is of note as it is the first mosquito species in Finland that has been definitively linked with *Sindbis virus*, which causes human disease outbreaks in the country.

Seven variants of viruses that were closely related to plant-specific viruses in *Virgaviridae* were recovered, belonging to three viruses (Figure 9, Table 2). They did not, however, cluster with established virgavirus genera in the ML tree, and as such were all named virga-like viruses “Enontekio virga-like virus 1 and 2” and “Pedersore virga-like virus”. The closest matches for these three novel viruses were as follows: Enontekio virga-like virus 1 was closest to mosquito-derived Atrato virga-like virus 6 (GenBank accession: QHA33758.1; amino acid identity: 62.86%) from Columbia; Enontekio virga-like virus 2 was distantly similar to the plant pathogen Plasmopara viticola lesion associated virga-like virus 1 (GenBank accession: QHD64722.1; amino acid identity: 34.46%) from Spain; and Pedersore virga-like virus was similar to an unnamed RNA virus which was sequenced from mosquitoes in China (GenBank accession: QTW97796.1; amino acid identity: 63.74–65.16%) as well as Atrato virga-like virus 3 (GenBank accession: QHA33742.1; amino acid identity: 48.74–56.47%), a mosquito-derived virus from Columbia.

**Figure 7 viruses-14-01489-f007:**
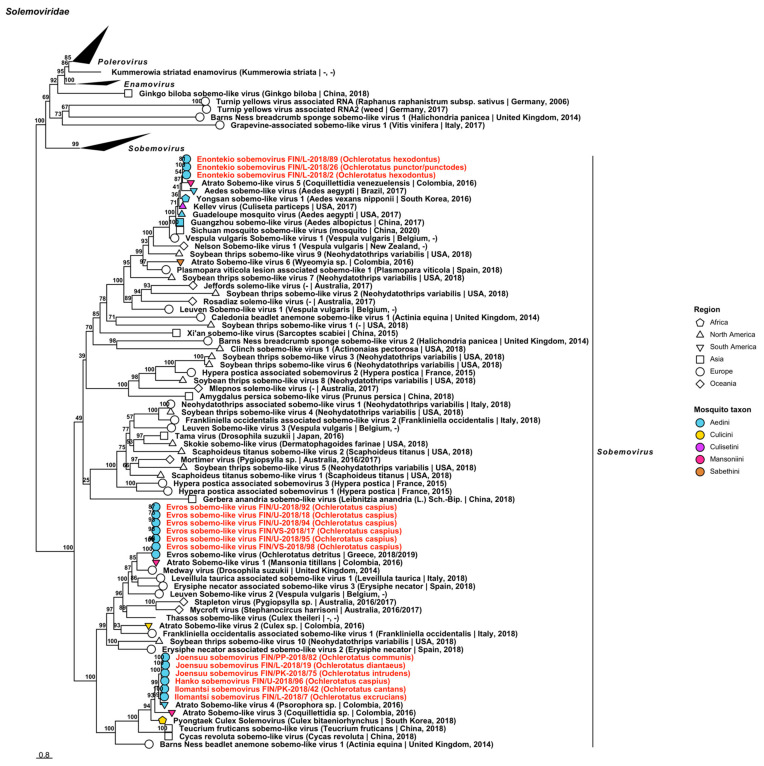
Maximum likelihood tree of *Solemoviridae*. Tentative novel viruses are displayed in red and the mosquito species from which they were derived are in parentheses. Sequences from GenBank are black and display the following information after the virus or species name: “(sampled organism(s)|collection country, collection year)”. Tip colours represent the tribe of mosquito from which viruses were obtained. Tip shape represents the continent or region from which the specimens were collected. Trees were constructed from amino acid sequences of virus polymerases >1000 nt, aligned with MAFFT and computed with IQ-TREE2 using ModelFinder and 1000 bootstraps.

**Figure 8 viruses-14-01489-f008:**
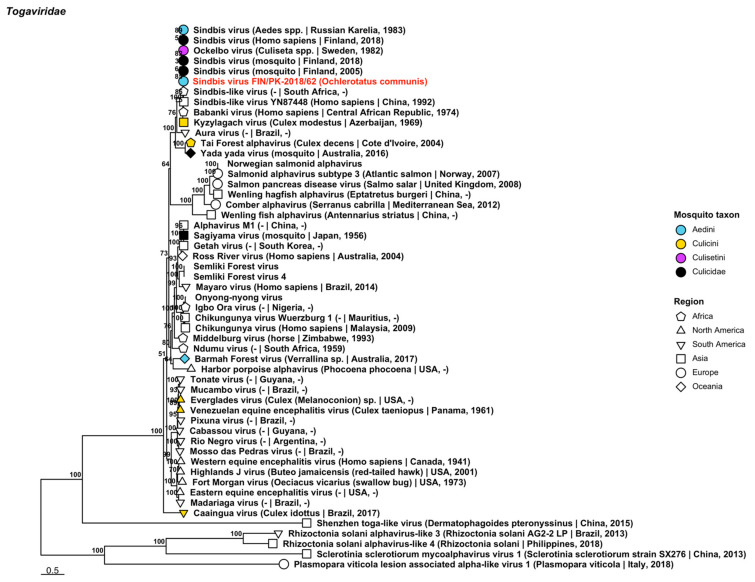
Maximum likelihood tree of *Togaviridae*. The novel strain of *Sindbis virus* is displayed in red and was derived from *Oc. communis*. Sequences from GenBank are black and display the following information after the virus or species name: “(sampled organism(s)|collection country, collection year)”. Tip colours represent the tribe of mosquito from which viruses were obtained. Tip shape represents the continent or region from which the specimens were collected. Trees were constructed from amino acid sequences of virus polymerases >1000 nt, aligned with MAFFT and computed with IQ-TREE2 using ModelFinder and 1000 bootstraps.

#### 3.1.2. Negative-Sense ssRNA Virus Sequences

Negative-sense ssRNA viruses belonging to nine virus families, *Aliusviridae*, *Aspiviridae*, *Chuviridae*, *Phasmaviridae*, *Phenuiviridae*, *Qinviridae*, *Rhabdoviridae*, *Xinmoviridae* and *Yueviridae* were recovered during this study. The −ssRNA viruses are listed below, with all tentative variant names and associated mosquito species in Table 3 and Table 4.

*Aliusviridae* is comprised of two genera, *Ollusvirus* and *Obscuruvirus*, and its member species have previously been from insects. One novel virus belonging to *Obscuruvirus* was sequenced from a pool of *Oc. communis*, which was tentatively named “Lestijarvi obscuruvirus” (Figure 10, Table 3). It was most similar to Atrato chu-like virus 5 (GenBank accession: QHA33675.1; amino acid identity: 41.87%), which was sequenced from *Psorophora ciliata*, an aedine mosquito from Columbia.

**Figure 9 viruses-14-01489-f009:**
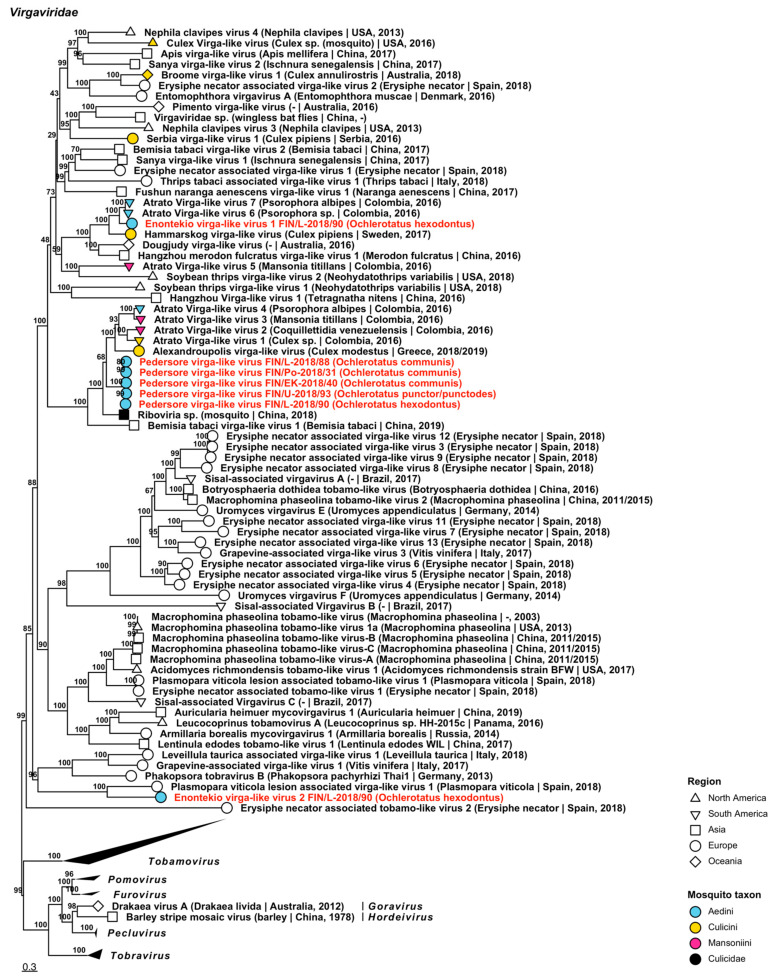
Maximum likelihood tree of *Virgaviridae*. Tentative novel viruses are displayed in red and the mosquito species from which they were derived are in parentheses. Sequences from GenBank are black and display the following information after the virus or species name: “(sampled organism(s)|collection country, collection year)”. Tip colours represent the tribe of mosquito from which viruses were obtained. Tip shape represents the continent or region from which the specimens were collected. Trees were constructed from amino acid sequences of virus polymerases >1000 nt, aligned with MAFFT and computed with IQ-TREE2 using ModelFinder and 1000 bootstraps.

#### 3.1.3. Negative-Sense ssRNA Virus Sequences

Negative-sense ssRNA viruses belonging to nine virus families, *Aliusviridae*, *Aspiviridae*, *Chuviridae*, *Phasmaviridae*, *Phenuiviridae*, *Qinviridae*, *Rhabdoviridae*, *Xinmoviridae* and *Yueviridae* were recovered during this study. The −ssRNA viruses are listed below, with all tentative variant names and associated mosquito species in Table 3 and Table 4.

*Aliusviridae* is comprised of two genera, *Ollusvirus* and *Obscuruvirus*, and its member species have previously been from insects. One novel virus belonging to *Obscuruvirus* was sequenced from a pool of *Oc. communis*, which was tentatively named “Lestijarvi obscuruvirus” (Figure 10, Table 3). It was most similar to Atrato chu-like virus 5 (GenBank accession: QHA33675.1; amino acid identity: 41.87%), which was sequenced from *Psorophora ciliata*, an aedine mosquito from Columbia.

Similarly, one virus grouped with *Aspiviridae*, a plant pathogenic family of viruses, and was tentatively named “Kilpisjarvi aspivirus” (Figure 10, Table 3). Its closest match was Wilkie ophio-like virus 1 (GenBank accession: ASA47457.1; amino acid identity: 50.45%), which was derived from a mosquito from Western Australia.

**Table 3 viruses-14-01489-t003:** Novel −ssRNA viruses sequenced from Finnish mosquitoes, part 1. Where more than one virus was sequenced from a pool, an additional code was appended to the pool number.

Virus Family/ Taxon	Virus Name	Pool/Variant No.	Associated Mosquito Species	GenBank Accession
*Aliusviridae*	Lestijarvi obscuruvirus	FIN/KP-2018/32	*Oc. communis*	ON955144
*Aspiviridae*	Kilpisjarvi aspivirus	FIN/L-2018/90	*Oc. hexodontus*	ON955145
*Chuviridae*	Hattula chuvirus	FIN/L-2018/01-1 FIN/L-2018/01-2 FIN/L-2018/02 FIN/PP-2018/10-1 FIN/PP-2018/10-2 FIN/PP-2018/28-1 FIN/PP-2018/28-2 FIN/KH-2018/29 FIN/KP-2018/32 FIN/KS-2018/35 FIN/EK-2018/40 FIN/PK-2018/74	*Oc. hexodontus* *Oc. hexodontus* *Oc. hexodontus* *Oc. communis* *Oc. communis* *Oc. intrudens* *Oc. intrudens* *Oc. pullatus* *Oc. communis* *Oc. communis* *Oc. communis* *Oc. communis*	ON955150 ON955151 ON955152 ON955154 ON955155 ON955156 ON955157 ON955147 ON955148 ON955149 ON955146 ON955153
*Chuviridae*	Kustavi chuvirus 1	FIN/VS-2018/17	*Oc. caspius*	ON955158
*Chuviridae*	Kustavi chuvirus 2	FIN/VS-2018/17	*Oc. caspius*	ON955159
*Phasmaviridae*	Hameenlinna orthophasmavirus 1	FIN/EK-2018/40 FIN/KH-2018/48 FIN/Pi-2018/51 FIN/Pi-2018/52	*Oc. communis* *Oc. communis* *Oc. communis* *Oc. communis*	ON955160 ON955161 ON955162 ON955163
*Phasmaviridae*	Hameenlinna orthophasmavirus 2	FIN/EK-2018/40 FIN/KH-2018/48	*Oc. communis* *Oc. communis*	ON955164 ON955165
*Phasmaviridae*	Kuusamo orthophasmavirus 1	FIN/PP-2018/83	*Oc. communis*	ON955166
*Phasmaviridae*	Kuusamo orthophasmavirus 2	FIN/PP-2018/83	*Oc. communis*	ON955167
*Phasmaviridae*	Kuusamo orthophasmavirus 3	FIN/PP-2018/83	*Oc. communis*	ON955168
*Phasmaviridae*	Kuusamo orthophasmavirus 4	FIN/EK-2018/40 FIN/PP-2018/83	*Oc. communis* *Oc. communis*	ON955169 ON955170
*Phasmaviridae*	Lestijarvi orthophasmavirus 1	FIN/KP-2018/34	*Oc. intrudens*	ON955171
*Phasmaviridae*	Lestijarvi orthophasmavirus 2	FIN/KP-2018/34	*Oc. intrudens*	ON955172

**Figure 10 viruses-14-01489-f010:**
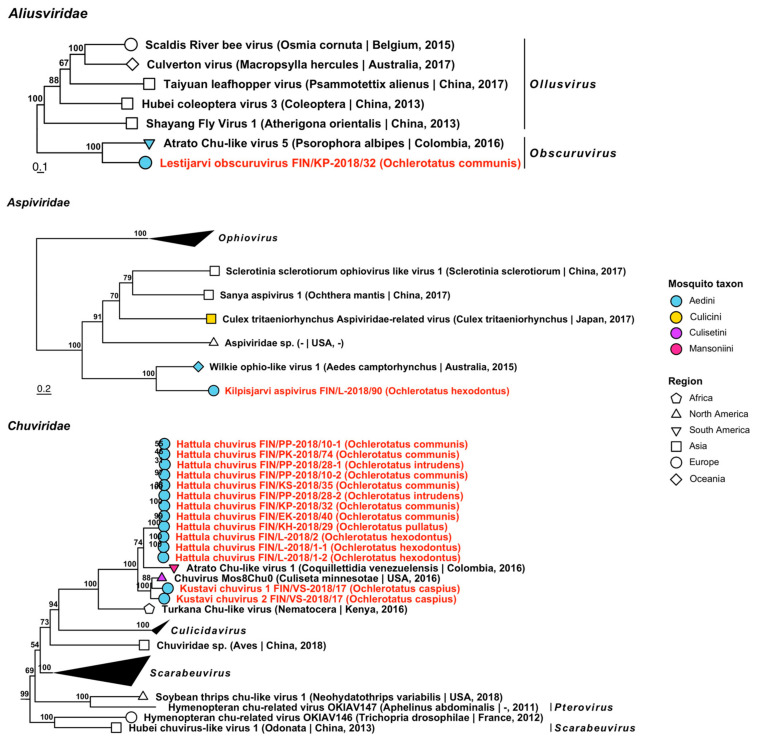
Maximum likelihood trees of *Aliusviridae*, *Aspiviridae* and *Chuviridae*. Tentative novel viruses are displayed in red and the mosquito species from which they were derived are in parentheses. Sequences from GenBank are black and display the following information after the virus or species name: “(sampled organism(s)|collection country, collection year)”. Tip colours represent the tribe of mosquito from which viruses were obtained. Tip shape represents the continent or region from which the specimens were collected. Trees were constructed from amino acid sequences of virus polymerases >1000 nt, aligned with MAFFT and computed with IQ-TREE2 using ModelFinder and 1000 bootstraps.

Thirteen variants from ten mosquito pools belonging to *Chuviridae* (arthropod-associated) were sequenced and grouped into three tentative novel species: “Hattula chuvirus” and “Kustavi chuvirus 1 and 2” (Figure 10, Table 3). By comparing amino acid identities, Hattula chuvirus is most similar to Atrato chu-like virus 1, which was detected in *Coquillettidia venezuelensis* from Colombia (GenBank accession: QHA33913.1, QHA33917.1; amino acid identity: 69.29–70.66%); and to Chuvirus Mos8Chu0 which was detected in *Culiseta minnesotae* from the USA (GenBank accession: API61887.1; amino acid identity: 51.79–63.21%). Kustavi chuviruses 1 and 2 were also most similar to Chuvirus Mos8Chu0 (amino acid identities: 82.24% and 79.7%, respectively); thus, all of the three novel species were most closely related to mosquito-derived chuviruses from the Americas.

Eight novel viruses closely related to species in genus *Orthophasmavirus* from family *Phasmaviridae* were identified from six mosquito pools comprised of *Oc. communis* or *Oc. intrudens* (Figure 11, Table 3). These include the tentatively named “Hameenlinna orthophasmavirus 1 and 2”, “Kuusamo orthophasmavirus 1 to 4” and “Lestijarvi orthophasmavirus 1 and 2”. Hameenlinna orthophasmavirus 1 is most similar to Coredo virus (GenBank accession: QHA33845.1; amino acid identity: 59.25–61.89%), a mosquito-derived virus from Mansoniini mosquitoes in Colombia. Hameenlinna orthophasmavirus 2 had a weak similarity to both *Wuhan mosquito orthophasmavirus 2* (GenBank accession: QTW97787.1; amino acid identity: 36.14%) and Culex phasma-like virus (officially *Culex orthophasmavirus*) (GenBank accession: YP_010085109.1; amino acid identity: 39.08%), mosquito-derived viruses from China and Australia, respectively. Kuusamo orthophasmavirus 1 had a low similarity to its closest matching virus, Coredo virus (amino acid identity: 41%) and Kuusamo orthophasmavirus 2 has a slightly higher similarity to Coredo virus (amino acid identity: 67.6%). Kuusamo orthophasmavirus 3 was most similar to Culex phasma-like virus (GenBank accession: ASA47365.1; amino acid identity: 45.95%) from Australia, and Kuusamo orthophasmavirus 4 to Flen bunya-like virus (GenBank accession: QGA87322.1; amino acid identity: 62.26–71.76%) from *Oc. cantans* that were collected in Sweden. Lastly, Lestijarvi orthophasmavirus 1 was similar to Coredo virus (amino acid identity: 64.1%) and Lestijarvi orthophasmavirus 2 to Culex phasma-like virus (GenBank accession: QHA33850.1; amino acid identity: 40.92%), the latter of which was derived from Columbian *Culex*.

Family *Phenuiviridae* mainly includes arthropod-specific and vector-borne viruses that primarily infect mammals. We detected one sequence representing a novel virus belonging to genus *Phasivirus* and 13 phenui-like viruses (Figure 12, Table 4). These were tentatively named “Hameenlinna phasivirus”, “Enontekio phenui-like virus 1 to 5”, “Hanko phenui-like viruses 1 to 3”, “Ilomantsi phenui-like virus”, “Kalajoki phenui-like viruses 1 and 2” and “Palkane phenui-like virus 1 and 2”. The complete genome of Hameenlinna phasivirus was sequenced (GenBank accession ON955138) and was most similar to Phasi Charoen-like phasivirus (GenBank accession: QEM39210.1, QHT65014.1, QKV44090.1, QKV44092.1, QKV44096.1, QKV44098.1, QKV44099.1, QKV44101.1, QKV44103.1, QKV44109.1, QPF16713.1, YP_009505332.1; amino acid identity: 62.78–87.14%). The closest matching viruses by amino acid identity for the putative novel phenui-like viruses were as follows: Enontekio phenui-like virus 1 had a low similarity to an unnamed bunyavirus that was sequenced from a Chinese mosquito (GenBank accession: QTW97784.1; amino acid identity: 34.56%); Enontekio phenui-like virus 2 to Kristianstad virus, a virus described from Sweden that was sequenced from a *Culex* mosquito [31] (GenBank accession: QGA70932.1; amino acid identity: 34.27%) despite clustering together with Enontekio phenui-like virus 1 and the unnamed bunyavirus sequence (amino acid identity: 35.63%); Enontekio phenui-like virus 3 and Enontekio phenui-like virus 5 to an unnamed RNA virus (GenBank accession: QTW97783.1; amino acid identities: 35.6% and 37.5%); and Enontekio phenui-like virus 4 to Hubei blood fluke virus 2 (GenBank accession: APG79250.1; amino acid identity: 54.2%). Curiously, a phylogenetic analysis suggested that Enontekio phenui-like virus 5 was highly divergent compared to other phenui-related viruses. Hanko phenui-like viruses 1 to 3 were distantly similar to Narangue virus (officially *Narangue mobuvirus*) (GenBank accession: QHA33858.1; protein identities: 51.77%, 39.16% and 65.15%, correspondingly. Ilomantsi phenui-like virus and Kalajoki phenui-like viruses 1 and 2 matched partially with Salari virus (GenBank accession: QGA70945.1; amino acid identities: 60.64%, 39.15–49.47% and 37.5–53.78%). Lastly, Palkane phenui-like virus 1 also matched closely to Salari virus (amino acid identity: 69.63%), while Palkane phenui-like virus 2 (FIN/Pi-2018/52, FIN/Pi-2018/53 and FIN/EK-2018/91) shared the highest amino acid identity with Narangue virus (GenBank accession: QHA33858.1; amino acid identity: 62.44–68.81%).

**Figure 11 viruses-14-01489-f011:**
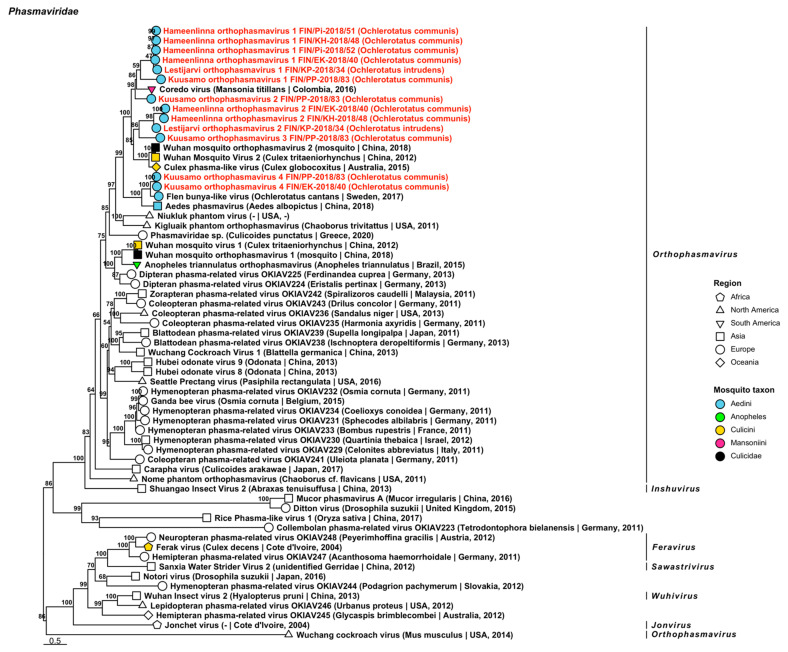
Maximum likelihood tree of *Phasmaviridae*. Tentative novel viruses are displayed in red and the mosquito species from which they were derived are in parentheses. Sequences from GenBank are black and display the following information after the virus or species name: “(sampled organism(s)|collection country, collection year)”. Tip colours represent the tribe (Culicinae) or genus (Anophelinae) of mosquito from which viruses were obtained. Tip shape represents the continent or region from which the specimens were collected. Trees were constructed from amino acid sequences of virus polymerases >1000 nt, aligned with MAFFT and computed with IQ-TREE2 using ModelFinder and 1000 bootstraps.

Three novel variants of *Qinviridae* were detected from pools of *Oc. communis* (Figure 13, Table 4), which were provisionally named “Ilomantsi qinvirus”, “Kalajoki qinvirus” and “Palkane qinvirus”. The first one was distantly similar to Nackenback virus (GenBank accession: QGA70919.1; amino acid identity: 63.3%), which was detected in Sweden from a *Culex* mosquito, while the two others were distantly similar to Wilkie qin-like viruses (GenBank accessions: ASA47357.1 and ASA47455.1; amino acid identities: 54.5–58.2% and 56.61–75.3%).

**Figure 12 viruses-14-01489-f012:**
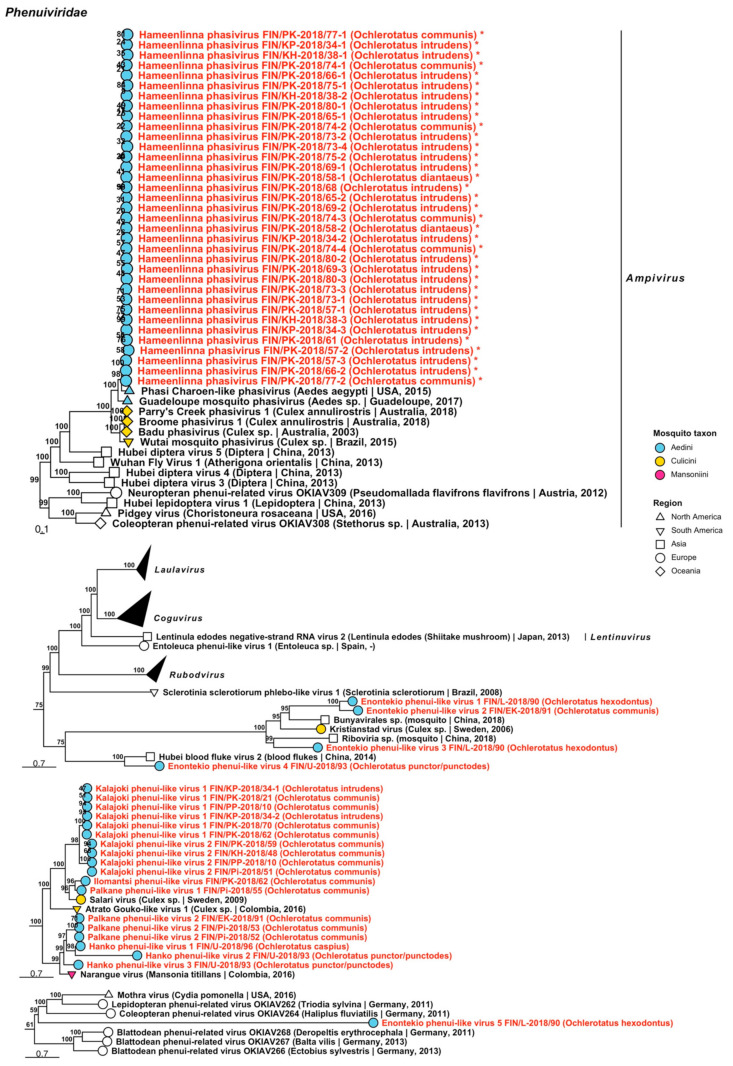
Maximum likelihood subtrees of *Phenuiviridae*. Tentative novel viruses are displayed in red and the mosquito species from which they were derived are in parentheses. Sequences from GenBank are black and display the following information after the virus or species name: “(sampled organism(s)|collection country, collection year)”. Tip colours represent the tribe of mosquito from which viruses were obtained. Tip shape represents the continent or region from which the specimens were collected. Trees were constructed from amino acid sequences of virus polymerases >1000 nt, aligned with MAFFT and computed with IQ-TREE2 using ModelFinder and 1000 bootstraps. Asterisks denote that the complete genome was recovered.

**Table 4 viruses-14-01489-t004:** −ssRNA viruses sequenced from Finnish mosquitoes, part 2. Previously described viruses are shaded grey.

Virus Family/ Taxon	Virus Name	Pool/Variant No.	Associated Mosquito Species	GenBank Accession
*Phenuiviridae*	Hameenlinna phasivirus	FIN/KP-2018/34-1 FIN/KP-2018/34-2 FIN/KP-2018/34-3 FIN/KH-2018/38-1 FIN/KH-2018/38-2 FIN/KH-2018/38-3 FIN/PK-2018/57-1 FIN/PK-2018/57-2 FIN/PK-2018/57-3 FIN/PK-2018/58-1 FIN/PK-2018/58-2 FIN/PK-2018/61 FIN/PK-2018/65-1 FIN/PK-2018/65-2 FIN/PK-2018/66-1 FIN/PK-2018/66-2 FIN/PK-2018/68 FIN/PK-2018/69-1 FIN/PK-2018/69-2 FIN/PK-2018/69-3 FIN/PK-2018/73-1 FIN/PK-2018/73-2 FIN/PK-2018/73-3 FIN/PK-2018/73-4 FIN/PK-2018/74-1 FIN/PK-2018/74-2 FIN/PK-2018/74-3 FIN/PK-2018/74-4 FIN/PK-2018/75-1 FIN/PK-2018/75-2 FIN/PK-2018/77-1 FIN/PK-2018/77-2 FIN/PK-2018/80-1 FIN/PK-2018/80-2 FIN/PK-2018/80-3	*Oc. intrudens* *Oc. intrudens* *Oc. intrudens* *Oc. intrudens* *Oc. intrudens* *Oc. intrudens* *Oc. intrudens* *Oc. intrudens* *Oc. intrudens* *Oc. diantaeus* *Oc. diantaeus* *Oc. intrudens* *Oc. intrudens* *Oc. intrudens* *Oc. intrudens* *Oc. intrudens* *Oc. intrudens* *Oc. intrudens* *Oc. intrudens* *Oc. intrudens* *Oc. intrudens* *Oc. intrudens* *Oc. intrudens* *Oc. intrudens* *Oc. communis* *Oc. communis* *Oc. communis* *Oc. communis* *Oc. intrudens* *Oc. intrudens* *Oc. communis* *Oc. communis* *Oc. intrudens* *Oc. intrudens* *Oc. intrudens*	ON955181 ON955182 ON955183 ON955178 ON955179 ON955180 ON955184 ON955185 ON955186 ON955187 ON955188 ON955189 ON955190 ON955191 ON955192 ON955193 ON955138 ON955194 ON955195 ON955196 ON955197 ON955198 ON955199 ON955200 ON955201 ON955202 ON955203 ON955204 ON955205 ON955206 ON955207 ON955208 ON955209 ON955210 ON955211
*Phenuiviridae*	Enontekio phenui-like virus 1	FIN/L-2018/90	*Oc. hexodontus*	ON955173
*Phenuiviridae*	Enontekio phenui-like virus 2	FIN/EK-2018/91	*Oc. communis*	ON955174
*Phenuiviridae*	Enontekio phenui-like virus 3	FIN/L-2018/90	*Oc. hexodontus*	ON955175
*Phenuiviridae*	Enontekio phenui-like virus 4	FIN/U-2018/93	*Oc. punctor/punctodes*	ON955176
*Phenuiviridae*	Enontekio phenui-like virus 5	FIN/L-2018/90	*Oc. hexodontus*	ON955177
*Phenuiviridae*	Hanko phenui-like virus 1	FIN/U-2018/96	*Oc. caspius*	ON955212
*Phenuiviridae*	Hanko phenui-like virus 2	FIN/U-2018/93	*Oc. punctor/punctodes*	ON955213
*Phenuiviridae*	Hanko phenui-like virus 3	FIN/U-2018/93	*Oc. punctor/punctodes*	ON955214
*Phenuiviridae*	Ilomantsi phenui-like virus	FIN/PK-2018/62	*Oc. communis*	ON955215
*Phenuiviridae*	Kalajoki phenui-like virus 1	FIN/PP-2018/10 FIN/PK-2018/21 FIN/KP-2018/34-1 FIN/KP-2018/34-2 FIN/PK-2018/62 FIN/PK-2018/70	*Oc. communis* *Oc. communis* *Oc. intrudens* *Oc. intrudens* *Oc. communis* *Oc. communis*	ON955221 ON955218 ON955216 ON955217 ON955219 ON955220
*Phenuiviridae*	Kalajoki phenui-like virus 2	FIN/PP-2018/10 FIN/KH-2018/48 FIN/Pi-2018/51 FIN/PK-2018/59	*Oc. communis* *Oc. communis* *Oc. communis* *Oc. communis*	ON955225 ON955222 ON955223 ON955224
*Phenuiviridae*	Palkane phenui-like viruses 1	FIN/Pi-2018/55	*Oc. communis*	ON955226
*Phenuiviridae*	Palkane phenui-like viruses 2	FIN/Pi-2018/52 FIN/Pi-2018/53 FIN/EK-2018/91	*Oc. communis* *Oc. communis* *Oc. communis*	ON955228 ON955229 ON955227
*Qinviridae*	Ilomantsi qinvirus	FIN/PK-2018/62	*Oc. communis*	ON955230
*Qinviridae*	Kalajoki qinvirus	FIN/PP-2018/10 FIN/Pi-2018/54 FIN/PK-2018/60	*Oc. communis* *Oc. communis* *Oc. communis*	ON955233 ON955231 ON955232
*Qinviridae*	Palkane qinvirus	FIN/Pi-2018/54 FIN/PK-2018/60-1 FIN/PK-2018/60-2	*Oc. communis* *Oc. communis* *Oc. communis*	ON955234 ON955235 ON955236
*Rhabdoviridae*	Enontekio merhavirus	FIN/L-2018/90	*Oc. hexodontus*	ON955141
*Rhabdoviridae*	Enontekio ohlsrhavirus	FIN/L-2018/30-1 FIN/L-2018/30-2 FIN/L-2018/30-3 FIN/L-2018/89	*Oc. hexodontus* *Oc. hexodontus* *Oc. hexodontus* *Oc. hexodontus*	ON955237 ON955238 ON955239 ON955240
*Rhabdoviridae*	Enontekio rhabdovirus	FIN/L-2018/03	*Oc. punctor/punctodes*	ON955241
*Rhabdoviridae*	Hattula rhabdovirus	FIN/KH-2018/29 FIN/KS-2018/35-1 FIN/KS-2018/35-2 FIN/PK-2018/59-1 FIN/PK-2018/59-2 FIN/PK-2018/62 FIN/PK-2018/76-1 FIN/PK-2018/76-2 FIN/L-2018/86-1 FIN/L-2018/86-2	*Oc. pullatus* *Oc. communis* *Oc. communis* *Oc. communis* *Oc. communis* *Oc. communis* *Oc. communis* *Oc. communis* *Oc. hexodontus* *Oc. hexodontus*	ON955242 ON955243 ON955244 ON955247 ON955248 ON955142 ON955249 ON955250 ON955245 ON955246
*Rhabdoviridae*	Inari rhabdovirus	FIN/L-2018/84	*Oc. excrucians*	ON955143
*Rhabdoviridae*	Joutseno rhabdovirus 1	FIN/EK-2018/91	*Oc. communis*	ON955251
*Rhabdoviridae*	Joutseno rhabdovirus 2	FIN/EK-2018/91	*Oc. communis*	ON955252
*Rhabdoviridae*	Ohlsdorf virus	FIN/L-2018/07 FIN/L-2018/84	*Oc. excrucians* *Oc. excrucians*	ON955253 ON955254
*Xinmoviridae*	Enontekio anphevirus 1	FIN/L-2018/90	*Oc. hexodontus*	ON955255
*Xinmoviridae*	Enontekio anphevirus 2	FIN/L-2018/90	*Oc. hexodontus*	ON955256
*Xinmoviridae*	Hanko anphevirus	FIN/U-2018/96	*Oc. caspius*	ON955257
*Xinmoviridae*	Joensuu anphevirus	FIN/PK-2018/74 FIN/PP-2018/82 FIN/PP-2018/83-1 FIN/PP-2018/83-2 FIN/U-2018/93-1 FIN/U-2018/93-2	*Oc. communis* *Oc. communis* *Oc. communis* *Oc. communis* *Oc. punctor/punctodes* *Oc. punctor/punctodes*	ON955258 ON955259 ON955260 ON955261 ON955262 ON955263
*Yueviridae*	Enontekio yuevirus	FIN/L-2018/90	*Oc. caspius*	ON955264

Twenty-one variants of *Rhabdoviridae*, viruses which infect vertebrates, invertebrates and plants, were sequenced from 13 mosquito pools and grouped into eight viruses (Figure 14, Table 4). Seven of these were novel tentative rhabdoviruses and one an established species. Of the tentative novel viruses, two fell within established genera, “Enontekio merhavirus” (*Merhavirus*) and “Enontekio ohlsrhavirus” (*Ohlsrhavirus*), while the remaining species, “Enontekio rhabdovirus”, “Hattula rhabdovirus”, “Inari rhabdovirus”, “Joutseno rhabdovirus 1” and “Joutseno rhabdovirus 2” did not. Two variants of Ohlsdorf virus (officially *Ohlsdorf ohlsrhavirus*) were also sequenced, which were nearly identical to the originally described virus from *Oc. cantans* mosquitoes from Germany [32] (GenBank accessions: YP_010086786.1; amino acid identity: 97.87–98.31%). Enontekio merhavirus had a low similarity to Culex tritaeniorhynchus rhabdovirus (officially *Tritaeniorhynchus merhavirus*) (GenBank accession: BBQ05111.1; amino acid identity: 42.06%), while Enontekio ohlsrhavirus had a moderate similarity to both Ohlsdorf virus (GenBank accessions: ATG83565.1, ATG83567.1 and YP_010086786.1; amino acid identity: 55.01–66.93%) and Riverside virus 1 (*Riverside ohlsrhavirus*), described from *Ochlerotatus* sp. mosquitoes from Hungary [33] (GenBank accession: AMJ52368.1; amino acid identity: 75.39%). Enontekio rhabdovirus shared a low amino acid identity with Culex rhabdovirus detected from *Culex* sp. mosquitoes in California, USA [34] (GenBank accession: AXQ04764.1; amino acid identity: 41.06%), Hattula rhabdovirus to Culex rhabdo-like virus (officially *Culex ohlsrhavirus*) (GenBank accessions: ASA47473.1; amino acid identity: 63.04%), Merida virus (officially *Merida merhavirus*) (*Culex pipiens*/*torrentium*, Sweden) (GenBank accessions: QGA70896.1 and YP_009552115.1; amino acid identity: 31.2–36.41%), Ohlsdorf virus (GenBank accession: ATG83563.1, ATG83566.1 and YP_010086786.1; amino acid identity: 38.4–45.43%) and *Perinet vesiculovirus* detected in Madagascar (GenBank accession: YP_009094388.1; amino acid identity: 45.78–46.12%); Inari rhabdovirus to Ohlsdorf virus (GenBank accession: ATG83565.1; amino acid identity: 40.96%); and both Joutseno rhabdovirus 1 and 2 to Primus virus, detected from *Aedes vexans* in Senegal (GenBank accession: QIS62334.1; amino acid identities: 70.55% and 48.48%, respectively). Complete genomes were sequenced for Enontekio merhavirus, Hattula rhabdovirus and Inari rhabdovirus (GenBank accessions ON955141, ON955142 and ON955143, respectively).

**Figure 13 viruses-14-01489-f013:**
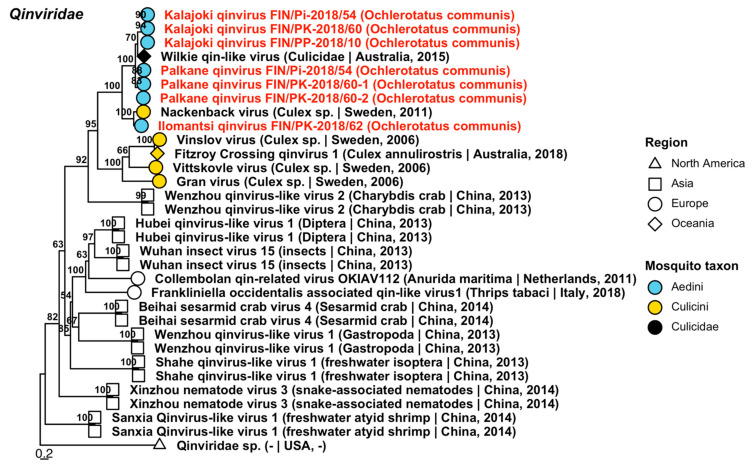
Maximum likelihood tree of *Qinviridae*. Tentative novel viruses are displayed in red and the mosquito species from which they were derived are in parentheses. Sequences from GenBank are black and display the following information after the virus or species name: “(sampled organism(s)|collection country, collection year)”. Tip colours represent the tribe of mosquito from which viruses were obtained. Tip shape represents the continent or region from which the specimens were collected. Trees were constructed from amino acid sequences of virus polymerases >1000 nt, aligned with MAFFT and computed with IQ-TREE2 using ModelFinder and 1000 bootstraps.

**Figure 14 viruses-14-01489-f014:**
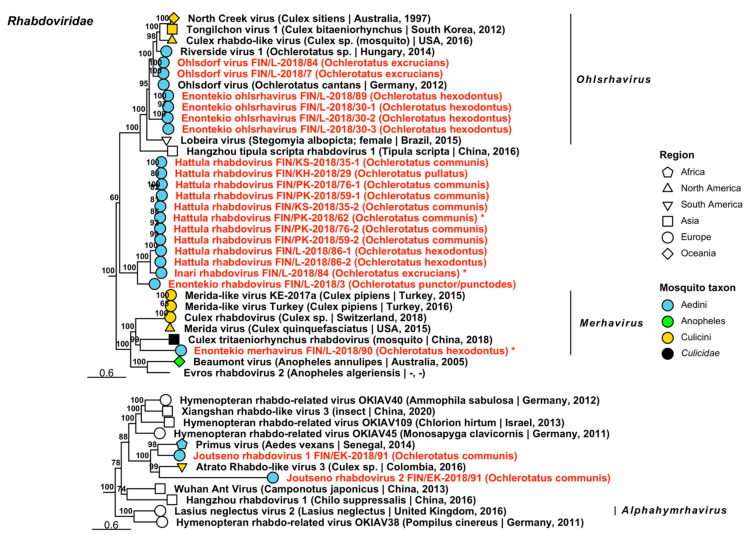
Maximum likelihood subtrees of *Rhabdoviridae*. Tentative novel viruses are displayed in red and the mosquito species from which they were derived are in parentheses. Sequences from GenBank are black and display the following information after the virus or species name: “(sampled organism(s)|collection country, collection year)”. Tip colours represent the tribe (Culicinae) or genus (Anophelinae) of mosquito from which viruses were obtained. Tip shape represents the continent or region from which the specimens were collected. Trees were constructed from amino acid sequences of virus polymerases >1000 nt, aligned with MAFFT and computed with IQ-TREE2 using ModelFinder and 1000 bootstraps. Asterisks denote that the complete genome was recovered.

*Xinmoviridae* includes member species that have been isolated from insects. Nine sequences from four mosquito pools grouped into four novel species, which were tentatively named “Enontekio anphevirus 1 and 2”, “Hanko anphevirus” and “Joensuu anphevirus” (Figure 15, Table 4). The closest sequences available on GenBank for each of these novel species were as follows: Enontekio anphevirus 1 had a medium protein similarity with Culex tritaeniorhynchus anphevirus (GenBank accession: BBQ04822.1; amino acid identity: 53.53%), which was sequenced from Japanese *Culex* mosquitoes; Enontekio anphevirus 2 with Aedes anphevirus (GenBank accession: AWW13453.1; amino acid identity: 60.48%), from a colony of aedine mosquitoes from Thailand; Hanko anphevirus with Serbia mononega-like virus 1 (GenBank accession: QNS17450.1; amino acid identity: 57.88%) from Serbian specimens of *Culex pipiens*; and Joensuu anphevirus with Guadeloupe mosquito mononega-like virus (GenBank accession: QEM39171.1; amino acid identity: 49.73–70.95%) in aedine mosquitoes from Guadeloupe. The variant sequences were detected in pools of *Oc. caspius*, *Oc. communis*, *Oc. hexodontus* and *Oc. punctor*/*punctodes* from across Finland.

**Figure 15 viruses-14-01489-f015:**
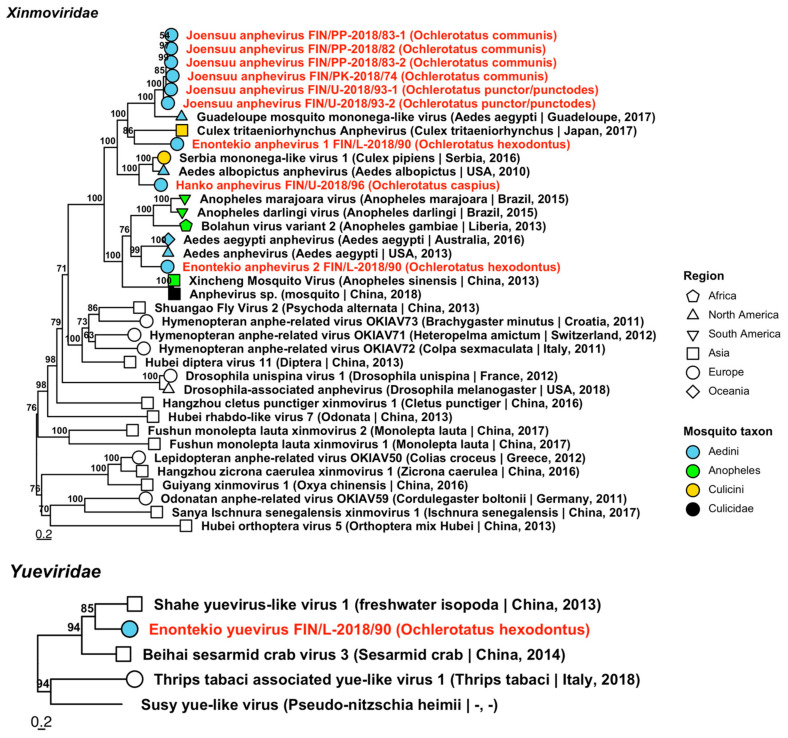
Maximum likelihood trees of *Xinmoviridae* and *Yueviridae*. Tentative novel viruses are displayed in red and the mosquito species from which they were derived are in parentheses. Sequences from GenBank are black and display the following information after the virus or species name: “(sampled organism(s)|collection country, collection year)”. Tip colours represent the tribe (Culicinae) or genus (Anophelinae) of mosquito from which viruses were obtained. Tip shape represents the continent or region from which the specimens were collected. Trees were constructed from amino acid sequences of virus polymerases >1000 nt, aligned with MAFFT and computed with IQ-TREE2 using ModelFinder and 1000 bootstraps.

*Yueviridae* is another recently validated virus family and includes viruses that have been detected from arthropods and marine diatoms. Among our specimens, we isolated one virus sequence from *Oc. hexodontus*, which we named “Enontekio yuevirus” (Figure 15, Table 4). It was very distantly similar to Shahe yuevirus-like virus 1 (officially *Shahe yuyuevirus*) (GenBank accession: YP_009337854.1; amino acid identity: 38.47%), which was sequenced from freshwater isopoda from China.

Finally, while analysing other sequence data that were generated during this study, a fragmentary genome of Inkoo virus (Family *Peribunyavirus*) was identified. The sequences comprised four contigs of 301 to 630 nucleotides which mapped to the M glycoprotein segment, with >99% nucleotide identity to Russian mosquito-derived strain LEIV-15248Iv (GenBank accession; KT288270). While of a different (polymerase) gene than was included in this study, they are still of interest, as Inkoo virus is pathogenic to humans. The sequences were derived from a pool of 60 *Oc. punctor*/*punctodes* (FIN/PK-2018/11), which were collected in late June 2015.

#### 3.1.4. Double-Stranded RNA Virus Sequences

Double-stranded RNA viruses belonging to five established viral families *Chrysoviridae*, *Partitiviridae*, *Sedoreoviridae*, *Spinareoviridae* and *Totiviridae* and one proposed family *Botybirnaviridae* were recovered during the analyses. The dsRNA viruses sequenced in this study are listed, below, with all variant names and associated mosquito species listed in Table 5, Table 6 and Table 7.

*Botybirnavirus* is a recently proposed virus taxon, whose species have been isolated from plants and phytopathogenic fungi. One novel virus was sequenced and tentatively named “Palkane botybirna-like virus”, which had a low resemblance to Bremia lactucae-associated dsRNA virus 1 (GenBank accession: QIP68006.1; amino acid identity: 40.17–44.61%.). Eight variants were found in six pools of *Oc. communis* and one of *Oc. intrudens* (Figure 16, Table 5).

**Figure 16 viruses-14-01489-f016:**
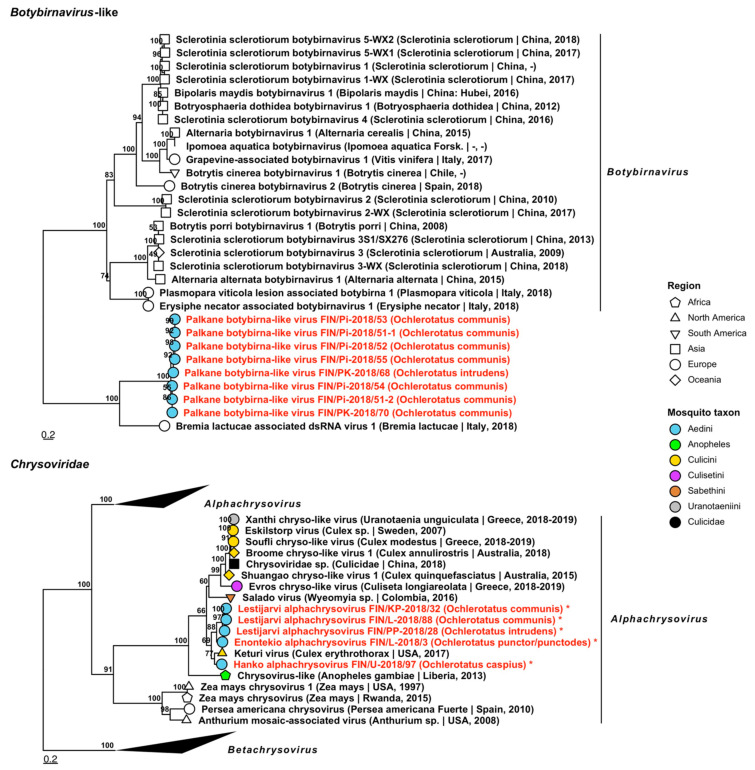
Maximum likelihood trees of *Botybirnavirus* and *Chrysoviridae*. Tentative novel viruses are displayed in red and the mosquito species from which they were derived are in parentheses. Sequences from GenBank are black and display the following information after the virus or species name: “(sampled organism(s)|collection country, collection year)”. Tip colours represent the tribe (Culicinae) or genus (Anophelinae) of mosquito from which viruses were obtained. Tip shape represents the continent or region from which the specimens were collected. Trees were constructed from amino acid sequences of virus polymerases >1000 nt, aligned with MAFFT and computed with IQ-TREE2 using ModelFinder and 1000 bootstraps. Asterisks denote that the complete genome was recovered.

**Table 5 viruses-14-01489-t005:** dsRNA viruses sequenced from Finnish mosquitoes, part 1. Previously described viruses are shaded grey.

Virus Family/ Taxon	Virus Name	Pool/Strain No.	Associated Mosquito Species	GenBank Accession
*Botybirnavirus*	Palkane botybirna-like virus	FIN/Pi-2018/51-1 FIN/Pi-2018/51-2 FIN/Pi-2018/52 FIN/Pi-2018/53 FIN/Pi-2018/54 FIN/Pi-2018/55 FIN/PK-2018/68 FIN/PK-2018/70	*Oc. communis* *Oc. communis* *Oc. communis* *Oc. communis* *Oc. communis* *Oc. communis* *Oc. intrudens* *Oc. communis*	OP019912 OP019913 OP019914 OP019915 OP019916 OP019917 OP019918 OP019919
*Chrysoviridae*	Enontekio alphachrysovirus	FIN/L-2018/03	*Oc. punctor/punctodes*	OP019837–OP019840
*Chrysoviridae*	Hanko alphachrysovirus	FIN/U-2018/97	*Oc. caspius*	OP019841–OP019844
*Chrysoviridae*	Lestijarvi alphachrysovirus	FIN/PP-2018/28 FIN/KP-2018/32 FIN/L-2018/88	*Oc. intrudens* *Oc. communis* *Oc. communis*	OP019911, OP019846–OP019848 OP019910 OP019845

Five variants of three novel *Chrysoviridae* viruses, which mainly infect fungi as well as plants and insects, were sequenced from pools of *Oc. caspius*, *Oc. communis*, *Oc. intrudens* and *Oc. punctor*/*punctodes* (Figure 16, Table 5). All species belonged to *Alphachrysovirus* and were provisionally named “Enontekio alphachrysovirus”, “Hanko alphachrysovirus” and “Lestijarvi alphachrysovirus”. These viruses had a moderate similarity to Keturi virus (GenBank accession: QRW42852.1; amino acid identities: 73.68%, 77.62% and 72.98–74.71%, respectively).

Fifty-five strains grouped into 23 novel species belonging to *Partitiviridae*, *viruses* traditionally associated with fungi, plants and protozoa, but recently associated also with arthropods [35,36,37] (Table 6). Eight of these species were partiti-like viruses and did not fall within an established genus, but the remaining fifteen belonged to three established genera: nine in *Alphapartitivirus* (Figure 17), three in *Betapartitivirus* and three in *Deltapartitivirus* (Figure 18). The novel alphapartitiviruses were named “Enontekio alphapartitivirus 1 to 2”, “Hanko alphapartitivirus 1 to 3”, “Kalajoki alphapartitivirus”, “Kuusamo alphapartitivirus” and “Palkane alphapartitivirus 1 and 2”. Enontekio alphapartitivirus 1 was most similar to Hubei partiti-like virus 27 (GenBank accession: APG78241.1; amino acid identity: 63.67%), while Enontekio alphapartitivirus 2, Hanko alphapartitivirus 1, Kuusamo alphapartitivirus and Palkane alphapartitiviruses 1 and 2 were most similar to Wilkie partiti-like virus 2 (GenBank accessions: ASA47308.1 and YP_009388578.1; amino acid identities: 42.66%, 59.29–61.27%, 59.89–62.01%, 56.58–61.03% and 60.8%, respectively). Hanko alphapartitivirus 2 shared a high amino acid identity with Erysiphe necator-associated partitivirus 5 (GenBank accession: QJW70310.1; amino acid identity: 84.66%) and Hanko alphapartitivirus 3 had a slightly lower amino acid identity to Gaeumannomyces tritici partitivirus 1 (GenBank accession: AZT88602.1; amino acid identity: 71.76%). Lastly, Kalajoki alphapartitivirus sequences had moderate similarity to soybean-leaf-associated partitivirus 1 (GenBank accession: ALM62245.1; amino acid identity: 56.23–61.67%). The novel betapartitiviruses detected included the tentatively named “Enontekio betapartitivirus 1”, “Enontekio betapartitivirus 2” and “Kalajoki betapartitivirus”. The closest matching virus to Enontekio betapartitivirus 1 was Partitivirus-like 5 (GenBank accession: AOR51392.1; amino acid identity: 74.96%), the closest to Enontekio betapartitivirus 2 was Wilkie partiti-like virus 1 (GenBank accession: ASA47307.1; amino acid identity: 59.36%) and the closest to Kalajoki betapartitivirus was Vivastbo virus (GenBank accession: QGA70914.1; amino acid identity: 46.63–46.89%). The novel deltapartitiviruses, which were provisionally named “Ilomantsi deltapartitivirus”, “Inari deltapartitivirus” and “Vaasa deltapartitivirus”, were all moderately similar to Culex pseudovishnui partitivirus based on amino acid identity (GenBank accession: BBQ05103.1; amino acid identities: 67.13–67.33%, 76.75% and 66.4–66.6%, respectively).

**Figure 17 viruses-14-01489-f017:**
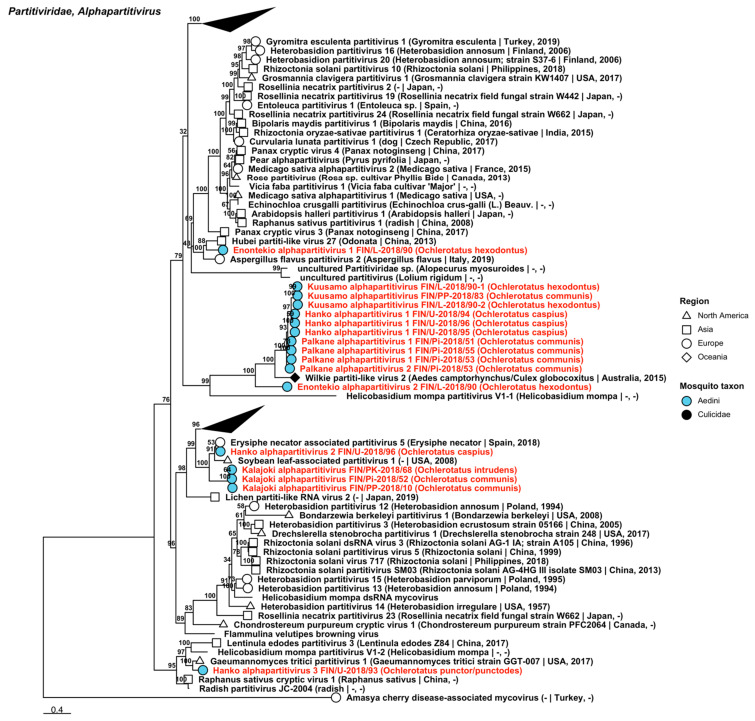
Maximum likelihood tree of *Alphapartitivirus* (*Partitiviridae*). Tentative novel viruses are displayed in red, with the mosquito species from which they were derived in parentheses. Sequences from GenBank are black and display the following information after the virus or species name: “(sampled organism(s)|collection country, collection year)”. Tip colours represent the tribe of mosquito from which viruses were obtained. Tip shape represents the continent or region from which the specimens were collected. Trees were constructed from amino acid sequences of virus polymerases >1000 nt, aligned with MAFFT and computed with IQ-TREE2 using ModelFinder and 1000 bootstraps.

**Table 6 viruses-14-01489-t006:** dsRNA viruses sequenced from Finnish mosquitoes, part 2, *Partitiviridae*.

Virus Family/ Taxon	Virus Name	Pool/Strain No.	Associated Mosquito Species	GenBank Accession
*Partitiviridae*	Enontekio alphapartitivirus 1	FIN/L-2018/90	*Oc. hexodontus*	OP019920
*Partitiviridae*	Enontekio alphapartitivirus 2	FIN/L-2018/90	*Oc. hexodontus*	OP019921
*Partitiviridae*	Hanko alphapartitivirus 1	FIN/U-2018/94 FIN/U-2018/95 FIN/U-2018/96	*Oc. caspius* *Oc. caspius* *Oc. caspius*	OP019929 OP019930 OP019931
*Partitiviridae*	Hanko alphapartitivirus 2	FIN/U-2018/96	*Oc. caspius*	OP019932
*Partitiviridae*	Hanko alphapartitivirus 3	FIN/U-2018/93	*Oc. punctor/punctodes*	OP019933
*Partitiviridae*	Kalajoki alphapartitivirus	FIN/PP-2018/10 FIN/Pi-2018/52 FIN/PK-2018/68	*Oc. communis* *Oc. communis* *Oc. intrudens*	OP019958 OP019956 OP019957
*Partitiviridae*	Kuusamo alphapartitivirus	FIN/PP-2018/83 FIN/L-2018/90-1 FIN/L-2018/90-2	*Oc. communis* *Oc. hexodontus* *Oc. hexodontus*	OP019963 OP019961 OP019962
*Partitiviridae*	Palkane alphapartitivirus 1	FIN/Pi-2018/51 FIN/Pi-2018/53 FIN/Pi-2018/55	*Oc. communis* *Oc. communis* *Oc. communis*	OP019967 OP019968 OP019969
*Partitiviridae*	Palkane alphapartitivirus 2	FIN/Pi-2018/53	*Oc. communis*	OP019970
*Partitiviridae*	Enontekio betapartitivirus 1	FIN/L-2018/90	*Oc. hexodontus*	OP019922
*Partitiviridae*	Enontekio betapartitivirus 2	FIN/L-2018/90	*Oc. hexodontus*	OP019923
*Partitiviridae*	Kalajoki betapartitivirus	FIN/PP-2018/10 FIN/Pi-2018/51	*Oc. communis* *Oc. communis*	OP019960 OP019959
*Partitiviridae*	Ilomantsi deltapartitivirus	FIN/PP-2018/20 FIN/PK-2018/58 FIN/PK-2018/63 FIN/PK-2018/64	*Oc. diantaeus* *Oc. diantaeus* *Oc. diantaeus* *Oc. diantaeus*	OP019944 OP019941 OP019942 OP019943
*Partitiviridae*	Inari deltapartitivirus	FIN/L-2018/85	*Oc. hexodontus*	OP019955
*Partitiviridae*	Vaasa deltapartitivirus	FIN/L-2018/07 FIN/Po-2018/09 FIN/PK-2018/41	*Oc. excrucians* *Oc. excrucians* *Oc. hexodontus*	OP019971 OP019972 OP019973
*Partitiviridae*	Enontekio partiti-like virus	FIN/L-2018/23	*Oc. pullatus*	OP019924
*Partitiviridae*	Hattula partiti-like virus	FIN/L-2018/05 FIN/PP-2018/16 FIN/KH-2018/29 FIN/PK-2018/78 FIN/PP-2018/82 FIN/L-2018/86 FIN/EK-2018/91	*Oc. communis* *Oc. communis* *Oc. pullatus* *Oc. communis* *Oc. communis* *Oc. hexodontus* *Oc. communis*	OP019936 OP019939 OP019935 OP019938 OP019940 OP019937 OP019934
*Partitiviridae*	Hameenlinna partiti-like virus	FIN/PK-2018/42 FIN/KH-2018/48 FIN/U-2018/50 FIN/L-2018/88	*Oc. cantans* *Oc. communis* *Oc. communis* *Oc. communis*	OP019927 OP019925 OP019928 OP019926
*Partitiviridae*	Ilomantsi partiti-like virus 1	FIN/L-2018/02 FIN/L-2018/08 FIN/PP-2018/15 FIN/PK-2018/72 FIN/L-2018/86 FIN/L-2018/89 FIN/L-2018/90	*Oc. hexodontus* *Oc. intrudens* *Oc. punctor/punctodes* *Oc. intrudens* *Oc. hexodontus* *Oc. hexodontus* *Oc. hexodontus*	OP019945 OP019946 OP019951 OP019950 OP019947 OP019948 OP019949
*Partitiviridae*	Ilomantsi partiti-like virus 2	FIN/PK-2018/67 FIN/PK-2018/71 FIN/PK-2018/76	*Oc. punctor/punctodes* *Oc. punctor/punctodes* *Oc. communis*	OP019952 OP019953 OP019954
*Partitiviridae*	Kuusamo partiti-like virus	FIN/PP-2018/82	*Oc. communis*	OP019964
*Partitiviridae*	Lestijarvi partiti-like virus	FIN/KP-2018/34 FIN/PK-2018/41	*Oc. intrudens* *Oc. hexodontus*	OP019965 OP019966
*Partitiviridae*	Vaasa partiti-like virus	FIN/Po-2018/09	*Oc. excrucians*	OP019974

**Figure 18 viruses-14-01489-f018:**
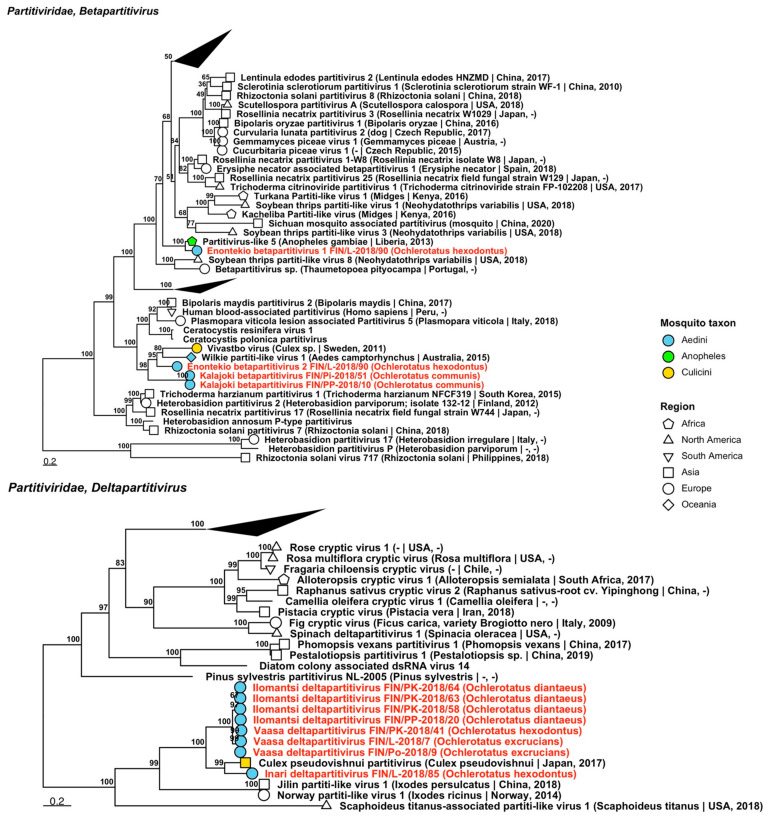
Maximum likelihood trees of *Betapartitivirus* and *Deltapartitivirus* (*Partitiviridae*). Tentative novel viruses are displayed in red and the mosquito species from which they were derived are in parentheses. Sequences from GenBank are black and display the following information after the virus or species name: “(sampled organism(s)|collection country, collection year)”. Tip colours represent the tribe (Culicinae) or genus (Anophelinae) of mosquito from which viruses were obtained. Tip shape represents the continent or region from which the specimens were collected. Trees were constructed from amino acid sequences of virus polymerases >1000 nt, aligned with MAFFT and computed with IQ-TREE2 using ModelFinder and 1000 bootstraps.

Finally, the eight partiti-like viruses included the tentatively named “Enontekio partiti-like virus”, “Hameenlinna partiti-like virus”, “Hattula partiti-like virus”, “Ilomantsi partiti-like virus 1”, “Ilomantsi partiti-like virus 2”, “Kuusamo partiti-like virus”, “Lestijarvi partiti-like virus” and “Vaasa partiti-like virus” (Figure 19, Table 6). Of these viruses, Ilomantsi partiti-like virus 1 shared the highest amino acid identity with Araticum virus detected from *Mansonia wilsoni* mosquitoes from Brazil (GenBank accession: ASV45859.1; amino acid identity: 78.17–78.49%); Ilomantsi partiti-like virus 2 with Atrato partiti-like virus 3 (GenBank accession: QHA33899.1; amino acid identity: 88.14%); Kuusamo partiti-like virus with Culex tritaeniorhynchus partitivirus from Japan [38] (GenBank accession: BBQ05106.1; amino acid identity: 72.03%); and Enontekio partiti-like virus, Hameenlinna partiti-like virus, Hattula partiti-like virus, Lestijarvi partiti-like virus and Vaasa partiti-like virus with different strains of Hubei partiti-like virus 22 (GenBank accessions: APG78217.1, BBQ05104.1 and BBQ05105.1; amino acid identities: 64.58%, 60–62.12%, 76.11–78.57%, 64.04–64.12% and 61.54%).

**Figure 19 viruses-14-01489-f019:**
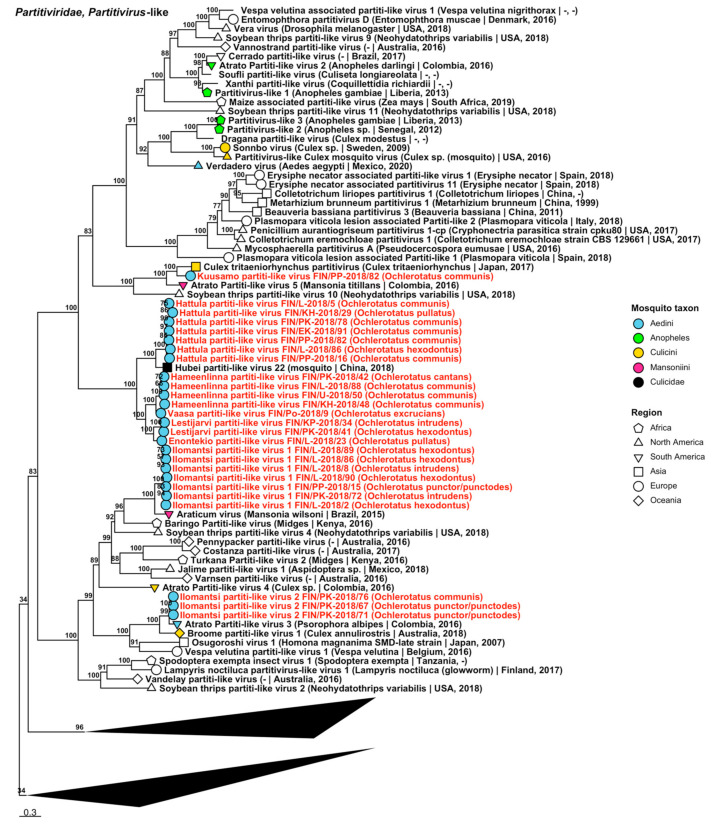
Maximum likelihood tree of partiti-like viruses (*Partitiviridae*). Tentative novel viruses are displayed in red and the mosquito species from which they were derived are in parentheses. Sequences from GenBank are black and display the following information after the virus or species name: “(sampled organism(s)|collection country, collection year)”. Tip colours represent the tribe (Culicinae) or genus (Anophelinae) of mosquito from which viruses were obtained. Tip shape represents the continent or region from which the specimens were collected. Trees were constructed from amino acid sequences of virus polymerases >1000 nt, aligned with MAFFT and computed with IQ-TREE2 using ModelFinder and 1000 bootstraps.

Five novel reoviruses belonging to *Reovirales*, a diverse order of viruses that infect organisms from several phyla, were sequenced (Figure 20, Table 7). Four novel viruses belonging to the family *Sedoreoviridae* were tentatively named “Ilomantsi reovirus 1”, “Ilomantsi reovirus 2”, “Ilomantsi reovirus 3” and “Ilomantsi reovirus 4”, while one novel virus belonging to *Spinareoviridae* was named “Enontekio reovirus”. According to the phylogenetic analyses, none of these five viruses clustered within established genera. Enontekio reovirus was distantly similar to Operophtera brumata reovirus (GenBank accession: YP_392501.1; amino acid identity: 29.59%), while Ilomantsi reoviruses 1–4 were moderately similar to Aedes camptorhynchus reo-like virus (GenBank accession: YP_009389547.1; amino acid identities: 64.96–66.33%, 67.77–70.69%, 74.94% and 64.98%, respectively). However, a phylogenetic analysis suggested that Atrato reo-like virus (GenBank accession: QHA33824.1) was more related to Ilomantsi reoviruses 1–3, while Ilomantsi reovirus 4 clustered near the root of the Ilomantsi reovirus clade.

**Figure 20 viruses-14-01489-f020:**
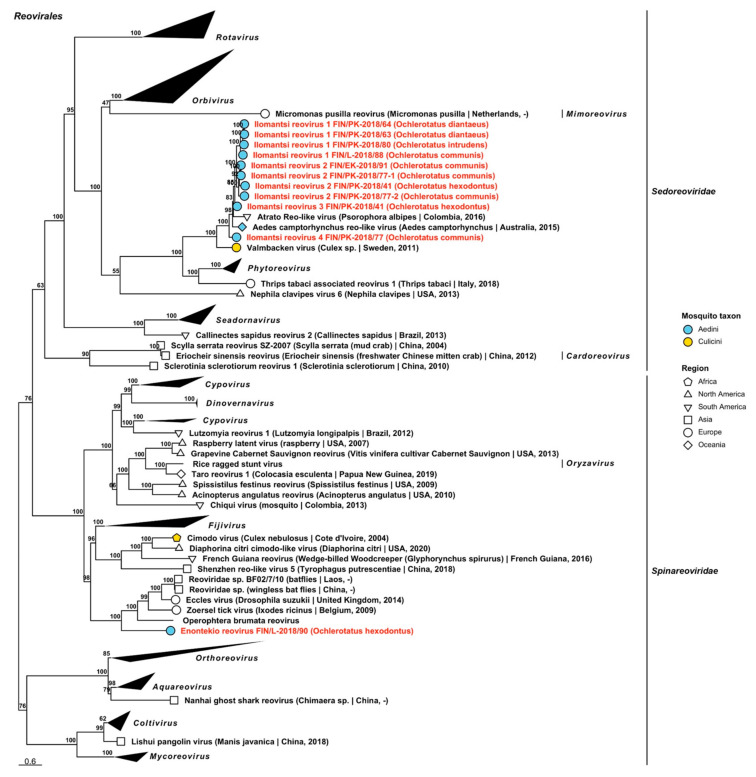
Maximum likelihood tree of *Reovirales*. Tentative novel viruses are displayed in red and the mosquito species from which they were derived are in parentheses. Sequences from GenBank are black and display the following information after the virus or species name: “(sampled organism(s)|collection country, collection year)”. Tip colours represent the tribe of mosquito from which viruses were obtained. Tip shape represents the continent or region from which the specimens were collected. Trees were constructed from amino acid sequences of virus polymerases >1000 nt, aligned with MAFFT and computed with IQ-TREE2 using ModelFinder and 1000 bootstraps.

**Table 7 viruses-14-01489-t007:** dsRNA viruses sequenced from Finnish mosquitoes, part 3. Previously described viruses are shaded grey.

Virus Family/ Taxon	Virus Name	Pool/Strain No.	Associated Mosquito Species	GenBank Accession
*Sedoreoviridae*	Ilomantsi reovirus 1	FIN/PK-2018/63 FIN/PK-2018/64 FIN/PK-2018/80 FIN/L-2018/88	*Oc. diantaeus* *Oc. diantaeus* *Oc. intrudens* *Oc. communis*	OP019977 OP019978 OP019979 OP019976
*Sedoreoviridae*	Ilomantsi reovirus 2	FIN/PK-2018/41 FIN/PK-2018/77-1 FIN/PK-2018/77-2 FIN/EK-2018/91	*Oc. hexodontus* *Oc. communis* *Oc. communis* *Oc. communis*	OP019981 OP019982 OP019983 OP019980
*Sedoreoviridae*	Ilomantsi reovirus 3	FIN/PK-2018/41	*Oc. hexodontus*	OP019984
*Sedoreoviridae*	Ilomantsi reovirus 4	FIN/PK-2018/77	*Oc. communis*	OP019985
*Spinareoviridae*	Enontekio reovirus	FIN/L-2018/90	*Oc. hexodontus*	OP019975
Totiviridae	Hanko toti-like virus 1	FIN/U-2018/92	*Oc. caspius*	OP020048
*Totiviridae*	Hanko toti-like virus 2	FIN/U-2018/94	*Oc. caspius*	OP019860
*Totiviridae*	Hanko toti-like virus 3	FIN/U-2018/93	*Oc. punctor/punctodes*	OP020049
*Totiviridae*	Enontekio toti-like virus 1	FIN/L-2018/90	*Oc. hexodontus*	OP019986
*Totiviridae*	Enontekio toti-like virus 2	FIN/L-2018/90	*Oc. hexodontus*	OP019987
*Totiviridae*	Enontekio toti-like virus 3	FIN/L-2018/90	*Oc. hexodontus*	OP019988
*Totiviridae*	Enontekio toti-like virus 4	FIN/L-2018/90	*Oc. hexodontus*	OP019849
*Totiviridae*	Enontekio totivirus 1	FIN/L-2018/90	*Oc. hexodontus*	OP019989
*Totiviridae*	Enontekio totivirus 2	FIN/L-2018/03 FIN/L-2018/90	*Oc. punctor/punctodes* *Oc. hexodontus*	OP019990 OP019850
*Totiviridae*	Enontekio totivirus 3	FIN/L-2018/23	*Oc. pullatus*	OP019991
*Totiviridae*	Enontekio totivirus 4	FIN/L-2018/23	*Oc. pullatus*	OP019992
*Totiviridae*	Enontekio totivirus 5	FIN/L-2018/05 FIN/PP-2018/16	*Oc. communis* *Oc. communis*	OP019993 OP019851
*Totiviridae*	Enontekio totivirus 6	FIN/L-2018/90	*Oc. hexodontus*	OP019994
*Totiviridae*	Enontekio totivirus 7	FIN/L-2018/90	*Oc. hexodontus*	OP019995
*Totiviridae*	Hameenlinna toti-like virus	FIN/L-2018/08 FIN/KP-2018/34-1 FIN/KP-2018/34-2 FIN/KH-2018/38 FIN/PK-2018/58 FIN/PK-2018/65	*Oc. intrudens* *Oc. intrudens* *Oc. intrudens* *Oc. intrudens* *Oc. diantaeus* *Oc. intrudens*	OP019999 OP019997 OP019998 OP019996 OP020000 OP020001
*Totiviridae*	Hameenlinna totivirus 1	FIN/L-2018/05 FIN/PP-2018/10 FIN/PP-2018/16 FIN/PK-2018/21 FIN/Po-2018/31 FIN/PK-2018/36 FIN/EK-2018/40 FIN/KH-2018/48 FIN/KH-2018/49 FIN/U-2018/50 FIN/Pi-2018/51 FIN/Pi-2018/52-1 FIN/Pi-2018/52-2 FIN/Pi-2018/53 FIN/Pi-2018/54-1 FIN/Pi-2018/54-2 FIN/Pi-2018/55 FIN/EK-2018/56 FIN/PK-2018/59 FIN/PK-2018/60 FIN/PK-2018/68 FIN/PK-2018/69 FIN/PK-2018/70 FIN/PK-2018/74 FIN/PK-2018/76 FIN/PK-2018/78 FIN/PK-2018/79 FIN/PP-2018/82 FIN/PP-2018/83 FIN/L-2018/85-1 FIN/L-2018/85-2 FIN/L-2018/88 FIN/EK-2018/91	*Oc. communis* *Oc. communis* *Oc. communis* *Oc. communis* *Oc. communis* *Oc. communis* *Oc. communis* *Oc. communis* *Oc. communis* *Oc. communis* *Oc. communis* *Oc. communis* *Oc. communis* *Oc. communis* *Oc. communis* *Oc. communis* *Oc. communis* *Oc. communis* *Oc. communis* *Oc. communis* *Oc. intrudens* *Oc. intrudens* *Oc. communis* *Oc. communis* *Oc. communis* *Oc. communis* *Oc. communis* *Oc. communis* *Oc. communis* *Oc. hexodontus* *Oc. hexodontus* *Oc. communis* *Oc. communis*	OP020006 OP020025 OP020026 OP020015 OP019856 OP020016 OP020002 OP020004 OP020005 OP020027 OP019854 OP020009 OP020010 OP020011 OP020012 OP020013 OP020014 OP020003 OP020017 OP020018 OP020019 OP020020 OP020021 OP020022 OP019855 OP020023 OP020024 OP019857 OP019858 OP020007 OP020008 OP019853 OP019852
*Totiviridae*	Hameenlinna totivirus 2	FIN/L-2018/08 FIN/KH-2018/38 FIN/PK-2018/75 FIN/PK-2018/80	*Oc. intrudens* *Oc. intrudens* *Oc. intrudens* *Oc. intrudens*	OP020029 OP020028 OP020030 OP020031
*Totiviridae*	Hameenlinna totivirus 3	FIN/L-2018/08-1 FIN/L-2018/08-2 FIN/KP-2018/34-1 FIN/KP-2018/34-2 FIN/KH-2018/38 FIN/PK-2018/57-1 FIN/PK-2018/57-2 FIN/PK-2018/61-1 FIN/PK-2018/61-2 FIN/PK-2018/65-1 FIN/PK-2018/65-2 FIN/PK-2018/68 FIN/PK-2018/73-1 FIN/PK-2018/73-2 FIN/PK-2018/75 FIN/PK-2018/80-1 FIN/PK-2018/80-2	*Oc. intrudens* *Oc. intrudens* *Oc. intrudens* *Oc. intrudens* *Oc. intrudens* *Oc. intrudens* *Oc. intrudens* *Oc. intrudens* *Oc. intrudens* *Oc. intrudens* *Oc. intrudens* *Oc. intrudens* *Oc. intrudens* *Oc. intrudens* *Oc. intrudens* *Oc. intrudens* *Oc. intrudens*	OP020034 OP020035 OP020032 OP020033 OP019859 OP020036 OP020037 OP020038 OP020039 OP020040 OP020041 OP020042 OP020043 OP020044 OP020045 OP020046 OP020047
*Totiviridae*	Hanko toti-like virus 1	FIN/U-2018/92	*Oc. caspius*	OP020048
*Totiviridae*	Hanko toti-like virus 2	FIN/U-2018/94	*Oc. caspius*	OP019860
*Totiviridae*	Hanko toti-like virus 3	FIN/U-2018/93	*Oc. punctor/punctodes*	OP020049
*Totiviridae*	Hanko totivirus 1	FIN/U-2018/92	*Oc. caspius*	OP020050
*Totiviridae*	Hanko totivirus 2	FIN/U-2018/92	*Oc. caspius*	OP020052
*Totiviridae*	Hanko totivirus 3	FIN/U-2018/18 FIN/U-2018/44 FIN/U-2018/92 FIN/U-2018/94 FIN/U-2018/95 FIN/U-2018/96 FIN/U-2018/97 FIN/VS-2018/99 FIN/VS-2018/100	*Oc. caspius* *Oc. caspius* *Oc. caspius* *Oc. caspius* *Oc. caspius* *Oc. caspius* *Oc. caspius* *Oc. caspius* *Oc. caspius*	OP020053 OP019861 OP019902 OP019909 OP019903 OP019904 OP019862 OP019905 OP020054
*Totiviridae*	Hanko totivirus 4	FIN/U-2018/18 FIN/U-2018/94 FIN/VS-2018/100	*Oc. caspius* *Oc. caspius* *Oc. caspius*	OP020055 OP020056 OP020057
*Totiviridae*	Hanko totivirus 5	FIN/U-2018/18 FIN/U-2018/44 FIN/U-2018/92-1 FIN/U-2018/92-2 FIN/U-2018/92-3 FIN/U-2018/94 FIN/U-2018/95 FIN/U-2018/96 FIN/U-2018/97 FIN/VS-2018/99 FIN/VS-2018/100	*Oc. caspius* *Oc. caspius* *Oc. caspius* *Oc. caspius* *Oc. caspius* *Oc. caspius* *Oc. caspius* *Oc. caspius* *Oc. caspius* *Oc. caspius* *Oc. caspius*	OP020058 OP020059 OP020060 OP020061 OP020062 OP020063 OP019906 OP020064 OP019863 OP020066 OP020065
*Totiviridae*	Hanko totivirus 6	FIN/U-2018/45	*Oc. punctor/punctodes*	OP020067
*Totiviridae*	Hanko totivirus 7	FIN/U-2018/45	*Oc. punctor/punctodes*	OP020068
*Totiviridae*	Hanko totivirus 8	FIN/U-2018/94 FIN/U-2018/95	*Oc. caspius* *Oc. caspius*	OP019864 OP019865
*Totiviridae*	Hanko totivirus 9	FIN/U-2018/44 FIN/U-2018/94 FIN/U-2018/95 FIN/U-2018/96	*Oc. caspius* *Oc. caspius* *Oc. caspius* *Oc. caspius*	OP020069 OP019866 OP019867 OP019900
*Totiviridae*	Hanko totivirus 10	FIN/U-2018/94	*Oc. caspius*	OP020051
*Totiviridae*	Hattula totivirus 1	FIN/L-2018/06 FIN/PP-2018/10 FIN/KH-2018/29 FIN/PK-2018/61 FIN/PK-2018/62 FIN/PK-2018/69 FIN/PK-2018/78 FIN/L-2018/88	*Oc. communis* *Oc. communis* *Oc. pullatus* *Oc. intrudens* *Oc. communis* *Oc. intrudens* *Oc. communis* *Oc. communis*	OP020071 OP019871 OP020070 OP019868 OP019869 OP019870 OP020072 OP019901
*Totiviridae*	Hattula totivirus 2	FIN/PP-2018/10 FIN/KH-2018/29-1 FIN/KH-2018/29-2 FIN/Po-2018/31 FIN/EK-2018/40-1 FIN/EK-2018/40-2 FIN/Pi-2018/52 FIN/Pi-2018/53 FIN/EK-2018/56 FIN/PK-2018/57 FIN/PK-2018/60 FIN/PK-2018/74 FIN/EK-2018/91	*Oc. communis* *Oc. pullatus* *Oc. pullatus* *Oc. communis* *Oc. communis* *Oc. communis* *Oc. communis* *Oc. communis* *Oc. communis* *Oc. intrudens* *Oc. communis* *Oc. communis* *Oc. communis*	OP019876 OP020075 OP020076 OP020080 OP020073 OP020074 OP020077 OP019874 OP019872 OP020078 OP019875 OP020079 OP019873
*Totiviridae*	Hattula totivirus 3	FIN/L-2018/03 FIN/L-2018/23 FIN/L-2018/26 FIN/U-2018/39 FIN/U-2018/45 FIN/KH-2018/47 FIN/PK-2018/60 FIN/PK-2018/62 FIN/PK-2018/66 FIN/L-2018/85	*Oc. punctor/punctodes* *Oc. pullatus* *Oc. punctor/punctodes* *Oc. punctor/punctodes* *Oc. punctor/punctodes* *Oc. punctor/punctodes* *Oc. communis* *Oc. communis* *Oc. intrudens* *Oc. hexodontus*	OP020083 OP020081 OP020082 OP019881 OP019882 OP019877 OP019879 OP019880 OP020084 OP019878
*Totiviridae*	Ilomantsi toti-like virus 1	FIN/PK-2018/65	*Oc. intrudens*	OP020085
*Totiviridae*	Ilomantsi toti-like virus 2	FIN/L-2018/07 FIN/PK-2018/41 FIN/PK-2018/42 FIN/PK-2018/69 FIN/PK-2018/76 FIN/L-2018/84	*Oc. excrucians* *Oc. hexodontus* *Oc. cantans* *Oc. intrudens* *Oc. communis* *Oc. excrucians*	OP020086 OP019883 OP020088 OP020089 OP020090 OP020087
*Totiviridae*	Ilomantsi toti-like virus 3	FIN/PK-2018/58	*Oc. diantaeus*	OP020091
*Totiviridae*	Ilomantsi totivirus 1	FIN/PK-2018/58 FIN/PP-2018/82	*Oc. diantaeus* *Oc. communis*	OP020092 OP019884
*Totiviridae*	Ilomantsi totivirus 2	FIN/PK-2018/42-1 FIN/PK-2018/42-2 FIN/PK-2018/76	*Oc. cantans* *Oc. cantans* *Oc. communis*	OP020093 OP020094 OP020095
*Totiviridae*	Ilomantsi totivirus 3	FIN/PK-2018/42	*Oc. cantans*	OP020096
*Totiviridae*	Inari toti-like virus	FIN/L-2018/84 FIN/U-2018/93	*Oc. excrucians* *Oc. punctor/punctodes*	OP020097 OP020098
*Totiviridae*	Inari totivirus 1	FIN/L-2018/07-1 FIN/L-2018/07-2 FIN/L-2018/84-1 FIN/L-2018/84-2	*Oc. excrucians* *Oc. excrucians* *Oc. excrucians* *Oc. excrucians*	OP020099 OP020100 OP019885 OP020101
*Totiviridae*	Inari totivirus 2	FIN/L-2018/19 FIN/L-2018/85 FIN/L-2018/88	*Oc. diantaeus* *Oc. hexodontus* *Oc. communis*	OP019886 OP019887 OP019888
*Totiviridae*	Joutseno totivirus	FIN/EK-2018/40	*Oc. communis*	OP019889
*Totiviridae*	Karstula totivirus	FIN/KS-2018/35	*Oc. communis*	OP020102
*Totiviridae*	Kustavi toti-like virus	FIN/VS-2018/17 FIN/U-2018/44 FIN/U-2018/92-1 FIN/U-2018/92-2 FIN/U-2018/94-1 FIN/U-2018/94-2 FIN/U-2018/95 FIN/U-2018/96 FIN/U-2018/97-1 FIN/U-2018/97-2 FIN/VS-2018/99 FIN/VS-2018/100	*Oc. caspius* *Oc. caspius* *Oc. caspius* *Oc. caspius* *Oc. caspius* *Oc. caspius* *Oc. caspius* *Oc. caspius* *Oc. caspius* *Oc. caspius* *Oc. caspius* *Oc. caspius*	OP020111 OP020103 OP020104 OP020105 OP020106 OP019890 OP019891 OP020107 OP020108 OP020109 OP020112 OP020110
*Totiviridae*	Kuusamo toti-like virus	FIN/PP-2018/83	*Oc. communis*	OP020113
*Totiviridae*	Kuusamo totivirus 1	FIN/PP-2018/15	*Oc. punctor/punctodes*	OP020114
*Totiviridae*	Kuusamo totivirus 2	FIN/PP-2018/15	*Oc. punctor/punctodes*	OP020115
*Totiviridae*	Lestijarvi totivirus	FIN/L-2018/19 FIN/KP-2018/33 FIN/PK-2018/58 FIN/PK-2018/63 FIN/PK-2018/64 FIN/PP-2018/82	*Oc. diantaeus* *Oc. diantaeus* *Oc. diantaeus* *Oc. diantaeus* *Oc. diantaeus* *Oc. communis*	OP019892 OP020116 OP019893 OP020117 OP020118 OP019894
*Totiviridae*	Palkane toti-like virus	FIN/Pi-2018/52 FIN/EK-2018/56 FIN/PK-2018/60 FIN/PK-2018/78 FIN/L-2018/85 FIN/L-2018/90	*Oc. communis* *Oc. communis* *Oc. communis* *Oc. communis* *Oc. hexodontus* *Oc. hexodontus*	OP020121 OP020119 OP019896 OP020122 OP020120 OP019895
*Totiviridae*	Palkane totivirus	FIN/EK-2018/40 FIN/Pi-2018/54-1 FIN/Pi-2018/54-2 FIN/Pi-2018/55 FIN/EK-2018/91	*Oc. communis* *Oc. communis* *Oc. communis* *Oc. communis* *Oc. communis*	OP019907 OP020123 OP020124 OP020125 OP019908
*Totiviridae*	Utsjoki toti-like virus	FIN/L-2018/88	*Oc. communis*	OP019897
*Totiviridae*	Vaasa toti-like virus	FIN/L-2018/07 FIN/Po-2018/09 FIN/PK-2018/69 FIN/L-2018/84	*Oc. excrucians* *Oc. excrucians* *Oc. intrudens* *Oc. excrucians*	OP020126 OP019899 OP020127 OP019898
*Totiviridae*	Vaasa totivirus	FIN/Po-2018/09 FIN/PK-2018/41	*Oc. excrucians* *Oc. hexodontus*	OP020129 OP020128

The most viral sequences in this study grouped within *Totiviridae*, which includes viruses of fungi and protozoans, among others. From 205 sequences, 52 viruses were identified, of which 50 were novel and two were strains of previously described, albeit unnamed, viruses (Figure 21, Figure 22, Figure 23, Figure 24 and Figure 25, Table 7). Virus strains were found in all nine mosquito species and from across the country. The novel viruses included 33 provisionally named viruses which clustered with member species of *Totivirus*. These included “Enontekio totivirus 1 to 7”, “Hameenlinna totivirus 1 to 3”, “Hanko totivirus 1 to 10”, “Hattula totivirus 1 to 3”, “Ilomantsi totivirus 1 to 3”, “Inari totivirus 1 and 2”, “Joutseno totivirus”, “Karstula totivirus”, “Kuusamo totivirus 1 and 2”, “Lestijarvi totivirus”, “Palkane totivirus” and “Vaasa totivirus”. Protein blast results suggested that the closest matching virus by relatively low amino acid identity values for Enontekio totivirus 1 was Wuhan insect virus 27 (GenBank accession: YP_009342434.1; amino acid identity: 55.87%). Similarly, Enontekio totivirus 2, Hanko totiviruses 8 and 9, Hattula totivirus 2, Ilomantsi totivirus 3, Joutseno totivirus and Vaasa totivirus had a low to moderate amino acid identity to an unnamed dsRNA from an environmental sample (GenBank accession: AJT39583.1; amino acid identities: 61–61.1%, 61.4–61.49%, 46.27–47.07%, 63.78–69.55%, 51.72%, 61.18% and 50–65.23%). Hameenlinna totivirus 3 sequences had a low amino acid identity with multiple previously established viruses including the aforementioned unnamed virus (amino acid identity: 36.36–54.53%), Aedes aegypti toti-like virus (GenBank accession: QEM39133.1; amino acid identity: 39.66–47.3%), Emileo virus (GenBank accession: QRW41692.1; amino acid identity: 44.42%) and Hubei toti-like virus 10 (GenBank accession: YP_009336493.1, amino acid identity: 53.33%). Enontekio totivirus 3 was also similar to Hubei toti-like virus 10 (amino acid identity: 56.29%). 

**Figure 21 viruses-14-01489-f021:**
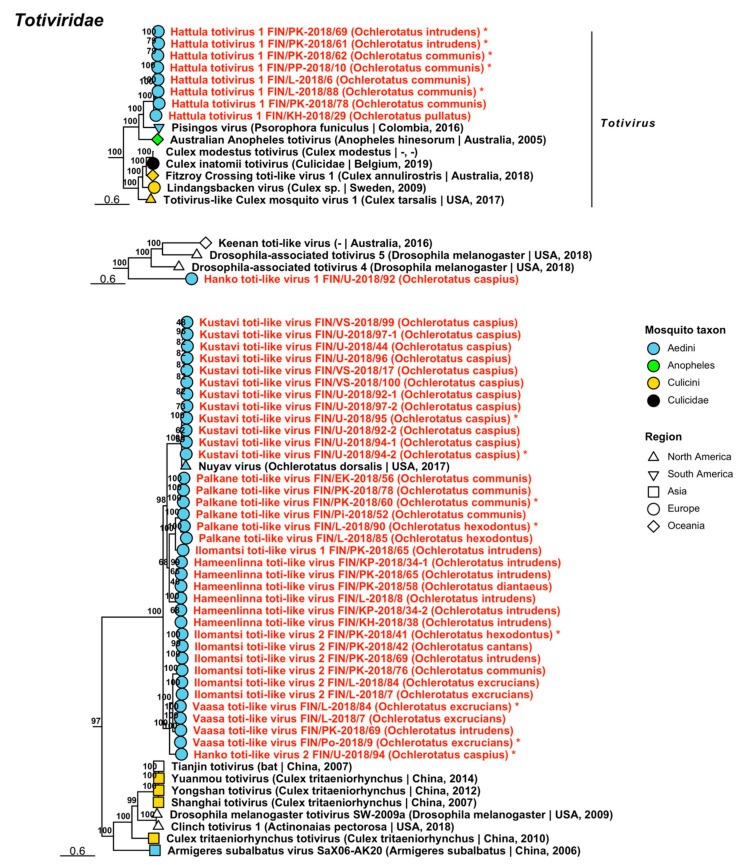
Maximum likelihood subtrees of *Totiviridae*. Tentative novel viruses are displayed in red and the mosquito species from which they were derived are in parentheses. Sequences from GenBank are black and display the following information after the virus or species name: “(sampled organism(s)|collection country, collection year)”. Tip colours represent the tribe (Culicinae) or genus (Anophelinae) of mosquito from which viruses were obtained. Tip shape represents the continent or region from which the specimens were collected. Trees were constructed from amino acid sequences of virus polymerases >1000 nt, aligned with MAFFT and computed with IQ-TREE2 using ModelFinder and 1000 bootstraps. Asterisks denote that the complete genome was recovered.

**Figure 22 viruses-14-01489-f022:**
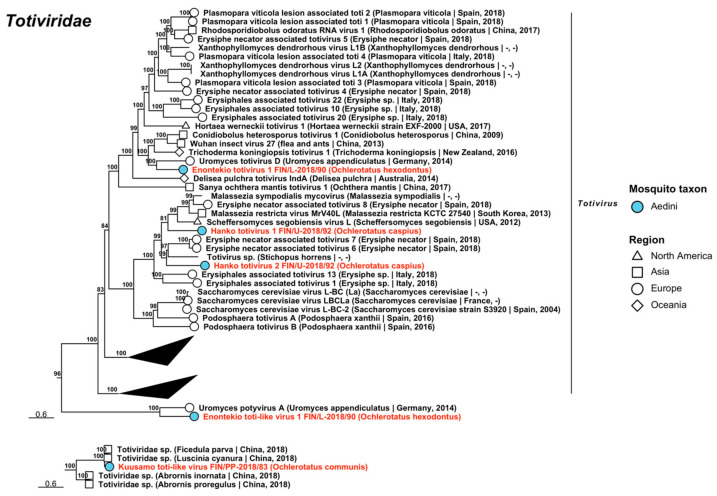
Maximum likelihood subtrees of *Totiviridae*. Tentative novel viruses are displayed in red and the mosquito species from which they were derived are in parentheses. Sequences from GenBank are black and display the following information after the virus or species name: “(sampled organism(s)|collection country, collection year)”. Tip colours represent the tribe of mosquito from which viruses were obtained. Tip shape represents the continent or region from which the specimens were collected. Trees were constructed from amino acid sequences of virus polymerases >1000 nt, aligned with MAFFT and computed with IQ-TREE2 using ModelFinder and 1000 bootstraps.

Enontekio totiviruses 4 and 5 as well as Hattula totivirus 3 were most similar to Murri virus (GenBank accession: QHA33714.1; amino acid identities: 57.2%, 57.8% and 57.48–71.43%, correspondingly). Hattula totivirus 3 also shared amino acid identity with Atrato virus (GenBank accession: QHA33710.1; amino acid identity: 57.46–62.73%), although based on our phylogenetic tree, the virus in general was more related to Murri virus. Enontekio totiviruses 6 and 7 had the most similarity with Beihai razor shell virus 4 (GenBank accession: YP_009333409.1; amino acid identities: 41.12% and 46.59%). Thirteen viruses shared a moderate amino acid identity with an unnamed uncultured virus (GenBank accession: AGW51771.1). These were Hameenlinna totiviruses 1 and 2 (amino acid identities: 48.73–65.96% and 62.42–70.73%), Hanko totiviruses 5, 7 and 10 (amino acid identities: 66.05–75.56%, 70.14% and 49.36%), Ilomantsi totiviruses 1 and 2 (amino acid identities: 54.61–64.58% and 59.68–72.16%), Inari totivirus 1 (amino acid identity: 48.98–57.46%), Karstula totivirus (amino acid identity: 41.18%), Kuusamo totiviruses 1 and 2 (amino acid identities: 73.75% and 45.61%), Lestijarvi totivirus (amino acid identity: 39.84–42.51%) and Palkane totivirus (amino acid identity: 60.16–61.96%). Despite sharing the highest amino acid identity with the uncultured virus, Karstula totivirus, Hanko totivirus 10 and Kuusamo totivirus 2 clustered with Fushun totivirus 4 and Sanya totivirus 7 (GenBank accessions: UHM27684.1 and UHM27502.1; these viruses did not appear in the BLASTx results). Hattula totivirus 1 was similar to Pisingos virus (GenBank accession: QHA33716.1; amino acid identity: 68.82–71.3%), Hanko totivirus 2 to Erysiphe necator associated totivirus 7 (GenBank accession: QJW70337.1; amino acid identity: 51.07%) and Hanko totivirus 6 to Aedes alboannulatus toti-like virus 1 (GenBank accession: YP_009388609.1; amino acid identity: 51.62%). Other detected totiviruses that shared similar amino acid identities to Aedes alboannulatus toti-like virus 1 were strains of the aforementioned Hameenlinna totivirus 1 (amino acid identity: 62.6%) and Inari totivirus 1 (amino acid identity: 62.2%). Several strains of our novel totiviruses were moderately similar to Aedes camptorhynchus toti-like virus 1 (GenBank accession: YP_009388611.1), Inari totivirus 2 (amino acid identity: 52.88–53.19%), Hameenlinna totivirus 1 (amino acid identity: 60.98%) and Lestijarvi totivirus (amino acid identity: 53.3%). Lastly, Hanko totivirus 1 shared a low amino acid identity with Malassezia sympodialis mycovirus (GenBank accession: QNJ34610.1; amino acid identity: 35.14%).

**Figure 23 viruses-14-01489-f023:**
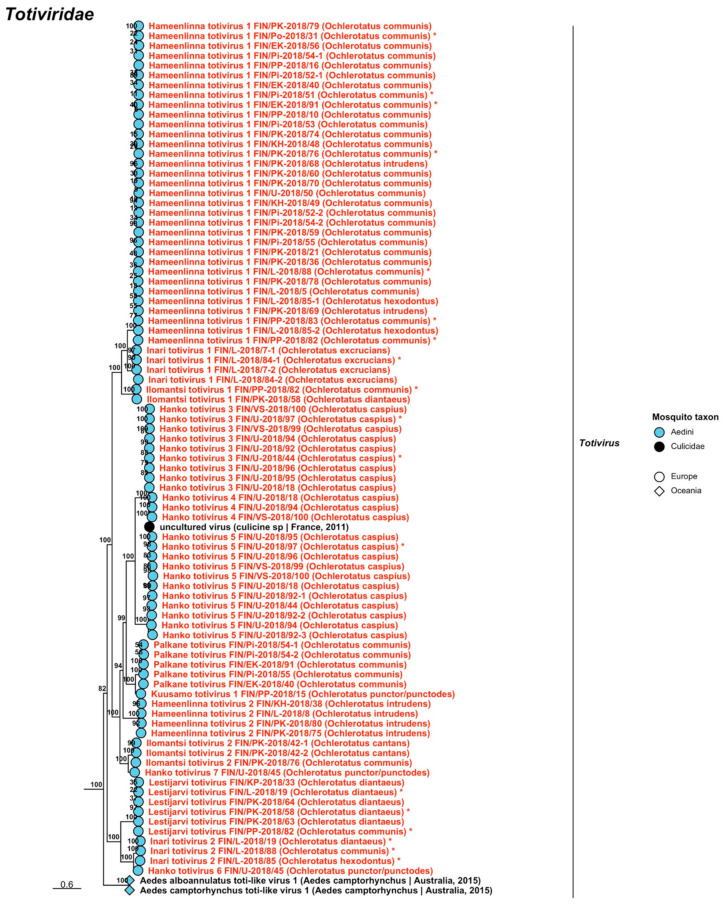
Maximum likelihood subtree of *Totiviridae*. Tentative novel viruses are displayed in red and the mosquito species from which they were derived are in parentheses. Sequences from GenBank are black and display the following information after the virus or species name: “(sampled organism(s)|collection country, collection year)”. Tip colours represent the tribe of mosquito from which viruses were obtained. Tip shape represents the continent or region from which the specimens were collected. Trees were constructed from amino acid sequences of virus polymerases >1000 nt, aligned with MAFFT and computed with IQ-TREE2 using ModelFinder and 1000 bootstraps. Asterisks denote that the complete genome was recovered.

**Figure 24 viruses-14-01489-f024:**
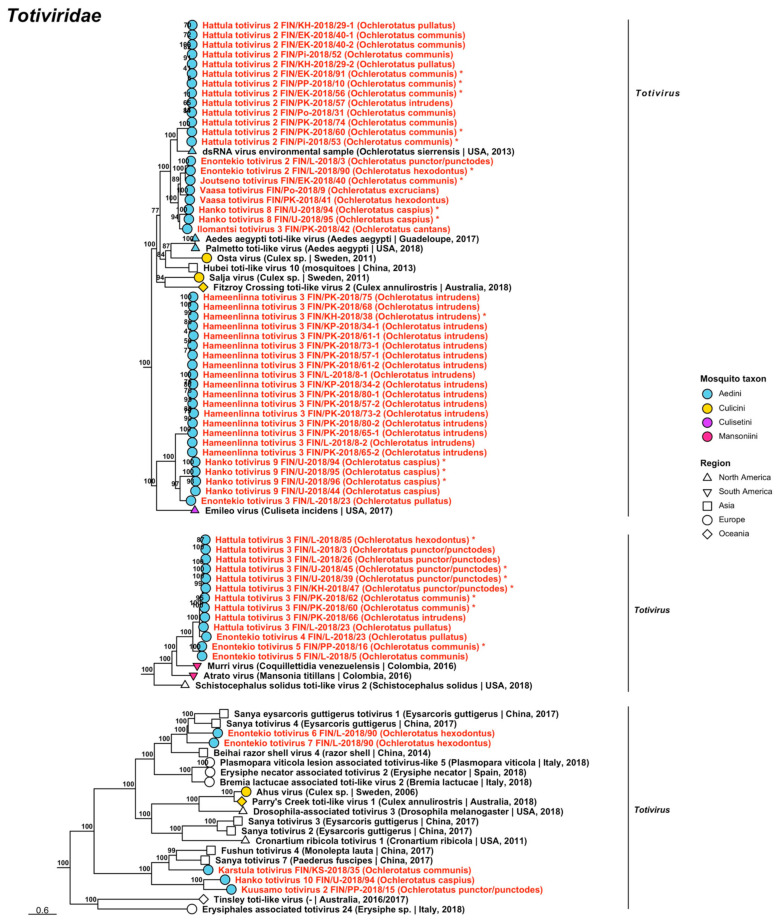
Maximum likelihood subtrees of *Totiviridae*. Tentative novel viruses are displayed in red and the mosquito species from which they were derived are in parentheses. Sequences from GenBank are black and display the following information after the virus or species name: “(sampled organism(s)|collection country, collection year)”. Tip colours represent the tribe of mosquito from which viruses were obtained. Tip shape represents the continent or region from which the specimens were collected. Trees were constructed from amino acid sequences of virus polymerases >1000 nt, aligned with MAFFT and computed with IQ-TREE2 using ModelFinder and 1000 bootstraps. Asterisks denote that the complete genome was recovered.

**Figure 25 viruses-14-01489-f025:**
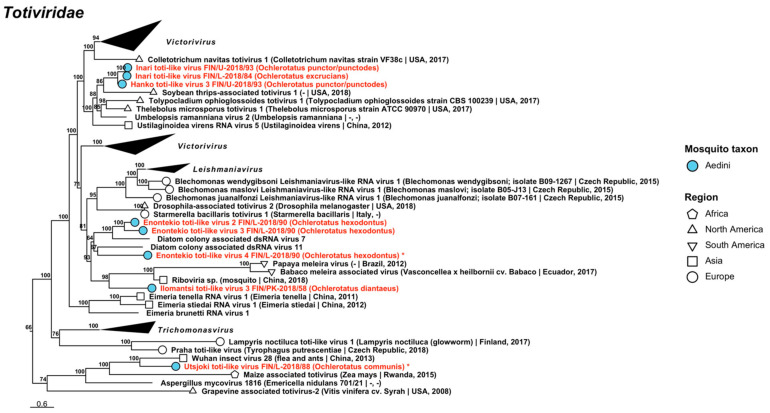
Maximum likelihood subtree of *Totiviridae*. Tentative novel viruses are displayed in red and the mosquito species from which they were derived are in parentheses. Sequences from GenBank are black and display the following information after the virus or species name: “(sampled organism(s)|collection country, collection year)”. Tip colours represent the tribe of mosquito from which viruses were obtained. Tip shape represents the continent or region from which the specimens were collected. Trees were constructed from amino acid sequences of virus polymerases >1000 nt, aligned with MAFFT and computed with IQ-TREE2 using ModelFinder and 1000 bootstraps. Asterisks denote that the complete genome was recovered.

Two totivirus sequences were Finnish strains of a previously described “uncultured virus” from France (GenBank accession: AGW51771.1), which have nearly identical protein identities. In the absence of a name, these viruses were therefore tentatively named “Hanko totivirus 3” and “Hanko totivirus 4” (amino acid identities: 96.76–97.35% and 94–94.5%). Complete genomes were detected for 16 aforementioned totiviruses, as follows: Enontekio totiviruses 2 and 5, Hameenlinna totiviruses 1 and 3, Hanko totiviruses 3, 5, 8 and 9, Hattula totiviruses 1 to 3, Ilomantsi totivirus 1, Inari totiviruses 1 and 2, Joutseno totivirus and Lestijarvi totivirus.

Seventeen novel toti-like viruses were recovered, which did not cluster within any established genera. As such, they were provisionally named “Enontekio toti-like virus 1 to 4”, “Hameenlinna toti-like virus”, “Hanko toti-like virus 1 to 3”, “Ilomantsi toti-like virus 1 to 3”, “Inari toti-like virus”, “Kustavi toti-like virus”, “Kuusamo toti-like virus”, “Palkane toti-like virus”, “Utsjoki toti-like virus” and “Vaasa toti-like virus”. Based on amino acid identity, Enontekio toti-like virus 1 was distantly similar to Uromyces totivirus D (GenBank accession: QED43018.1; amino acid identity: 43.84%), yet clustered with Uromyces potyvirus A (GenBank accession: QED42911.1). Enontekio toti-like viruses 2 and 3 had a low similarity to diatom-colony-associated dsRNA virus 7 (GenBank accession: YP_009553338.1; amino acid identities: 46.85% and 52.13%), while Enontekio toti-like virus 4 to diatom-colony-associated dsRNA virus 11 (GenBank accession: YP_009552795.1; amino acid identity: 35.11%). Seven of the viruses matched Nuyav virus (GenBank accession: QRW41699.1). These included Hameenlinna toti-like virus (amino acid identity: 68.09–73.6%), Hanko toti-like virus 2 (amino acid identity: 68.98%), Ilomantsi toti-like viruses 1 and 2 (amino acid identities: 76.01% and 66.82–71.27%), Kustavi toti-like virus (amino acid identity: 88.62–91.38), Palkane toti-like virus (amino acid identity: 72.59–76.81%) and Vaasa toti-like virus (amino acid identity: 68.53–69.73%). Ilomantsi toti-like virus 3 were distantly similar to an unnamed RNA virus (GenBank accession: QTW97791.1; amino acid identity: 37.9%). Hanko toti-like virus 1 was distantly related to Keenan toti-like virus (GenBank accession: QIJ70132.1; amino acid identity: 39.55%) and according to the phylogenetic analysis (Figure 21) to two Drosophila-associated totiviruses (GenBank accession: UFT26914.1 and UFT26909.1). Both Hanko toti-like virus 3 and Inari toti-like virus were related to Umbelopsis ramanniana virus 2 (GenBank accession: VFI65724.1; protein identities: 55.56% and 52.29%). One strain of the Inari toti-like virus also shared a low amino acid identity with Thelebolus microsporus totivirus 1 (GenBank accession: AZT88643.1; amino acid identity: 49.27%). However, our phylogenetic analysis suggested that both viruses were more related to soybean-thrips-associated totivirus 1 (GenBank accession: QQP18682.1). Lastly, Kuusamo toti-like virus had a high protein similarity to an unnamed totivirus (GenBank accession: QJI53453.1; amino acid identity: 86.43%), while Utsjoki toti-like virus had a low similarity to Wuhan insect virus 28 (GenBank accession: YP_009342430.1; amino acid identity: 34.55%). Complete genomes were sequenced for seven of our novel toti-like viruses. These were Enontekio toti-like virus 4, Hanko toti-like virus 2, Ilomantsi toti-like virus 2, Kustavi toti-like virus, Palkane toti-like virus, Utsjoki toti-like virus and Vaasa toti-like virus (GenBank accessions OP019849, OP019860, OP019883, OP019890, OP019895, OP019897 and OP019898, respectively).

### 3.2. Viruses by Mosquito Species

Variable numbers of pools, ranging from 1 to 35, were prepared for each mosquito species included in this study, with pooled material obtained from multiple collection locations. This made direct comparison of some results between species less meaningful, but each species was associated with multiple viruses.

*Ochlerotatus cantans*, which was the least represented species in the study with only one pool of 20 specimens collected in late June 2015 in Ilomantsi, PK, was found to have six viral sequences. These represented five novel species, and clustered within *Solemoviridae*, *Partitiviridae* and *Totiviridae* (Table 8).

*Ochlerotatus caspius* was represented with 11 mosquito pools comprised of 305 specimens collected from the southern, coastal regions of Uusimaa and Varsinais-Suomi in July and August 2017. In total, 76 viral sequences grouped into 26 virus species, and of these, 20 represented new virus species within *Chrysoviridae*, *Chuviridae*, *Iflaviridae*, *Negevirus*, *Partitiviridae*, *Phenuiviridae*, *Solemoviridae*, *Totiviridae* and *Xinmoviridae*. The seven previously described viruses fell within *Flaviviridae*, *Picornaviridae*, *Solemoviridae* and *Totiviridae* (Table 8). It was found to be virus-positive for Hanko virus in Uusimaa (FI 1010 and FI 1011), but not in Varsinais-Suomi (FI 988 and FI 1015) (see Figure 1).

*Ochlerotatus communis* was overrepresented in this study since it is one of the most common human-biting mosquitoes in Finland and is active across the summer months. As such, 35 pools were constructed, comprised of 866 specimens that were collected from around the country in May to August of 2015 and 2017. Inevitably, it also had the most unique viral sequences, with 179 that grouped into 62 species, of which 58 were novel. The three established viruses were Cordoba and Dezidougou viruses (*Negevirus*) and *Sindbis virus* (*Alphavirus*, *Togaviridae*). This is the first confirmed mosquito species to be associated with *Sindbis virus* in Finland. The single strain was found in Mekrijärvi, Pohjois-Karjala, an area where the only other Finnish mosquito-borne *Sindbis virus* strains have been recovered. The remaining 58 novel species belong to *Aliusviridae*, *Botybirnavirus*, *Chrysoviridae*, *Chuviridae*, *Iflaviridae*, *Negevirus*, *Partitiviridae*, *Phasmaviridae*, *Phenuiviridae*, *Qinviridae*, *Sedoreoviridae*, *Rhabdoviridae*, *Solemoviridae*, *Totiviridae*, *Virgaviridae* and *Xinmoviridae* (Table 8).

*Ochlerotatus diantaeus* was represented by six pools, comprised of 108 specimens, which were collected from northern, eastern and central Finland in June and July 2015. From these, 20 virus sequences were assembled, which grouped into 11 novel viruses in *Flaviviridae*, *Iflaviridae*, *Partitiviridae*, *Phenuiviridae*, *Sedoreoviridae*, *Solemoviridae* and *Totiviridae* (Table 8).

**Table 8 viruses-14-01489-t008:** Virus families detected by NGS, their host/vector associations and the (number of virus species/novel virus variants) by mosquito species. Brackets next to mosquito species names denotes the number of pools studied. Where a single digit is given, no novel viruses were detected for the given virus family/mosquito.

	Virus Family	No. Virus Variants	No. of Viruses	No. Novel Viruses	Host Associations	Oc. cantans (1)	Oc. caspius (11)	Oc. communis (35)	Oc. diantaeus (6)	Oc. excrucians (3)	Oc. hexodontus (8)	Oc. intrudens (14)	Oc. pullatus (2)	Oc. punctor/punctodes (11)	
+ ssRNA	*Endornaviridae*	2	2	1	Plants, fungi and oomycetes; host-specific [39].	-	-	-	-	-	-	-	-	2/1	No. of virus species per mosquito species/no. of novel virus species
	*Flaviviridae*	9	5	4	Arthropod-borne; mammalian hosts [40] mosquitoes [7,8,9,41,42,43].	-	1	-	1/1	-	2/2	1/1	-	-	
	*Iflavirus*	17	5	5	Arthropoda [44], mosquitoes, inc. Culex sp. [45].	-	2/2	2/2	1/1	-	1/1	2/2	-	1/1	
	*Negevirus*	41	7	4	Phlebotomine sandflies and mosquitoes [10,46].	-	1/1	5/3	-	1/1	5/3	1	-	3/2	
	*Permutotetraviridae*	6	1	1	Insecta: Setothosea asigna [47], Euprosterna elaeasa [48]. Fungus: Botrytis cinerea [49]. Mosquitoes [42].	-	-	-	-	1/1	1/1	-	-	1/1	
	*Picornaviridae*	5	2	0	Vertebrates (six of the seven classes) [50], Culex mosquitoes [51] and fleas [52].	-	2	-	-	-	-	-	-	-	
	*Quenyavirus*	2	1	1	Insecta: Crocallis elinguaria, Drosophila sp. and Lysiphlebus fabarum [53].	-	-	-	-	-	1/1	-	-	1/1	
	*Solemoviridae*	15	5	4	Plants (monocotyledons and dicotyledons) [54].	1/1	2/1	1/1	1/1	1/1	1/1	1/1	-	1/1	
	*Togaviridae*	1	1	0	Humans and nonhuman primates, mosquitoes, amphibians, arthropods, birds, equids, pigs, reptiles, rodents, salmonids and sea mammals; most are mosquito-borne [55]. Mosquitoes [43].	-	-	1	-	-	-	-	-	-	
	*Virgaviridae*	7	3	3	Plants, plasmodiophorids, nematodes and pollen [56].	-	-	1/1	-	-	3/3	-	-	1/1	
− ssRNA	*Aliusviridae*	1	1	1	Mosquitoes, Coleoptera, Hymenoptera, leafhopper [57] fleas [52]	-	-	1/1	-	-	-	-	-	-	
	*Aspiviridae*	1	1	1	Plants and plant-infecting fungi [58].	-	-	-	-	-	1/1	-	-	-	
	*Chuviridae*	14	3	3	Mosquitoes [59,60], earwigs, Odonata, ticks, cockroaches, snakes, fish [61].	-	2/2	1/1	-	-	1/1	1/1	1/1	-	
	*Phasmaviridae*	13	8	8	Mosquitoes [43], Hymenoptera, Hemiptera, Coleoptera [59].	-	-	6/6	-	-	-	2/2	-	-	
	*Phenuiviridae*	58	14	14	Mosquitoes [43,62], fleas [52], ticks, Coleoptera, phlebotomine sandflies, plants, humans [61].	-	1/1	7/7	1/1	-	3/3	2/2	-	3/3	
	*Qinviridae*	7	3	3	Insects [63,64], marine diatoms [57].	-	-	3/3	-	-	-	-	-	-	
	*Rhabdoviridae*	21	8	7	Vertebrates, invertebrates and plants [61]. Insect vectors infect vertebrates [65], mosquitoes [32].	-	-	3/3	-	2/1	3/3	-	1/1	1/1	
	*Xinmoviridae*	9	4	4	Mosquitoes [66,67], Odonata & Hymenoptera [68].	-	1/1	1/1	-	-	2/2	-	-	1/1	
	*Yueviridae*	1	1	1	Invertebrates (freshwater isopoda/sesarmid crab) [35].	-	-	-	-	-	1/1	-	-	-	
dsRNA	*Botybirnavirus*	8	1	1	Phytopathogenic fungi [69].	-	-	1/1	-	-	-	1/1	-	-	
	*Chrysoviridae*	5	3	3	Fungi and plants; possibly insects [70].	-	1/1	1/1	-	-	-	1/1	-	1/1	
	*Partitiviridae*	55	23	23	Plants, fungi, protozoa [71], mosquitoes [60].	1/1	2/2	9/9	1/1	2/2	10/10	3/3	2/2	3/3	
	*Sedoreoviridae*	10	4	4	Pathogenic viruses; arthropods [72,73,74], mammals, inc. humans [75,76], plants [77].	-	-	3/3	1/1	-	2/2	1/1	-	-	
	*Spinareoviridae*	1	1	0	Pathogenic viruses; mosquitoes [78], plants [79], fish [79], reptiles, birds and mammals [80].	-	-	-	-	-	1/1	-	-	-	
	*Totiviridae*	205	52	50	Fungi [81], protozoa [82], mosquitoes [60], fleas [52].	3/3	11/9	16/16	5/5	5/5	14/14	10/10	5/5	8/8	
**Totals**		514	159	147		5	26	62	11	12	52	26	9	27	

*Ochlerotatus excrucians* was represented by 99 specimens divided into three (unequal) pools, which were collected from northern and western Finland in June and July 2015. Twenty sequences were assembled, which grouped into 12 virus species, 11 of which were novel. Two strains of the previously described Ohlsdorf virus (*Rhabdoviridae*) were found from Inari in Lapland. The other 11 species grouped within *Negevirus*, *Partitiviridae*, *Permutotetraviridae*, *Solemoviridae* and *Totiviridae* (Table 8).

*Ochlerotatus hexodontus*, despite being represented by only eight pools comprised of 222 specimens, had the second most viruses of the nine mosquito species analysed herein. Most collections were made in Lapland in July 2015, with only one being from Ilomantsi, PK, eastern Finland in June 2015. In all, 78 virus sequences were assembled, which grouped into 52 species, of which 50 were novel. Two Negeviruses, Cordoba virus and Mekrijärvi negevirus, have previously been described. Novel viruses all belonged to *Aspiviridae*, *Chuviridae*, *Flaviviridae*, *Iflaviridae*, *Negevirus*, *Partitiviridae*, *Permutotetraviridae*, *Phenuiviridae*, *Quenyavirus*, *Rhabdoviridae*, *Sedoreoviridae*, *Solemoviridae*, *Spinareoviridae*, *Totiviridae*, *Virgaviridae*, *Xinmoviridae* and *Yueviridae* (Table 8).

*Ochlerotatus intrudens* was also well represented, with 14 pools assembled from 309 specimens, which were collected in June and July 2015, from around the country, but in particular from eastern Finland. Eighty-three virus sequences were assembled, which grouped into 26 species, of which 25 were novel: *Botybirnavirus*, *Chrysoviridae*, *Chuviridae*, *Flaviviridae*, *Iflaviridae*, *Negevirus*, *Partitiviridae*, *Phasmaviridae*, *Phenuiviridae*, *Sedoreoviridae*, *Solemoviridae* and *Totiviridae*. Two strains of the previously described Mekrijärvi negevirus were also sequenced (Table 8).

*Ochlerotatus pullatus* was the second least represented species in this study, with 46 specimens divided into two pools: one from Lapland and the other from Hattula, KH. Both were collected in 2017, in May and July. Ten virus sequences were detected, which grouped into nine novel species, which belong to *Chuviridae*, *Partitiviridae*, *Rhabdoviridae* and *Totiviridae* (Table 8).

*Ochlerotatus punctor*/*punctodes* was also represented by 11 pools, comprised of 358 specimens that were collected around Finland between May to August in 2015 and 2017. Forty-one strains were sequenced, which grouped into 27 species, of which 25 were novel. The novels species belong to *Chrysoviridae*, *Endornaviridae*, *Iflaviridae*, *Negevirus*, *Partitiviridae*, *Permutotetraviridae*, *Phenuiviridae*, *Quenyavirus*, *Rhabdoviridae*, *Solemoviridae*, *Totiviridae*, *Virgaviridae* and *Xinmoviridae*. The two established species were Hallsjon virus (*Endornaviridae*) and Cordoba virus (*Negevirus*) (Table 8). Short M glycoprotein sequences from Inkoo virus were also recovered in addition to the RdRp sequences used to assess species diversity.

## 4. Discussion

This is the first in-depth study of the viromes of mosquitoes from Finland. The aim was to investigate RNA viromes of identified *Ochlerotatus* mosquitoes, thereby ascertaining both the diversity of associated viruses and the potential vector associations of these mosquito species. RNA sequences were generated from nine identified species of female *Ochlerotatus* (2333 specimens), which were divided into 91 species-specific pools. Viral sequences were present in all mosquito pools, but only 90 contained sequences of RdRp greater than 1000 nucleotides and were included in further analyses. In total, 514 viral RNA sequences were identified that grouped into 159 species, 147 of which were likely to be novel. Strains for 12 viruses which had previously been described were sequenced, although only nine had been named when published: Hallsjon virus (*Endornaviridae*), Hanko virus (*Flaviviridae*), Cordoba virus, Dezidougou virus and Mekrijärvi negevirus (*Negevirus*), Jotan virus (*Picornaviridae*), Ohlsdorf virus (*Ohlsrhavirus*, *Rhabdoviridae*), Evros sobemo-like virus (*Solemoviridae*) and Sindbis virus (*Togaviridae*). The remaining three unnamed viruses were given suggested names in this study, to correspond with where the Finnish sequences originated: Hanko picorna-like virus (*Picornaviridae*), Hanko totivirus 3 and Hanko totivirus 4 (*Totiviridae*). Only two of these previously described viruses are currently recognised by the ICTV: *Sindbis virus* and Ohlsdorf virus. Three viruses, which had previously been detected using virus cell culture with Finnish mosquitoes, were sequenced and linked to named mosquitoes as follows: Hanko virus with *Oc. caspius*, Inkoo virus with *Oc. punctor*/*punctodes* and Sindbis virus with *Oc. communis*. These results affirm the high degree of viral diversity found in mosquitoes from Finland, despite only nine of the forty-three endemic mosquito species [11] being included in this study.

### 4.1. Classification and Interpretation of the Viruses Detected in this Study

Constructing phylogenetic trees of RNA viruses using RNA-dependent RNA polymerase sequences is a common practice to infer evolutionary relationships and classify newly detected viruses. This is because RdRp is a core viral protein which has conserved sequence motifs that make it a preferable gene to utilise in phylogenetic analyses. The nucleotide sequences of RNA viruses change constantly due to the high mutation rate in RNA viruses, but in contrast, amino acid sequences remain relatively stable and conserved [83]. Phylogenetic trees made from RdRps, therefore, tend to be more accurate compared to those made using other core proteins [84,85,86].

The putative viruses sequenced in this study were assigned as novel based on several criteria: (1) novel virus RNA dependent RNA polymerases submitted to NCBI BLASTx had to have an amino acid identity value lower than 90% compared to the most similar virus; (2) phylogenetic analyses were run to ascertain their evolutionary relationship with previously described viruses and their likely classification within virus families; (3) associated GenBank records from closely related taxa were examined and compared with potentially novel viruses. These included the country of sampling, collection date and the organism from which the virus was isolated, to infer their novelty. Finally, (4) the criteria set by ICTV, including sequence lengths, amino acid identity and clustering in phylogenetic trees were also considered. Additionally, we computed supplementary pairwise distances from the protein alignments to ascertain the novelty of the detected viruses (Appendix A). Certain viruses named in this study were on the borderline of being novel, since they came close to, but above, the 90% amino acid identity threshold with the closest described viruses. Such cases were noted where relevant. These viruses might indeed turn out to be Finnish strains of established viruses, but confirmation would require additional research including more sequence information on the related viral genetic diversity, especially from other geographical regions. All of the virus names proposed in this study are working names, as the final decision on their nomenclature and classification will be made by the ICTV.

Most (147) of the 159 viruses reported in this study were designated as novel since they had low similarities in RdRp amino acid identity with the most similar existing viruses (average 65.88%). The lowest amino acid identity was seen with Enontekio reovirus, which was only 29.6% similar to Operophtera brumata reovirus (*Spinareoviridae*), but low values were encountered many times throughout the analysis. This highlights the issues presented by “viral dark matter”, i.e., the lack of available sequences in databases to which viral sequences can be aligned [87], as well as the capacity of Lazypipe, the virus discovery and annotation pipeline established in our laboratory [18], to unravel viral sequences that are only remotely related to previously known viruses. Palkane botybirna-like virus (described in Section 3.1.3), shared a low average amino acid identity with another unclassified botybirna-like virus, Bremia lactucae associated dsRNA virus 1. Whether these two viruses are distant relatives, are ancestral viruses to the taxon, whether they can be classified as botybirnaviruses or whether they constitute a novel group of viruses remains undetermined. This study falls short of suggesting new virus genera, but it is likely that many of these sequences will form new genera in future revisions of the affected virus families.

Many more new viruses could have been named from the sequences obtained from this study but were excluded as their contigs fell below the 1000-nucleotide minimum length requirement that was set for any sequences to be considered for analysis. These discarded sequences formed approximately 75.5% of the total viral sequence data generated. In particular, short sequences of the pathogenic species Inkoo virus and *Chatanga virus* (*Peribunyaviridae*) were affected by these strict parameters, since short contigs containing polymerase, glycoprotein and nucleocapsid sequences were recovered.

A common pattern observed in the phylogenetic trees was that novel viruses clustered with available sequences of mosquito-derived viruses, inferring that these might be more mosquito-specific than insect-specific. Moreover, many novel viruses obtained during this study clustered with viruses that were sequenced from other mosquitoes, many of which belonged to *Aedini*, a cosmopolitan tribe of Culicidae with 1263 extant species and which includes 35% of all valid mosquito species [13]. One explanation could be that these viruses share a common ancestry [85]. Viral sequences that grouped within Iflaviridae, Aliusviridae and especially in *Flaviviridae* (*Flavivirus*) clustered near to or with insect-specific and Aedini-associated viruses. Several of the novel viruses which grouped within *Picornaviridae*, *Chuviridae* and *Chrysoviridae* were the first mosquito-associated viruses detected which belonged to tribe Aedini. These findings could be indicative of broader mosquito association ranges among these RNA virus families. Among the virus families which infect plants and fungi, e.g., *Alphapartitivirus*, the discovery of these novel Finnish viruses would suggest that *Ochlerotatus* mosquitoes (and most likely mosquitoes in general) act in some capacity as vectors for these viruses, whether by mechanical transmission or otherwise.

The proportion of totivirus sequences detected in all *Ochlerotatus* pools was very high in this study (Appendix A), despite them being viruses traditionally more associated with fungi and protozoa (https://ictv.global/taxonomy/ (accessed on 20 May 2022)). GenBank records show that totiviruses have been found in arthropods, plants, mammals and fish, thus indicate that these viruses might have a wider host range than is currently recognised by the ICTV (https://ictv.global/taxonomy/ (accessed on 20 May 2022)). Another factor to potentially explain the high prevalence of totiviruses could be that they are part of the core virome of *Ochlerotatus* species [88]. Either way, this study highlights the need for an expert group to subject *Totiviridae* to a critical review, since at present only 28 species belonging to five genera are currently officially recognised by the ICTV (https://ictv.global/taxonomy/ (accessed on 20 May 2022)), but in this study alone, 52 novel viruses were proposed. Similarly, partitiviruses were the next most represented species in this study, with 52 strains belonging to 23 viruses.

In recent years, most of the novel mosquito-borne viruses have been detected and reported from temperate and equatorial regions, since that is where most of the known mosquito-borne diseases are distributed [89]. The number of viromic studies from northern latitudes are increasing [31,32,90,91], but the uneven distribution of global research effort emphasises the importance of investigating mosquito viromes of these regions for more accurate information about the virosphere.

### 4.2. Reflections on the Methods and Their Impact upon Interpreting the Results

Since the lab work for this study was completed, a viromics study of Swedish mosquitoes was published in which a rinse step was added prior to homogenisation to remove surface contaminants from their specimens [90]. On reflection, this additional step would have been very beneficial to exclude any viruses which may have been mechanically transmitted to mosquitoes, or which were associated with bacteria/protozoa on the mosquito’s integument. Many of the viruses that were sequenced during this study, e.g., *Chrysoviridae*, *Endornaviridae*, *Solemoviridae*, *Totiviridae* and *Virgaviridae* are more traditionally associated with protozoa, plants or fungi than mosquitoes (see Table 8) [39,54,56,70,81,82]. Species of *Virgaviridae* even use pollen grains to disperse and infect new hosts [56]. The downside of viromics is not knowing the association of the novel viruses that are recovered, e.g., whether the mosquito happened to be covered in pollen grains which were in turn covered in viruses; whether the viruses were present in undigested gut contents; whether they infected the mosquito; or whether the mosquito is a vector for that virus, and so on.

Mosquitoes also have many interactions with other organisms in the environment. Some species are known to feed on honeydew, a sugar-rich excrement that some insects including ants (Hymenoptera) and aphids (Hemiptera) excrete after feeding on plants [92,93]. It would be interesting to determine, since some species actively seek out honeydew [93], if such interactions affect virus transmission between insects, particularly since so many plant-associated viruses were recovered in this study. In addition, three of the females that were included in the study, one *Oc. excrucians* (FIN/L-2018/007) and two *Oc. punctor*/*punctodes* (FIN/PP-2018/015 and FIN/L-2018/026) were noted to have parasitic or phoretic mites attached to them (nine mites, one and one mite, respectively). If truly phoretic, then the mites may just have been temporarily attached to the mosquito for dispersal, so they may not have been so relevant for interspecies transmission. If, however, they were parasitic, then the transfer of viruses between mites and mosquitoes is not out of the realm of possibility [94]. More work is required in the future to elucidate these relationships.

Taking these points into consideration, a further laboratory step would have also increased our understanding of which viruses may be vectored by the mosquitoes included in these analyses. Honey-baited nucleic acid cards, such as FTA^®^ Elute Cards (Whatman, Maidstone, UK), have been used in several studies in recent years in order to collect mosquito saliva, preserve any viral RNA, and ultimately sequenced to determine which viruses/virus species are present [95,96,97]. By first collecting mosquitoes, and then allowing them time to feed upon such cards either singularly, or in small groups, it would certainly be possible to refine results from metagenomic studies such as this one to see which viruses were common to the nucleic acid cards/saliva and mosquitoes, and which were only present in the mosquitoes, thereby determining which viruses have higher or lower likelihoods of being pathogenic. This could be then tested further using virus cell culture methods to isolate possible viruses on vertebrate or mosquito cells.

When mosquitoes were collected for this study, a note was made whenever a female was noticeably blood fed or gravid, but not if they had distended abdomens which looked as though they had recently fed upon plant juices. All but three of the females were not visibly blood-fed with only one female from pools FIN/L-2018/07 and FIN/L-2018/27 being confirmed as such, and one female which looked like it had possibly blood-fed several days earlier, in pool FIN/L-2018/88. Pool FIN/L-2018/07 contained seven viruses, one described from *Oc. cantans* in Germany, Ohlsdorf virus [32] (*Rhabdoviridae*), and six novel viruses belonging to *Partitiviridae* (1), *Permutotetraviridae* (1), *Solemoviridae* (1) and *Totiviridae* (3). Pool FIN/L-2018/27 only contained a single virus, Hanko iflavirus 1 (*Iflaviridae*). Pool FIN/L-2018/88 contained eight novel viruses, which belonged to *Chrysoviridae* (1), *Partitiviridae* (1), *Sedoreoviridae* (1), *Totiviridae* (4) and *Virgaviridae* (1). Competition between different viruses within mosquitoes might inhibit the replication or transmission of other viruses, resulting in the over representation of more competitive viruses [98,99,100]. Defective viral genomes have also been observed to inhibit replication or transmission of other viruses in mosquitoes [98,99], or in the case of identical or closely related viruses, the virus which manages to infect a host cell first might inhibit the replication of another via a process named “superinfection exclusion” [100].

Viromes of other mosquitoes which are native to Finland would also be of interest to study further in the future. This study only included nine of 43 (21%) currently recognised endemic species [11], and 38% of the pools were *Oc. communis*, creating a heavy bias to one species. Additional topics that would be of interest to explore further include the geographic and seasonal variations in the virome, as well as differences between males and females and at different developmental stages. Seasonal variation has been observed in *Aedes* (*Stegomyia*) *albopictus* [101] and *Culex* mosquitoes [102], though the core virome remains similar across different life stages in *Ae. albopictus* [103]. The sole focus on female mosquitoes might also limit virus discovery, akin to a study done with *Ae. albopictus* mosquitoes, in which Aedes iflavi-like virus genomes were only detected in a pool of male mosquitoes [104]. The authors do however note that the explanation for this is uncertain and that there might be other causal factors, such as the location of mosquito sampling [104].

### 4.3. Geographical Distribution of Viruses in Finland

This study has significantly increased the number of locations from which virus-positive mosquitoes have been collected in Finland. Prior studies have detected Hanko and Inkoo viruses from Uusimaa [3,4,8], Lammi virus from Kainuu, Pohjois-Karjala and Päijät-Häme [7,9], Chatanga virus from Kainuu (same location as for Lammi virus) and Pohjois-Karjala [5], Ilomantsi virus from Pohjois-Karjala [9], *Sindbis virus* from Pohjois-Karjala [1,2] and finally Mekrijärvi negevirus from Pohjois-Karjala [10]. In all, these viruses were found in only 4 of the 19 regions, and from only seven approximate locations, since six publications all included specimens from around Mekrijärvi in Pohjois-Karjala. The previously most northern mosquito-associated viruses in Finland were found in mosquitoes from around Sotkamo in Kainuu, approximately N64°08′, E28°23′ [5,9].

In contrast, this study included specimens which were collected from 49 collection efforts at 43 sites (min 1 km separation) in 11 regions and extended the sampling locations of virus-positive mosquitoes to the entire country (see Table A1 for a list of the proposed novel viruses by collection location, which can be compared with Figure 1). Moreover, the most northerly record of mosquito-positive viruses in Europe is now from collection FI 607 from Utsjoki in Lapland at N69°47′, E27°03′, where six viruses (Hattula chuvirus, Cordoba virus, Utsjoki negeviruses 1–3 and Hattula totivirus 1) were sequenced from pools FIN/L-2018/01 and FIN/L-2018/06. This overtakes the previous northernmost European record of a mosquito-associated bunyavirus from Masi, Norway, which was located at N69°26′, E23°39′ in the 1970s [105]. Two other collections which contribute to this study, FI 654 and FI 655 from Inari, Lapland, were also made further north than the Norwegian study (pools FIN/L-2018/07 and FIN/L-2018/19).

Hanko virus, an insect-specific virus which was first described from Finland [8], was sequenced in this study from mosquitoes that were collected near to the type locality in Hanko, Uusimaa. The four virus-positive pools all comprised *Oc. caspius*, which were collected in late August 2017. This is the first instance where a named mosquito species is confirmed to be associated with the virus. With future analyses, it will be interesting to see if Hanko virus is restricted to *Oc. caspius*, a halophilic/coastal mosquito species [16], or if it is also associated with mosquito species with larger distributions in Finland. Other specimens of *Oc. caspius* were included in the analysis, from Kustavi in Varsinais-Suomi from collections made in July and August 2017 (collection numbers FI 988 and FI 1015 in Figure 1), but the virus was not found therein.

A disproportionate number of pools were comprised of specimens which were collected from around Mekrijärvi, or in the municipality of Ilomantsi, Pohjois-Karjala. This was in part because the material in this study was all snap-frozen, identified and stored at −70 °C immediately following identification, to permit virus cell culture experiments. Such specialist facilities are located at a few field stations around Finland, which also explains why many collections were also made around the municipalities of Enontekiö and Utskoki in Lapland and in Hanko, Uusimaa. There were other factors, however, which influenced the decision to include material from eastern Finland. Prior to this study, Pohjois-Karjala was the only region where *Sindbis virus* [1,2], and one of only two locations from which Chatanga virus, has been found in Finnish mosquitoes [5], and vector species had not been confirmed. However, *Sindbis virus* has been detected in other parts of Finland in recent years [2]. Chatanga virus was not confirmed within the parameters of the study, but *Sindbis virus* was, as already mentioned, sequenced from a pool of *Oc. communis* mosquitoes. This sampling strategy did provide the first record for Inkoo virus in *Oc. punctor*/*punctodes* mosquito outside of Uusimaa, so from that perspective, it was very interesting, particularly as seroprevalence to California serogroup viruses is high amongst the Finnish population [106], but virus-positive mosquitoes have rarely been encountered. Since Ilomantsi, Hanko and Enontekiö had the majority of mosquito pools, they also had the most unique virus detections. The most widespread virus families in turn were *Totiviridae* and *Partitiviridae*. Totiviruses were detected in all sampled regions, which supports them being part of the core virome of *Ochlerotatus* mosquitoes. Similarly, partitiviruses were detected in all regions with the exception of Keski-Suomi (Central Finland) and Varsinais-Suomi (Southwest Finland). This, however, is very likely explained by sampling bias, since only one mosquito pool included specimens from Keski-Suomi and four pools from Varsinais-Suomi.

### 4.4. Brief Comparison with Other Virome Studies

Metagenomics studies published in recent years have identified diverse viromes in mosquitoes from around the world [31,34,37,43,45,66,90,107,108,109,110,111,112].These viromes appear to differ between species and can include anywhere from tens to hundreds of different virus species in a given sample, which often comprises several individuals of a species [31,34,37,43,45,66,90,107,108,109,110,111,112]. The nearest comparable study to Finland is a single-year, two-location study of 953 specimens of six mosquito species in Sweden [90]. It examined the viromes of *Coquillettidia richiardii*, *Oc. communis*, *Oc. annulipes*, *Oc. cantans*, *Culex pipiens* and *Cx. torrentium*, all species which are common to both Sweden and Finland, and two of which were common to both studies. They found viruses which belonged to multiple families, but ultimately there were none that were common to both studies [90]. They did, however, find viruses belonging to several families/orders which are yet to be detected in Finnish mosquitoes, including *Nodaviridae*, *Orthomyxoviridae*, *Tombusviridae* and *Articulavirales* [90]. Another Swedish study focused on comparing the viromes of *Culex pipiens* and *Cx. torrentium* collected from two locations over several years. They found 40 viruses (28 novel viruses) belonging to 14 families/orders: *Bunyavirales*, *Endornaviridae*, *Luteoviridae*, *Mogonegavirales*, *Negevirus*, *Nidovirales*, *Orthomyxoviridae*, *Partitiviridae*, *Picornaviridae*, *Qinviridae*, *Reoviridae*, *Togaviridae* (wrongly attributed to “*Alphaviridae*”, an invalid family) *Totiviridae* and *Virgaviridae* [31]. *Sindbis virus*, Hallsjon virus and Jotan virus were common to both this and the Swedish study [31], but viruses from *Luteoviridae* and *Orthomyxoviridae* were not sequenced in this study.

It is also of interest to compare these findings with those of other virus studies from Finland, to determine if other distant host taxa share any close virus associations and therefore explore the potential origins or pathogenicity of novel viruses. A study of glow-worms (Coleoptera: Lampyridae) amplified targeted RNA sequences from adults collected in central and southern Finland [113]. They recovered 11 novel viruses belonging to *Flaviviridae*, *Iflaviviridae*, *Tymoviridae*, *Bunyavirales*, *Rhabdoviridae*, *Partitiviridae*, *Totiviridae* and *Metaviridae*. Lampyris noctiluca flavivirus 1 grouped within the same clade as Lestijarvi flavi-like virus in the Flavivirus tree (Figure 2), in a branch separate from all other flavivirus sequences that were recovered in this study. Similarly, Lampyris noctiluca iflavirus 2 grouped in the same clade with Mekrijarvi iflavirus (Figure 3), in a branch away from all of the other iflaviruses. The glow-worm totivirus and rhabdovirus sequences also featured within the trees generated for this study, but not as closely as for the two named viruses. Since glow-worms are not haematophagous, have predatory larvae, do not feed as adults and are nonsocial, their associated viruses have limited sources [113]. The origins, host associations and pathogenicity of the novel viruses in this study are still to be determined.

### 4.5. Viruses Which Have Pathogenic Associations in Vertebrates

Two of the viruses which were sequenced in this study, Inkoo virus (*Peribunyaviridae*: *Orthobunyavirus*) and Sindbis virus (*Togaviridae*: *Alphavirus*) have known disease associations in Finland, and, although infrequent, can cause severe enough symptoms for patients to require hospitalisation [6,106,114]. These viruses have been detected in mosquitoes in previous Finnish studies, but it is worth mentioning that two mosquito species have now been implicated as being at the very least hosts for these viruses, if not vectors, *Oc. punctor*/*punctodes* and *Oc. communis*, respectively. The first isolations of Inkoo virus in the 1960s did include mixed pools containing *Oc. communis* and *Oc. punctor*/*punctodes* but now *Oc. punctor*/*punctodes* is confirmed as being virus positive. While most of the detected viral diversity has not been, and likely will not be, associated with pathogenic traits, it is nevertheless notable that without targeted sampling to capture outbreaks spatially or temporally, we have been able to detect sequences of the two previously well-established mosquito-borne pathogenic viruses in Finland.

*Reovirales* (until recently *Reoviridae*) is an order comprised of two families, *Sedoreoviridae and Spinareoviridae* (formerly *Sedoreovirinae* and *Spinareovirinae*), each of which has pathogenic virus species among their members. It is for this reason that the five novel *Reovirales* viruses in this study are of particular interest for future examination, to determine if hitherto unrecognised pathogenic mosquito-borne viruses are present in Finland. The proposed *Sedoreoviridae* viruses (Ilomantsi reovirus 1 to 4) all group with other viruses which were sequenced from mosquitoes, and are related to *Phytoreovirus*, which includes plant-pathogenic viruses based on a phylogenetic analysis. Valmbacken virus (see Figure 20), which is at the root of the novel virus cluster, is likely a mosquito-associated virus [31], indicating that the novel viruses potentially could have such associations. The single tentative *Spinareoviridae* virus, Enontekio reovirus, is distantly related to *Fijivirus*, which includes plant-infecting viruses that may spread via an insect vector. The sequences which group together in Figure 20 are all derived from insects.

## 5. Conclusions

This study, by using high throughput next-generation sequencing methods and an unbiased virus discovery pipeline, has vastly increased the knowledge of viruses associated with mosquitoes in Northern Europe and has confirmed the number of known mosquito-associated viruses and virus families in Finland from seven and four, to 159 and 25, respectively. Such a large increase in knowledge of the diversity of mosquito-associated viruses is certainly interesting and begins to enlighten the “viral dark matter”, but inevitably brings with it new questions and challenges. It also highlights the pressing need for additional study to bring relevance to the names and sequences presented herein, as well as to investigate arthropod viromes of northern regions more thoroughly. It is evident from the points we have raised that the floodgates have opened, and the real work of elucidating the relationships between mosquitoes, viruses, the environment and host species must now begin.

## Figures and Tables

**Figure 1 viruses-14-01489-f001:**
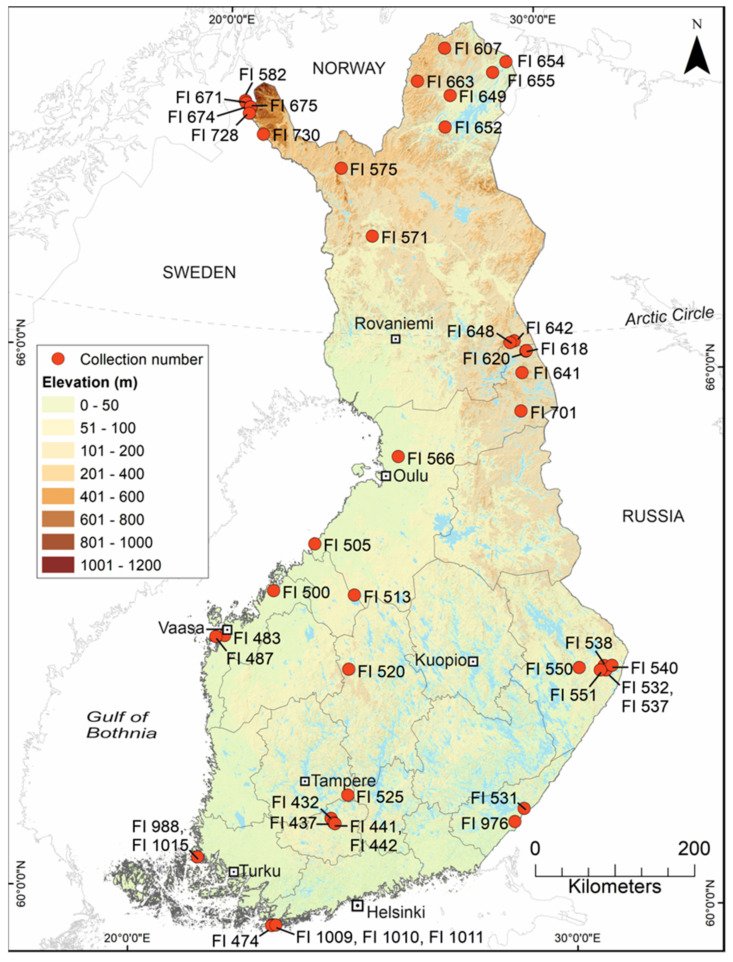
Locations of collections in Finland from which mosquitoes were pooled. Owing to the large numbers of mosquito pools from certain locations, the collection site number is given and not the pool number. Collections were made from a variety of unstandardised habitats while attempting to collect distribution data for all of Finland’s species. See Table 1 for the pool numbers, mosquito species and collection dates, and Table A1 for the viruses found at each location.

**Figure 2 viruses-14-01489-f002:**
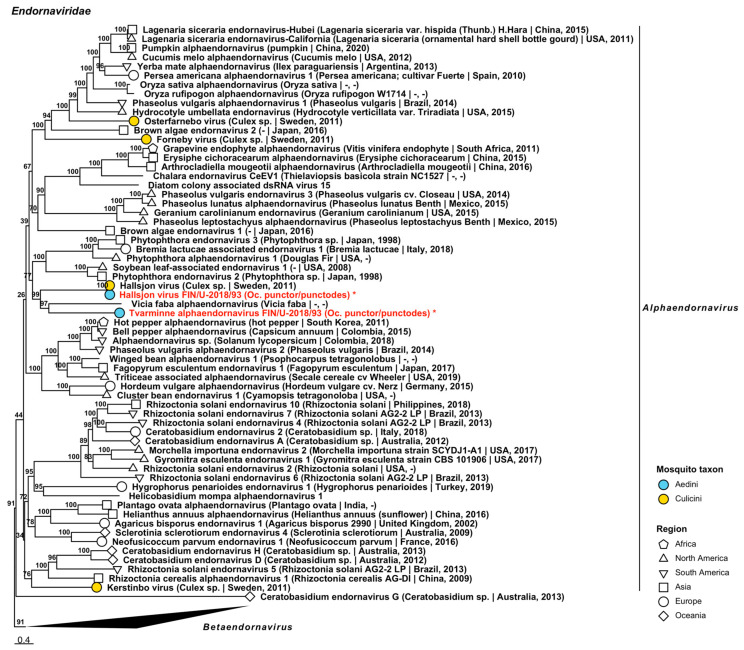
Maximum likelihood tree of *Endornaviridae*. Tentative novel viruses are displayed in red and the mosquito species from which they were derived are in parentheses. Sequences from GenBank are black and display the following information after the virus or species name: “(sampled organism(s)|collection country, collection year)”. Tip colours represent the tribe of mosquito from which viruses were obtained. Tip shape represents the continent or region from which the specimens were collected. Trees were constructed from amino acid sequences of virus polymerases >1000 nt, aligned with MAFFT and computed with IQ-TREE2 using ModelFinder and 1000 bootstraps. Asterisks denote that the complete genome was recovered.

**Figure 6 viruses-14-01489-f006:**
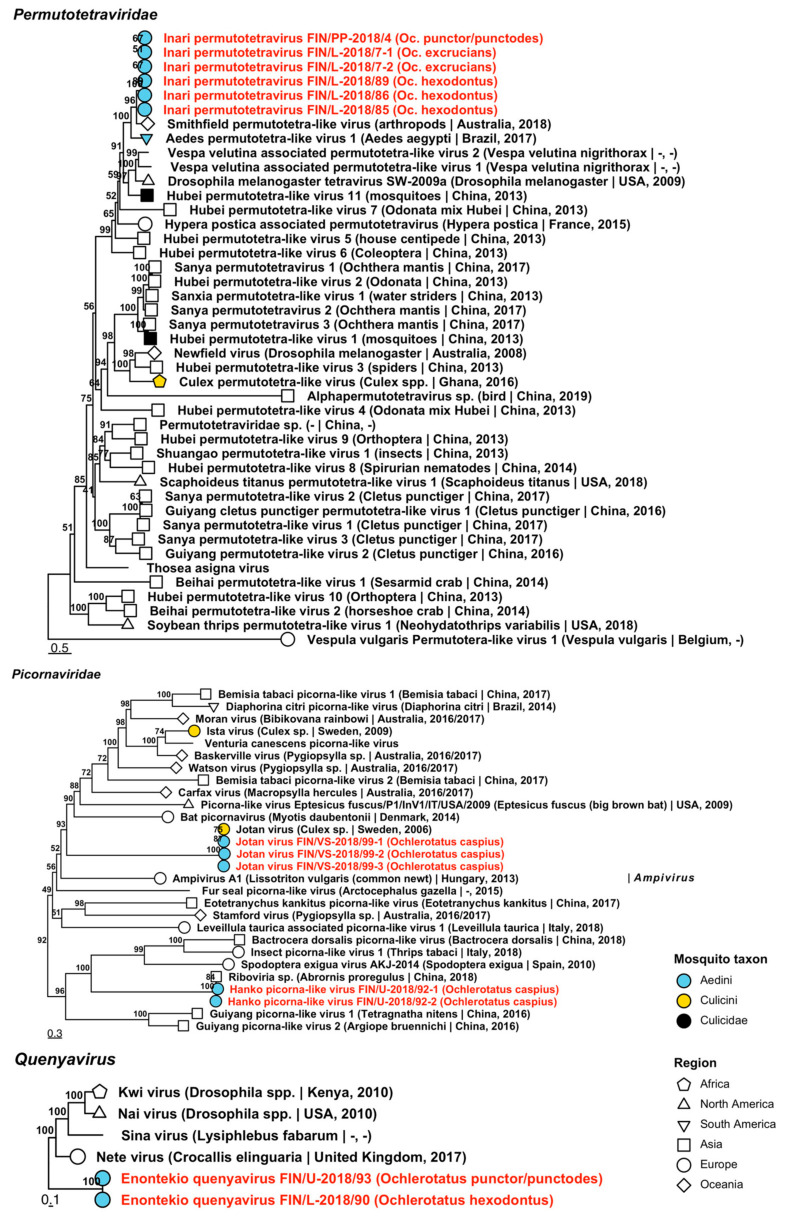
Maximum likelihood trees of *Permutotetraviridae*, *Picornaviridae* and *Quenyavirus*. Tentative novel viruses are displayed in red and the mosquito species from which they were derived are in parentheses. Sequences from GenBank are black and display the following information after the virus or species name: “(sampled organism(s)|collection country, collection year)”. Tip colours represent the tribe of mosquito from which viruses were obtained. Tip shape represents the continent or region from which the specimens were collected. Trees were constructed from amino acid sequences of virus polymerases >1000 nt, aligned with MAFFT and computed with IQ-TREE2 using ModelFinder and 1000 bootstraps.

## Data Availability

The produced sequence data is openly available in the NCBI BioProject database with the accession code PRJNA852425 and the virus sequences are publicly available in the NCBI GenBank database (see Table 2,Table 3,Table 4,Table 5,Table 6,Table 7 for accession codes).

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
