# Peer review of "Characterisation of the RNA Virome of Nine Ochlerotatus Species in Finland"

_viruses, 2022, doi:10.3390/v14071489_

Round 1
Reviewer 1 Report
Truong Nguyen et al collected mosquitoes in Finland between 2012 and 2018,and analyzed RNA viromes of nine commonly encountered Ochlerotatus mosquito species using next generation sequencing.This is important for understanding the local mosquito-borne virus community. Although the total number of mosquitoes were detected, the time of mosquito collection in a single collection site was not continuous and the number of mosquitoes was small.More specific comments are below.
Comments and Suggestions for Authors
1. It is true that some important viruses exist in specific mosquito species, such as Aedes aegypti, which is the carrier of many important viruses. Although Ochlerotatus is a widely distributed genus in Finland.Are there other mosquito species in addition to Ochlerotatus collected at each sampling site? The proportion of mosquitoes of different species at each sampling site? The ratio of male to female mosquitoes?
2. Mosquito activity and abundance are closely related to surrounding environment, season, weather (temperature) and other factors. mosquito species are also differences between the day and night. Is there a residential area or animal farm near the sampling site?These information is also unclear in the manuscript.
3. Sample pretreatment is very important in virome sequencing, which directly affects the depth and quality of sequencing. Which part will be sequenced, the whole mosquito, or parts of the mosquito (salivary glands or gut)? How to remove mosquito genome DNA contamination during RNA extraction?
4. The total number of reads per sample in the sequencing results, the average read length, should be supplemented. Different relative abundances of different viruses could be calculated by the ratio of the number of reads in total reads of different viruses.
5. Line 32 One sequence of Sindbis virus, which causes Pogosta disease in humans. If sufficient samples are available, PCR validation or virus isolation can be performed?
6. Line 154 The novel viruses discovered in this study were named according to the nearest town or municipality. However, some viruses have high homology with reported viruses in the genetic evolution tree, so it is inappropriate to name them as novel viruses, and some new viruses should be named in accordance with the rules of the ICTV.
7. Line 659 In all, 78 virus sequences were assembled, which grouped into 52 species, of which 50 were novel. How long is the sequence of each virus assembled? Is it just assembling RdRp sequences, or longer sequences? Is there a gap in the assembled sequence? Some short gaps can be filled by PCR amplification.
Author Response
Truong Nguyen et al collected mosquitoes in Finland between 2012 and 2018, and analyzed RNA viromes of nine commonly encountered Ochlerotatus mosquito species using next generation sequencing. This is important for understanding the local mosquito-borne virus community. Although the total number of mosquitoes were detected, the time of mosquito collection in a single collection site was not continuous and the number of mosquitoes was small. More specific comments are below.
1. It is true that some important viruses exist in specific mosquito species, such as Aedes aegypti, which is the carrier of many important viruses. Although Ochlerotatus is a widely distributed genus in Finland. Are there other mosquito species in addition to Ochlerotatus collected at each sampling site? The proportion of mosquitoes of different species at each sampling site? The ratio of male to female mosquitoes?
Response: The manuscript has now been amended to introduce the other mosquito genera which have been recorded from Finland, and to set the context of this study within the context of other studies of Finland’s mosquitoes. The collections that were made for this study were mainly made by one of the authors working alone, and were primarily made to collect specimens for distribution studies of each of the native species. While on intense field trips, and where possible (due to available resources – mainly time and freezer/dry ice availability), mosquitoes that were collected were processed in order to be suitable for one or more projects. That meant that a variable number of the mosquitoes that were collected from a site went to virus studies, from none to all of them (if a species had never been encountered before it would not have gone for virus studies unless collected in very high numbers). The sites that were chosen were therefore also highly variable.
Since the focus of this study was to explore and compare the viromes of identified Ochlerotatus species and to identify any potential vectors of known mosquito-borne or insect-specific viruses in Finland, many of the potential specimens were therefore not suitable for inclusion at this time. Additionally, since virus research in Finland has, up until now, focused on very restricted areas of Finland, it was of interest to include specimens from around the country such that geographical variation could be observed, if relevant. We are aware that the study could have included many other species, from many other sites, etc., but metagenomic studies are time and resource consuming endeavours, thus there was a trade-off to include identified adult female Ochlerotatus specimens to first get an idea which viruses were associated with them, since they have been implicated as vectors of the three mosquito-borne diseases in Finland. In time, we will follow up these results with studies of other species/life stages, and using refined methodology.
2. Mosquito activity and abundance are closely related to surrounding environment, season, weather (temperature) and other factors. Mosquito species are also differences between the day and night. Is there a residential area or animal farm near the sampling site? These information is also unclear in the manuscript.
Response: Data for the collection date and location are provided in table 1, which is meant to supplement Figure 1 (the map of collection sites). The region of the collection site can also be inferred from the code assigned to each mosquito pool. The specimens were collected around the entire country, as mentioned above, for the primary purpose of collecting mosquito distribution data (as has been amended in the manuscript). That meant a mix of habitats were sampled, from northern Lapland which is sparsely populated by humans to southern Finland where the population is much higher. The collection methodology was discussed in an earlier article (Culverwell et al., 2021) which was referenced in the methods section. Some of the collections were made during the night and others during the day, and while the collection time, as you rightly point out, makes a difference to the species that are potentially encountered, it should not affect the presence of viruses in the specimens that were collected. We must also point out that as Finland is in the northern hemisphere, June and July are summer months when mosquitoes are highly abundant (which has been amended in the manuscript), and that in the very far north collections were made during the Arctic summer when the sun did not set. The corrections in the manuscript do not mention daylight/night, but otherwise we hope that they address your points in a satisfactory way.
3. Sample pretreatment is very important in virome sequencing, which directly affects the depth and quality of sequencing. Which part will be sequenced, the whole mosquito, or parts of the mosquito (salivary glands or gut)? How to remove mosquito genome DNA contamination during RNA extraction?
Response: We have amended the methods to state that whole mosquitoes were homogenised, as this was accidentally omitted from the submitted manuscript. The homogenised supernatant from homogenised mosquitoes was used as a starting material for nucleic acid sample preparation and RNA NGS. In the manuscript we have already outlined several steps that were followed to remove genomic DNA from the samples (described in Methods section 2.3). In order to eliminate free DNA and RNA that was not protected by virions, the filtered mosquito homogenates were treated with nucleases, using micrococcal nuclease (DNA and RNA exonuclease) and benzonase (DNA and RNA endonuclease). Then RNA was extracted using a phenol-chloroform isothiocyanate phase separation method, using TRIzol reagent, optimising DNA free RNA extraction, and further, DNAse I treated to digest potential carry-over DNA from the organic phase during Trizol RNA extraction. Thereafter, the RNA was cleaned using Agencourt RNA Clean XP magnetic beads.
4. The total number of reads per sample in the sequencing results, the average read length, should be supplemented. Different relative abundances of different viruses could be calculated by the ratio of the number of reads in total reads of different viruses.
Response: The read information (reads per pool and the proportion of virus reads) has been included as supplementary figure 1. The requested read lengths and numbers have been compiled to supplementary table 1.
5. Line 32 One sequence of Sindbis virus, which causes Pogosta disease in humans. If sufficient samples are available, PCR validation or virus isolation can be performed?
Response: We attempted to isolate Sindbis from the virus positive pool, but the isolation was unsuccessful. The sequence that was obtained using NGS was sufficiently different from any other Sindbis virus strains that have been studied in our lab, so we are satisfied that this is not an instance of contamination. We agree that this is an interesting result to pursue and will explore the result further in the future.
6. Line 154 The novel viruses discovered in this study were named according to the nearest town or municipality. However, some viruses have high homology with reported viruses in the genetic evolution tree, so it is inappropriate to name them as novel viruses, and some new viruses should be named in accordance with the rules of the ICTV.
Response: We agree with this comment and are not proposing new virus species, though can see that the wording in the manuscript is confusing. We have amended the manuscript to remove this terminology. Formal taxonomic designation of truly novel viruses will be decided by ICTV, no doubt after additional studies, but the virus (not species) names that are proposed in this manuscript should conform to the ICTV rules of orthography.
7. Line 659 In all, 78 virus sequences were assembled, which grouped into 52 species, of which 50 were novel. How long is the sequence of each virus assembled? Is it just assembling RdRp sequences, or longer sequences? Is there a gap in the assembled sequence? Some short gaps can be filled by PCR amplification.
Response: The lengths of all assembled virus sequences have been added to supplementary table. Lazypipe, the pipeline we utilized for sequence assembly and analysis, assembles contigs of varying lengths from all regions (individual genes or polyproteins) of the viral genome, and from those we chose only sequences that were over 1,000 nucleotides long and contained complete polymerases for our dataset. The sequences/genomes analysed in this paper do not contain gaps.
Reviewer 2 Report
In the manuscript by Truong Nguyen and Culverwell et al., the authors report on a large sequencing project where they identified a number of viruses in Ochlerotatus species of mosquitoes in Finland. The authors present the results well in the main text, describing all aspects of the results, how they were collected, and analyses performed. However, some aspects of the manuscript remain unclear. Below are my suggestions for revisions:
Introduction – the goals of this study need to be described better with improved justification for undertaking the study. For example, in line 64, the authors state “particular interest” and this should be defined better. What exactly is of particular interest? Also, the major results should be summarised at the end of this section.
Figures 2-5 are completely illegible. These either need to be split up into multiple figures or only a single legible dendrogram should be shown.
“Viruses/MDPI” headings appear on random figures at page headings for unknown reasons.
Lines 403-404 – “Culex” and “Aedes” should be italicized. All Latin names should be italicized throughout the manuscript. Lines 403-404 is just one example that needs correcting.
Table 7, page 7 – random scribbles appear on this part of the table.
Lines 681-683 – why are these findings not described here? Either delete that statement or describe the results.
Author Response
In the manuscript by Truong Nguyen and Culverwell et al., the authors report on a large sequencing project where they identified a number of viruses in Ochlerotatus species of mosquitoes in Finland.  The authors present the results well in the main text, describing all aspects of the results, how they were collected, and analyses performed.  However, some aspects of the manuscript remain unclear.  Below are my suggestions for revisions:
Introduction – the goals of this study need to be described better with improved justification for undertaking the study.  For example, in line 64, the authors state “particular interest” and this should be defined better.  What exactly is of particular interest?  Also, the major results should be summarised at the end of this section.
Response: Thank you for drawing this to our attention. We have amended the introduction to elaborate upon several reasons why this study was undertaken, and agree that brevity had been taken too far in the initially submitted manuscript.
Figures 2–5 are completely illegible. These either need to be split up into multiple figures or only a single legible dendrogram should be shown.
Response: All figures which contribute to the manuscript have now been reinserted, either singularly, grouped together (in the case of less well represented families), or split into multiple figures (as for Totiviridae). All figures should now be legible.
“Viruses/MDPI” headings appear on random figures at page headings for unknown reasons.
Response: Thank you for bringing this to our attention. The template Word file that is provided by Viruses did not like the addition of landscape pages amongst the portrait pages and automatically formatted the pages this way. We have managed to rectify this problem now.
Lines 403-404 – “Culex” and “Aedes” should be italicized.  All Latin names should be italicized throughout the manuscript.  Lines 403-404 is just one example that needs correcting.
Response: Thank you for bringing to our attention the error for ‘Japanese Culex specimens’ as it was accidentally missed. The other mosquito species names that are discussed in the manuscript are all in italics, from what we can see. When mosquito genus and species names appear as part of a virus name, they do not follow the rules of the International Code of Zoological Nomenclature (which states that they should be in italics), but rather the International Committee on Taxonomy of Viruses, which follows a separate set of rules for displaying virus names and virus species names. When using the virus name it is incorrect to format the mosquito species, which forms part of that name, in italics, as it is not the mosquito which is being named, but the virus. See ICTV:
“A virus name should never be italicized, even when it includes the name of a host species or genus, and should be written in lower case. This ensures that it is distinguishable from a species name, which otherwise might be identical. The first letters of words in a virus name, including the first word, should only begin with a capital when these words are proper nouns (including host genus names but not virus genus names) or start a sentence. Single letters in virus names, including alphanumerical strain designations, may be capitalized. In most texts, virus names are used much more frequently than species names and may, therefore, be abbreviated. Examples:
- Isolates of dengue virus 2 were obtained ....
- Detection of West Nile virus in human serum ....
- Salmonella phage SE1 was isolated ....
- Sida ciliaris golden mosaic virus (SCGMV) causes ....”
Table 7, page 7 – random scribbles appear on this part of the table.
Response: We have looked at both page 7 and in table 7 and cannot see any scribbles in the manuscript. We are not sure to what this comment refers.
Lines 681-683 – why are these findings not described here?  Either delete that statement or describe the results.
Response: The statement has been deleted.
Round 2
Reviewer 1 Report
-
I agree with the author's modification.